

**A European aerosol phenomenology-6: Scattering properties of atmospheric aerosol**
**particles from 28 ACTRIS sites**
Marco Pandolfi[1], Lucas Alados-Arboledas[2], Andrés Alastuey[1], Marcos Andrade[3], Begoña
Artiñano[4], John Backman[5,6], Urs Baltensperger[7], Paolo Bonasoni[8], Nicolas Bukowiecki[7], Martine
Collaud Coen[9], Sebastian Conil[10], Esther Coz[4], Vincent Crenn[11,12], Vadimas Dudoitis[13], Marina
Ealo[1], Kostas Eleftheriadis[14], Olivier Favez[15], Prodromos Fetfatzis[14], Markus Fiebig[16], Harald
Flentje[17], Patrick Ginot[18], Martin Gysel[7], Bas Henzing[19], Andras Hoffer[20], Adela Holubova
Smejkalova[21,22], Ivo Kalapov[23], Nokos Kalivitis[24,25], Giorgos Kouvarakis[24], Adam Kristensson[26],
Markku Kulmala[5], Heikki Lihavainen[6], Chris Lunder[16], Krista Luoma[5], Hassan Lyamani[2], Angela
Marinoni[8], Nikos Mihalopoulos[24,25], Marcel Moerman[19], José Nicolas[27], Colin O'Dowd[28], Tuukka
Petäjä[5], Jean-Eudes Petit[11,15], Jean Marc Pichon[27], Nina Prokopciuk[13], Jean-Philippe Putaud[29],
Sergio Rodríguez[30], Jean Sciare[11,a], Karine Sellegri[27], Dimiter B. Stamenov[23], Erik Swietlicki[26],
Gloria Titos[1,2], Thomas Tuch[31], Peter Tunved[32], Vidmantas Ulevicius[13], Aditya Vaishya[28,33], Milan
Vana[21,22], Aki Virkkula[5], Stergios Vratolis[14], Ernest Weingartner[7,b], Alfred Wiedensohler[31], and
Paolo Laj[5,8,18]
[1] Institute of Environmental Assessment and Water Research, c/ Jordi-Girona 18-26, 08034, Barcelona,
Spain
[2] Andalusian Institute for Earth System Research, IISTA-CEAMA, University of Granada, Granada 18006,
Spain
[3] Atmospheric Physics Laboratory, ALP, UMSA, Campus Cota Cota calle 27, Endifico FCPN piso 3, La Paz,
Bolivia
[4] Centro de Investigaciones Energéticas, Medioambientales y Tecnológicas, CIEMAT, Unidad Asociada en
Contaminación Atmosférica, CIEMAT-CSIC, Avda. Complutense, 40, 28040 Madrid
[5] University of Helsinki, UHEL, Division of Atmospheric Sciences, PO BOX 64, FI-00014, Helsinki, Finland
[6] Finnish Meteorological Institute, FMI, Erik Palmenin aukio 1, FI-00560, Helsinki, Finland
[7] Paul Scherrer Institut, PSI, Laboratory of Atmospheric Chemistry (LAC), OFLB, , 5232, Villigen PSI,
Switzerland
[8] Institute of Atmospheric Sciences and Climate, ISAC, Via P. Gobetti 101, I-40129, Bologna, Italy
[9] Federal Office of Meteorology and Climatology, MeteoSwiss, Chemin de l'aérologie, 1530 Payerne,
Switzerland
[10] ANDRA – DRD – Observation Surveillance, Observatoire Pérenne de l'Environnement, Bure, France
[11] LSCE-Orme point courrier 129 CEA-Orme des Merisiers, 91191 Gif-sur-Yvette, France
[12] ADDAIR, BP 70207 - 189, rue Audemars, 78530, Buc, France
[13] SRI Center for Physical Sciences and Technology, CPST, Sauletekio ave. 3, LT-10257, Vilnius, Lithuania
[14] Institute of Nuclear & Radiological Science & Technology, Energy & Safety, N.C.S.R. "Demokritos",
Athens, 15341, Greece
[15] Institut National de l'Environnement Industriel et des Risques, Verneuil en Halatte, 60550, France
[16] Norwegian Institute for Air Research, Atmosphere and Climate Department, NILU, Instituttveien 18, , 2007,
Kjeller, Norway
[17] Deutscher Wetterdienst, Met. Obs. Hohenpeissenberg, DE-82383 Hohenpeissenberg, Germany
[18] Univ. Grenoble-Alpes, CNRS, IRD, INPG, IGE F-38000 Grenoble, France
[19] TNO B&O, Princetonlaan 6, 3584TA, The Hague, The Netherlands
[20] MTA-PE Air Chemistry Research Group, Veszprém, P.O. Box 158, H-8201, Hungary
[21] Global Change Research Institute AS CR, Belidla 4a, 603 00, Brno, Czech Republic
[22] Czech Hydrometeorological Institute, Na Sabatce 17 , 143 06, Praha, Czech Republic.
[23] Institute for Nuclear Research and Nuclear Energy, Basic Environmental Observatory Moussala, 72
Tsarigradsko Chaussee Blvd, 1784-Sofia, Bulgaria
[24] Environmental Chemical Processes Laboratory, Dept. of Chemistry, Univ. of Crete, Heraklion, 71003,
Greece
[25] Institute for Environmental Research & Sustainable Development, National Observatory of Athens (NOA),
I. Metaxa & Vas. Pavlou, 15236 Palea Penteli, Greece



[26] Lund University, Department of Physics, P. O. Box 118, SE-22100, Lund, Sweden
[27] CNRS-LaMP Université Blaise Pascal 4, Avenue Blaise Pascal 63178 Aubiere Cedex, France
[28] School of Physics and Centre for Climate & Air Pollution Studies, Ryan Institute, National University of
Ireland Galway, University Road, Galway, Ireland
[29] EC Joint Research Centre, EC-JRC-IES, Institute for Environment and Sustainability, Via Enrico Fermi
2749, 21027, Ispra, Italy
[30] Agencia Estatal de Meteorologia, AEMET, Izaña Atmospheric Research Center, La Marina 20, E-38071,
Santa Cruz de Tenerife, Spain
[31] Leibniz Institute for Tropospheric Research, (TROPOS), Permoserstraße 15, 04318, Leipzig, Germany
[32] Department of Environmental Science and Analytical Chemistry (ACES) and the Bolin Centre for Climate
Research, Stockholm University, SE-106 91 Stockholm, Sweden
[33] Space Physics Laboratory, Vikram Sarabhai Space Centre, ISRO, Thiruvananthapuram – 695022, India.
[a] now at: EEWRC, The Cyprus Institute, Nicosia, Cyprus
[b] now at: Institute for Aerosol and Sensor Technology, University of Applied Sciences (FHNW), Windisch,
Switzerland





## 1  Abstract

This paper presents the light scattering properties of atmospheric aerosol particles measured over
the past decade at 28 ACTRIS observatories, located mainly in Europe. The data include particle
light scattering ($\sigma_{sp}$) and hemispheric backscattering ($\sigma_{bsp}$) coefficients, scattering Ångström
exponent (SAE), backscatter fraction (BF) and asymmetry parameter ($g$). A large range of $\sigma_{sp}$ was
observed across the network. Low $\sigma_{sp}$ values were on average measured in Nordic and Baltic
countries and in Western Europe whereas the highest $\sigma_{sp}$ were measured at regional sites in
eastern and central Europe. In these regional areas the SAE was also high indicating the
predominance of fine-mode particles. On average, the SAE was lower in the Nordic and Baltic,
western and southern countries suggesting a lower fraction of fine-mode particle compared to
central and eastern Europe. An increasing gradient of $\sigma_{sp}$ was observed when moving from
mountain to regional and to urban sites. Conversely, the mass-independent SAE and $g$ parameters
did not show the same gradient. At all sites, both SAE and $g$ varied greatly with aerosol particle
loading. The lowest values of $g$ were always observed under low $\sigma_{sp}$ indicating a larger contribution
from particles in the smaller accumulation mode. Then, $g$ steeply increased with increasing $\sigma_{sp}$
indicating a progressive shift of the particle size distribution toward the larger end of the
accumulation mode. Under periods of high particle mass concentrations, the variation of $g$ was less
pronounced whereas the SAE increased or decreased suggesting changes mostly in the coarse
aerosol particles mode rather than in the fine mode. The station placement seemed to be the main
parameter affecting the intra-annual variability. At mountain sites, higher $\sigma_{sp}$ was measured in
summer mainly because of the enhanced boundary layer influence. Conversely, less horizontal
and vertical dispersion in winter led to higher $\sigma_{sp}$ at all low altitude sites in central and eastern
Europe compared to summer. On average, these sites also showed SAE maxima in summer (and
correspondingly $g$ minima). Large intra-annual variability of SAE and $g$ was observed also at
Nordic and Baltic countries due to seasonal-dependent transport of different air masses to these
remote sites. Statistically significant decreasing trends of $\sigma_{sp}$ were observed at 5 out of 13 stations
included in trend analyses. The total reductions of $\sigma_{sp}$ were consistent with those reported for $PM_{2.5}$
and $PM_{10}$ mass concentrations over similar periods across Europe.



## 1. Introduction

Atmospheric aerosol particles are recognized as an important atmospheric constituent with demonstrated effects on climate and health. Radiative forcing of aerosol particles, estimated as −0.9 [−1.9 to −0.1] W/m$^2$ (IPCC, 2014), has two competing components: a cooling effect from most particle types and a partially offsetting warming contribution from black carbon (BC) particle light absorption of solar radiation. The aerosol cooling is the dominant effect; thus aerosol particles are counteracting a substantial portion of warming effect from well mixed greenhouse gases (GHGs). This process is driven by the scattering properties of most aerosol particle types (e.g. secondary sulfate and nitrate particles, mineral and organic matter), which reduces the amount of solar radiation reaching the Earth surface reflecting it back to space and modifying the Earth's radiative balance. However, the high temporal and spatial variability of atmospheric aerosol particles, due to the wide variety of aerosol sources and sinks, their short and variable lifetime (hours to weeks in the planetary boundary layer) and spatial non-uniformity, contribute to the largest uncertainty in the estimation of the total radiative forcing. Reducing these uncertainties is mandatory in view of global warming experienced over the past 50 years. In fact, there are evidences suggesting that the observed (and projected) decrease in emissions of anthropogenic aerosol particles, in response to air quality policies, would eventually unmask the global warming (Rotstayn et al., 2013). Thus, current emission controls could increase climate warming while improving air quality (e.g. Stohl et al., 2015). The measurements of aerosol particle optical properties such as light scattering and absorption, together with measurements of physical and chemical properties, are fundamental in order to better understand the current conflict involving a trade-off between the impacts of aerosols on environmental health and Earth's climate. Several international projects are providing in the last decades important information on the atmospheric particle properties worldwide. Near-surface in-situ observations of aerosol particle properties are performed worldwide under the GAW/WMO program completed with policy-oriented programs such as IMPROVE (Interagency Monitoring of Protected Visual Environments; http://vista.cira.colostate.edu/Improve/) in USA or EMEP (European Monitoring and Evaluation Programme; http://www.emep.int/) in Europe. Additional information specifically targeting advanced aerosol particle properties are obtained in Europe using information from the European research infrastructure ACTRIS (Aerosols, Clouds, and Trace gases Research InfraStructure; http://www.actris.eu) or from short-term RTD projects such as EUCAARI (European Integrated Project on Aerosol Cloud Climate and Air Quality Interactions; http://www.cas.manchester.ac.uk/resprojects/eucaari/). The implementation of the GAW program in Europe is performed under ACTRIS for the advanced observation of aerosol particle properties. ACTRIS is providing harmonized measurement of different (physical, chemical and optical) aerosol properties in a systematic way at major observation sites in Europe. More than 60 measuring sites worldwide are currently providing ground-based in-situ aerosol particle light scattering measurements (EBAS database; www. http://ebas.nilu.no/) and the number has increased substantially in the last decade.



The objective of this work is to integrate the total aerosol light scattering coefficient ($\sigma_{sp}$) and
hemispheric backscattering coefficient ($\sigma_{bsp}$) measurements performed over several years at the
ground based in-situ ACTRIS stations. A total of 28 stations (25 European + 3 non-European) are
included in order to document the variability in near-surface aerosol particle light scattering across
the ACTRIS network. Moreover, at some of the ACTRIS stations more than 10 years of $\sigma_{sp}$ data
are available allowing us to perform trends analysis. The study of the trend of $\sigma_{sp}$ is important given
that decreasing or increasing trend of $\sigma_{sp}$ would mirror the effectiveness of the air quality control
measures. In fact, many studies have shown that the concentrations of particulate matter (PM),
and other air pollutants such as sulfur dioxide ($SO_2$) and carbon monoxide (CO), have clearly
decreased during the last 20 years in many European Countries (Barmpadimos et al., 2012;
Cusack et al., 2012; EEA, 2013; Querol et al., 2014; Guerreiro et al., 2014; Pandolfi et al., 2016,
Tørseth et al., 2012, among others).
Previous studies presenting multi-site ground-based in-situ aerosol particle optical measurements
were for example performed by Delene and Ogren (2002), Collaud Coen et al. (2013) and Andrews
et al. (2011). Delene and Ogren (2002) reported the variability of aerosol particle optical properties
at four North American surface monitoring sites. Collaud Coen et al. (2013) presented long term
(>8-9 years) aerosol particle light scattering and absorption measurements performed at 24
regional/remote observatories, among which 5 of them are located in Europe. Andrews et al.
(2011) reported the aerosol particle optical measurements performed at 12 (4 located in Europe)
mountain top observatories. Thus, the number of papers reporting aerosol particle optical
properties measured at different sites is rather scarce and unfortunately almost inexistent outside
Europe and the United States.
Our work is focused mainly on European observatories aiming at a representative phenomenology
of aerosol particle light scattering coefficient measured at ACTRIS stations. Thanks to the
establishment of European monitoring networks and/or research projects five papers have been
published related with aerosol phenomenology in Europe: Van Dingenen et al. (2004) and Putaud
et al. (2004) on the physical and chemical, respectively, characteristics of particulate matter (PM)
at kerbside, urban, rural and background sites in Europe; Putaud et al. (2010) on the physical and
chemical characteristics of PM measured at 60 sites across Europe; Cavalli et al. (2016) on the
harmonized concentrations of carbonaceous aerosol at ten regional background sites in Europe;
Zanatta et al. (2016) presenting a climatology of BC optical properties at nine European regional
background sites. The importance of these studies and of the present work relies on the evidence
that a reliable assessment of the physical, chemical and optical properties of aerosol particles at
European scale is of crucial importance for an accurate estimation of the radiative forcing of
atmospheric aerosols. This work is the first European phenomenology study dedicated to the light
scattering properties of aerosol particles measured in-situ at near-surface ground-based
observatories. Moreover, the trend analyses presented can be used to evaluate how the European
mitigation strategies adopted to improve air quality affected the aerosol particle optical properties.





In fact, starting from the $\sigma_{sp}$ measurements performed at the ACTRIS observatories, three intensive aerosol particle optical parameters can be estimated, namely the scattering Ångström exponent (SAE), the backscattering fraction (BF), and the asymmetry parameter (*g*). These intensive properties do not depend on the PM mass concentration and are directly related to aerosol particles properties such as size, shape, size distribution and chemical composition. The SAE can be considered a proxy for the aerosol particle size range with higher (lower) SAE associated to predominance of fine (course) aerosol particles (e.g. Seinfeld and Pandis, 1998; Esteve et al., 2012; A. Valenzuela et al., 2015 among others). The BF and *g* parameters are calculated quantities that influence the variability of the radiative forcing efficiency and that represent the angular light scattering of aerosol particles. For computational efficiency, the angular light scattering is often represented by a single value (BF, $\sigma_{sp}/\sigma_{bsp}$ or *g*) (Ogren et al., 2006). For some of the ACTRIS data used in this work, trends of these intensive aerosol particle optical properties are investigated as well.

## 2. Experimental

### 2.1 Atmospheric Observatories

Figure 1 shows the location of the observatories which are grouped based on their geographical location as performed in other European phenomenology studies (e.g. Putaud et al., 2010). Observatories information and measurements periods are summarized in Table 1. The observatories are also divided in five different categories depending on their placement in each geographical sector. Mountain: includes those observatories located at more than 1 km above sea level; coastal: includes observatories located close to the sea coast; regional: includes those observatories mostly affected by regional sources and closer to large pollution sources compared to continental sites; continental: comprise observatories located in remote continental areas; urban/sub-urban: includes observatories located in urban background or suburban areas.

Nordic and Baltic stations are represented by Birkenes (BIR, Norway; regional), Hyytiälä (SMR, Finland; regional), Pallas (PAL, Finland; continental), Vavihill (VHL, Sweden; continental), and Preila (PLA, Lithuania; coastal). Western European sites are Puy De Dome (PUY, France; mountain), Mace Head (MHD, Ireland; coastal), Cabauw (CBW, The Netherlands; regional), SIRTA (SIR, France; suburban), and Observatory Perenne (OPE, France; regional). Central European stations are Jungfraujoch (JFJ, Switzerland; mountain), Hohenpeissenberg (HPB, Germany; mountain), Melpitz (MPZ, Germany; regional), Ispra (IPR, Italy; semi regional), Mt. Cimone (CMN, Italy; mountain) and Košetice (KOS, Czech Republic; regional). Eastern European stations are Beo Moussala (BEO, Bulgaria; mountain) and K-Puszta (KPS, Hungary; regional). South-western European stations are represented by Izaña (IZO, Spain; mountain), Montsec (MSA, Spain; mountain), Montseny (MSY, Spain; regional), Madrid (MAD, Spain; sub-urban), and Granada



(UGR, Spain; urban) whereas south-eastern European stations are Athens (DEM, Greece; sub-
urban) and Finokalia (FKL, Greece; coastal). Finally, Arctic and Antarctic stations are Zeppelin
(ZEP) and Troll (TRL), respectively. Another non-European mountain station included is Mt.
Chacaltaya (CHC, Bolivia; mountain). The altitude of the mountain stations considered here ranges
between 985 m at HPB to 5240 m at CHC. Some of the mountain stations included in this
investigation have been already included in the work by Andrews et al. (2011), namely IZO, JFJ,
CMN, and BEO. Moreover, FKL, HPB, JFJ, MHD, and PAL stations have been included in the
study by Collaud Coen et al. (2013). Both studies presented in-situ aerosol particle optical
measurements performed at these stations. Main results from the previous investigation are
summarized in the results section.
At JFJ, HPB, IPR (Central Europe), UGR (southwestern Europe), MHD (western Europe), PAL and
SMR (Nordic and Baltic), at least 10 years of data are available for trend analysis. However, in
order to improve the spatial coverage, trends are also studied at CMN, MPZ (central Europe), IZO
(southwestern Europe), PUY (western Europe), KPS and BEO (eastern Europe), where >8-9 years
of data are available (cf. Table 1). The stations included in this work report the data to ACTRIS and
GAW/EMEP, consequently the data are quality assured given that the nephelometer instruments
are run following the ACTRIS/GAW standards (WMO-GAW Report, 2016) and regularly inter-
compared.
**2.2 Scattering measurements**
**2.2.1 Instruments**
The measurements of $\sigma_{sp}$ and $\sigma_{bsp}$ included in this study were obtained from TSI and Ecotech
integrating nephelometers (Table 1). These optical instruments measure the amount of light
scattered by particles in the visible spectrum and provide $\sigma_{sp}$ and $\sigma_{bsp}$ coefficients of sampled
aerosols. Most used nephelometer models are the TSI3563 and the Ecotech AURORA3000, both
providing $\sigma_{sp}$ and $\sigma_{bsp}$. Model TSI3563 measures $\sigma_{sp}$ and $\sigma_{bsp}$ at 450, 550, and 700 nm whereas the
Ecotech AURORA3000 measures at 450, 525 and 635 nm. Other used models are the M9003
from Ecotech (SIR and CMN) and the RR (Radiance Research) nephelometer model M903 (FKL)
measuring $\sigma_{sp}$ at 520 nm and 532 nm, respectively. Due to the non-homogeneity of the light source
of the model M9003, the light source was changed at SIR in 2013 with the AURORA3000 light
source and at CMN in 2009 with an opal glass light source. The detailed description of the main
characteristics and working principle of the integrating nephelometers can be found e.g. in Müller
et al. (2011) for the Ecotech AURORA3000 and in Anderson and Ogren (1998) for the model TSI
3563. Following the ACTRIS and WMO-GAW recommendations, the nephelometers are regularly
calibrated using span gas and zero-adjusted using particle-free air. Recommended quality
assurance procedures during on-site operation as described in GAW (WMO/GAW, 2016),
guarantee the quality and comparability of the data. Moreover, most of the integrating



nephelometers involved in ACTRIS have undergone performance checks at scheduled times at the
World Calibration Center for Aerosol Physical properties of ACTRIS/GAW.
**2.2.2 Data treatment**
**2.2.2.1 Truncation correction**
Data from integrating nephelometers used here are corrected for non-ideal illumination of the light
source (deviation from Lambertian distribution of light) and for truncation of the sensing volumes in
the near-forward (around 0-10º) and near-backward direction (around 170-180º) (Müller et al.,
2009 and Anderson and Ogren, 1998). Correction schemes have been provided by Müller et al.
(2011) for the Ecotech AURORA3000 and by Anderson and Ogren (1998) for the TSI3563. Both
methods provide a simple linear correction scheme based on the scattering Ångström exponent
(SAE) determined from raw nephelometer data to correct for the size distribution-dependent
truncation error. It has been demonstrated that for an aerosol particle population with a single
scattering albedos (SSA) greater than 0.8 this simple correction scheme provides a suitable
quantification of the truncation error (Müller et al., 2011). However, for SSA < 0.8 a correction
scheme based on particle number size distribution should be used (Müller et al., 2011). The
aerosol particle light scattering data used here are corrected for non-ideal illumination and for
truncation by the data providers or in this work. This information is reported in Table S1 of the
Supporting Material. Only at SIR, FKL, and CMN, $\sigma_{sp}$ data are not corrected for truncation because
$\sigma_{sp}$ at these observatories was measured at one wavelength. At CMN, the 3-$\lambda$ TSI3563 is operative
since 2014 (cf. Table 1). However, not correcting for truncation doesn't prevent from comparing
non-corrected $\sigma_{sp}$ with truncation corrected $\sigma_{sp}$. The truncation correction increases with particle
size being more important for coarse aerosol particles. For example, using the SAE calculated at
CMN for the years 2014-2015 and the correction scheme provided for the TSI3563 by Anderson
and Ogren (1998), the difference between non-corrected $\sigma_{sp}$ and corrected mean $\sigma_{sp}$ is lower than
7% at this site.
**2.2.2.2 Relative humidity**
The integrating nephelometer measurements within ACTRIS and WMO-GAW should be performed
at low relative humidity (RH) in order to avoid enhanced scattering due to water uptake of aerosol
particles and to make measurements comparable. For the Ecotech integrating nephelometers the
RH threshold can be set by using a processor-controlled automatic heater inside the
nephelometer. At some mountain sites where whole air is sampled (cf. Table 1), the natural
temperature difference between outside and inside air dries cloud droplets to the aerosol phase
when a cloud is present at the station. RH is also controlled by de-humidifying in the inlet pipe as





reported in the GAW report 226 to ensure sampling RH of less than 40%. This recommendation intends to make the data comparable across the network, which otherwise would be a strong function of the highly variable sample RH. Currently, at the majority of ACTRIS observatories, the aerosol particles light scattering measurements are performed at RH lower than 40%. However, given that at some stations the 40% RH threshold is sometimes exceeded, we selected in this work a RH threshold of 50% in order to improve the data coverage. Estimating the aerosol particle light scattering enhancement due to an increase of RH from 40% to 50% is difficult using the data available here because $\sigma_{sp}$ measurements at RH>40% are not evenly distributed over the measurement periods. In fact, at the majority of the stations RH higher than 40% is registered mostly in summer. However, the scattering enhancement due to a change in RH between 40% and 50% should be small and will not exceed around 3-5% even for more hygroscopic particles (e.g. Fierz-Schmidhauser et al., 2010a,b). Table S2 in the Supporting Material reports the number of RH hourly data reported at each observatory and the number and % of hourly RH data >50%. The frequency distributions of measured RH are shown in Figure S1. Finally, $\sigma_{sp}$ and $\sigma_{bsp}$ data reported to EBAS and used in this work are referenced to standard T (273.15 ºC) and P (1013 hPa) conditions.

**2.2.2.3 Available wavelengths**

In this work we present and discuss the $\sigma_{sp}$, BF and $g$ measurements obtained using the green wavelength of the integrating nephelometers. The available wavelengths ranged from 520 nm (2 stations, CMN and VHL) to 550 nm (18 stations). Other used wavelengths are 525 nm (6 stations), 532 nm (used at FKL until 2010; cf. Table 2). An exception is SIR where only $\sigma_{sp}$ at 450 nm is available. The measurements of $\sigma_{sp}$ reported here are not adjusted to 550 nm which generally is the most used wavelength (e.g. Andrews et al., 2011) because of the different data availability of $\sigma_{sp}$ and SAE at the measuring stations. As discussed in the following Sections SAE is calculated for $\sigma_{sp}$ data higher than 0.8 Mm$^{-1}$, thus leading to different data coverage for $\sigma_{sp}$ and SAE and thus preventing the adjustment of all measured $\sigma_{sp}$ to 550 nm. Moreover, SAE is not available at FKL and SIR (and at CMN until 2014) thus preventing any wavelength adjustment at these stations. Using the mean SAE calculated at those stations where $\sigma_{sp}$ is measured at different wavelength than 550 nm (cf. Tables S4 and S5 in Supporting material), we estimate differences in $\sigma_{sp}$ lower than 6% after adjusting to 550 nm. At FKL and SIR, where SAE is not available and assuming a SAE of 1.5, the difference by adjusting to 550 nm is 4.9% at FKL and 26% at SIR, respectively. The higher difference at SIR is due to the fact that measurements at this station are performed at 450 nm. Finally, at CMN the effect of the adjustment of $\sigma_{sp}$ to 550 nm (from 520 nm) using a mean SAE of 2 (cf. Table S5) is lower than 10%.




**2.2.3 Calculation of aerosol particle intensive optical properties**
In addition to the direct $\sigma_{sp}$ and $\sigma_{bsp}$ measurements obtained with the above detailed
instrumentation, the following aerosol intensive parameters are calculated from hourly-averaged in-
situ data.
The scattering Ångström exponent (SAE) characterizes the wavelength dependency of $\sigma_{sp}$ and it
can be calculated as follows (with $\lambda_1 > \lambda_2$):

$$SAE = -\frac{\log\left(\sigma_{sp}^{\lambda_1}/\sigma_{sp}^{\lambda_2}\right)}{\log\left(\lambda_1/\lambda_2\right)}$$
(Eq. 1)

Here, the SAE is calculated as linear estimation of $\sigma_{sp}$ measured at the three available
wavelengths. The SAE depends on particle size distribution. It takes values greater than 2 when
the light scattering is dominated by fine particles (radii ≤ 0.5 $\mu$m as e.g. in Schuster et al. (2006)),
while it is lower than one when the light scattering is increasingly dominated by coarse particles
(Seinfeld and Pandis, 1998; Schuster et al., 2006).
The asymmetry parameter ($g$) (Andrews et al., 2006; Delene and Ogren, 2002) describes the
probability that the radiation is scattered in a given radiation and it is defined as the cosine-
weighted average of the phase function. Thus, $g$ gives information on the amount of radiation that
a particle can scatter in the forward direction compared to the backward direction. Theoretically,
the values of $g$ can range from −1 for only back scattering to +1 for complete forward scattering
(0º), with a value of 0.7 commonly used in radiative transfer models (Ogren et al., 2006). The g
parameter can be estimated from the backscatter fraction (BF) which is the ratio between $\sigma_{bsp}$ and
$\sigma_{sp}$ (Andrews et al., 2006):
$$g = -7.14(BF)^3 + 7.46(BF)^2 - 3.96(BF) + 0.9893$$
(Eq. 2)

**2.2.4 Data coverage**
Table S3 in the Supporting Material reports the number of hours and data availability for each
atmospheric observatory. The data coverage reported in Table S3, refers to scattering and
backscattering measurements performed at RH<50%. The data coverage for the extensive
measured aerosol particle optical properties ($\sigma_{sp}$ and $\sigma_{bsp}$) is generally high ranging from around
60% to 95%. Exception are $\sigma_{sp}$ measurements in the blue (450 nm) and in the red (700 nm) and
$\sigma_{bsp}$ measurements at CMN where the three wavelengths nephelometer was implemented starting
from 2014. Consequently, also SAE and $g$ has low data coverage at CMN. Moreover, lower data
coverage (< 40%) was registered at PLA and VHL. The data coverage for the intensive aerosol
particle optical properties (SAE and $g$) is generally lower compared to the data coverage of $\sigma_{sp}$ and



$\sigma_{bsp}$. This is because the intensive optical properties are calculated from hourly $\sigma_{sp}$ and $\sigma_{bsp}$ data higher than 0.8 Mm$^{-1}$ to avoid noise in the calculations. As a consequence, the data coverage of the intensive properties is lower at those stations measuring usually low $\sigma_{sp}$ and $\sigma_{bsp}$ (e.g. mountain and remote sites). For example, at JFJ the SAE and $g$ data coverage is of around 54% and 22%, respectively. At TRL these values are even lower, with 21% and 1%, respectively. However, as reported in Table S3, at the majority of the stations the data coverage of SAE and $g$ is higher than 60%.

## 3. Results/Discussion

### 3.1 Variability of $\sigma_{sp}$

Figure 2 shows the box-and-whiskers plots of $\sigma_{sp}$ measured at the stations included in this investigation. Table S4 and Figure S2 in the Supplementary Material report, respectively, the statistics of $\sigma_{sp}$ (mean, standard deviation, minimum and maximum values and 5$^{th}$, 25$^{th}$, 50$^{th}$, 75$^{th}$, and 95$^{th}$ percentiles) and frequency and cumulative frequency distributions.

In Fig. 2, data are grouped based on their geographical location (cf. Fig. 1) and ordered based on their placement, from mountain sites to urban sites. In each geographical sector, an increasing gradient of $\sigma_{sp}$ is generally observed when moving from mountain to regional and to urban sites. Thus, $\sigma_{sp}$ measured at mountain sites is always lower compared to measurements performed at other placements (coastal to urban) even if exceptions are observed in some sectors. A large range of $\sigma_{sp}$ coefficients is observed across the network ranging from median values lower than 10 Mm$^{-1}$ to values higher than 40 Mm$^{-1}$. The observed variation is consistent with the differences in particulate matter (PM) mass concentrations, PM chemical composition and particle number concentration observed across Europe as described for example by Putaud et al. (2010) and Asmi et al. (2011). Figure 3 shows the relationship between the mean particle number concentration measured at different stations during 2008 – 2009 reported in Asmi et al. (2011) and the mean $\sigma_{sp}$ measured over the same period (where available). As reported in Fig. 3, a good correlation is observed between N50 (mean/median particle number between 50 nm and 500 nm) and N100 (mean/median particle number between 100 nm and 500 nm) and mean $\sigma_{sp}$. Overall, the lowest $\sigma_{sp}$ is on average measured at remote stations either because of: a) their altitude, for example JFJ located in central Europe at more than 3500 m a.s.l. and CHC in Bolivia at around 5300 m a.s.l., or b) because their large distance from pollution sources, for example the coastal ZEP and TRL stations and some regional/continental sites in the Nordic and Baltic sector such as BIR, SMR and PAL. The Arctic (ZEP) and Antarctic (TRL) monitoring stations are located in undisturbed environments with minimal influence from the local settlement since these are located above the inversion layers. The PAL station (Nordic and Baltic) is located in a remote continental area and the low $\sigma_{sp}$ measured at this site are mainly due to the absence of large local and regional pollution



sources (e.g. Aaltonen et al., 2006). Conversely, higher $\sigma_{sp}$ (medians > 40 Mm$^{-1}$) are on average registered at more polluted sites such as some urban sites in southern Europe (UGR and DEM), some regional sites in eastern and central Europe (KPS and IPR, respectively), and one coastal site in the Nordic and Baltic sector (PLA). Finally, at all stations included in this work, the skewness of $\sigma_{sp}$ distributions (cf. Table S4) is higher than one and ranged between 1.4 at PLA and 10.6 at TRL (skewness calculated from hourly averaged data). Positive skewness is usually observed for positive defined parameters having a frequency distribution with a pronounced right tail indicating the presence of high positive values. Figure S2 in the Supporting Material shows the frequency and cumulative frequency distributions for $\sigma_{sp}$ for each station evidencing the presence of these right tails.

### 3.1.1 $\sigma_{sp}$ at mountain observatories

Differences can be observed among stations with similar placements but different geographical locations. Among the mountain stations higher mean $\sigma_{sp}$ is on average measured at HPB and IZO (cf. Table S4). HPB station is likely to be more influenced by the PBL than other mountain stations due to its lower altitude (Nyeki et al., 2012; Collaud Coen et al., 2017), whereas IZO is largely influenced by Saharan dust outbreaks transporting dust toward the station (e.g. Rodriguez et al., 2011) thus increasing $\sigma_{sp}$. In fact, at IZO the median value of $\sigma_{sp}$ is among the lowest measured at these mountain sites (around 7 Mm$^{-1}$; cf. Table S4) indicating that sporadic but extremely intense pollution episodes due to Saharan mineral dust outbreaks strongly affect the mean $\sigma_{sp}$ at this station. The lowest median $\sigma_{sp}$ at mountain sites are on average measured at JFJ probably due to the higher altitude of this station compared to other mountain stations included in this work and/or the distance from important pollution sources. Moreover, Collaud Coen et al. (2017) reported a low PBL influence at this site due to the location of the station in a dominant position in the whole mountainous massif. CHC registers higher median $\sigma_{sp}$ compared to IZO or JFJ despite its location at around 5300 m a.s.l. likely due to the influence of the emissions from the city of La Paz (3600 m a.s.l.) located around 30 km far from CHC and the local topography which facilitates the uplift of air masses toward the CHC observatory (Collaud Coen et al., 2017).

### 3.1.2 $\sigma_{sp}$ at regional/continental observatories

Regional sites present a large variability in $\sigma_{sp}$ coefficients across Europe with the lowest values measured at BIR and SMR (Nordic and Baltic) and the highest at IPR (central Europe) and KPS (eastern Europe). At both IPR and KPS, the frequent wintertime episodes linked to strong stable air with thermal inversion strongly affect the level of pollution at these sites (e.g. Putaud et al., 2014; Molnár et al., 2016). On the other side, it is known that the IPR station, even though it lies several tens of kilometers away from large pollution sources, is located in an area (the Po Valley) which is



one of the most polluted regions in Europe (e.g. van Donkelaar et al., 2010). Among the
continental sites, VHL registers on average higher $\sigma_{sp}$ compared to PAL and compared to BIR and
SMR regional sites likely because VHL is located closer to the continent and it is consequently
more affected by polluted continental air masses. Moreover, the emissions from densely populated
areas of Helsingborg and Malmö and the city of Copenhagen located 25 km to the west, 50 km to
the south, and 45 km to the south-east, respectively, could also explain the relatively high $\sigma_{sp}$
measured at VHL (Kecorius et al., 2016). The $\sigma_{sp}$ values at regional level in Western Europe (OPE
and CBW) are on average higher compared to those measured in the Nordic and Baltic regions
and lower compared to those measured at regional level in south Europe (MSY).

### 11    3.1.3 $\sigma_{sp}$ at urban observatories

Among the urban background sites, lower $\sigma_{sp}$ are measured at MAD and SIR compared to DEM
and UGR. Low $\sigma_{sp}$ at MAD during the period presented here (only 2014 available for MAD) could
be related to the reduced formation of secondary nitrate aerosols due to the limitation in the
availability of ammonia in this urban environment (Revuelta et al., 2014). However, it should be
considered that winter episodes with high secondary nitrate concentrations are not uncommon in
Madrid and we are presenting here only one year of measurements for this station. On the other
hand, secondary inorganic aerosol concentrations recorded at SIR sub-urban observatory can be
considered as representative of a large geographical zone, given the rather flat orography of the
Parisian basin. At UGR, the accumulation, mainly in winter, of fine particles from traffic, domestic
heating and biomass burning explains the relatively higher $\sigma_{sp}$ (e.g. Lyamani et al., 2012; Titos et
al., 2017). Traffic emissions, high formation of secondary sulfate and organic aerosols in summer
together with the transport of dust from Africa are the main reasons explaining the high $\sigma_{sp}$ at DEM
where high $PM_{2.5}$ and $PM_{10}$ are usually measured compared to other important Mediterranean
cities (e.g.: Diapouli et al., 2017; Eleftheriadis et al., 2014; Karanasiou et al. 2014; Querol et al.,

26    2009).

### 28    3.1.4 $\sigma_{sp}$ at coastal observatories

The PLA coastal station registered $\sigma_{sp}$ values higher compared to both other Nordic and Baltic
stations and other coastal sites (e.g. MHD and FKL) and amongst the highest in Europe. Kecorius
et al. (2016) have shown that ship emissions in the Baltic Sea contribute strongly to pollution levels
at PLA and that up to 50% of particles arriving at PLA are generated by processes and emissions,
including shipping, taking place in areas upwind the station. Moreover, Asmi et al. (2011)
presented some similarities in particle number concentrations measured at PLA with those
measured at some central European sites such as IPR due to the influence from multiple source



areas (cf. Fig. 3). It should be noted however, that the period with available $\sigma_{sp}$ measurements is
very short at PLA (cf. Table 1 and Figure 7) and the data coverage is also low (cf. Table S3).
Consequently, more measurements at this site are needed in order to confirm the $\sigma_{sp}$ values
reported here. The other two coastal stations (MHD and FKL) register median $\sigma_{sp}$ values in the
upper range of $\sigma_{sp}$ measured across the network mostly due to the contribution of marine aerosol in
winter and mineral dust in summer at MHD and FKL, respectively (cf. Paragraph 3.5).
**3.2 Variability of SAE**
Figure 4 shows the box-and-whiskers plots of SAE calculated at the different stations. Table S5
and Figure S3 in the Supplementary Material report the statistics of SAE and frequency and
cumulative frequency distributions, respectively. It should be noted that the comparison of SAE
among the different stations could be slightly biased by the different particle size cuts upstream the
integrating nephelometers used in this work (cf. Table 1). Currently, all ACTRIS integrating
nephelometers measure whole air or $PM_{10}$. Whole air is currently measured at mountain sites
(BEO, CMN, JFJ, PUY, CHC) and one coastal (MHD), and two urban/suburban (UGR and SIR)
observatories (cf. Table 1). At some stations, the inlet was changed from whole air to $PM_{10}$ at a
given time, namely at OPE, FKL, and TRL. Given the lower scattering efficiency of aerosol
particles larger than 10 $\mu$m, no important differences in the SAE should be expected between
aerosol particles sampled with whole air and $PM_{10}$ cut-off. At other stations the inlet was changed
during the measurement period from a cut-off lower than 10 $\mu$m (1 $\mu$m at KPS; 2.5 $\mu$m or 5 $\mu$m at
PAL, MSA and MAD) to $PM_{10}$. For PAL (where a median SAE of around 1.8 was measured; cf.
Table S5), Lihavainen et al. (2015a) assumed that the inlet changes (from $PM_5$ to $PM_{2.5}$ in 2005
and from $PM_{2.5}$ to $PM_{10}$, cf. Table 1) had only minor effects on scattering, because the number
concentration of coarse particles is very low at PAL. Similarly, KPS observatory registers among
the highest SAE observed in the network (median value around 2) suggesting an aerosol particle
size distribution dominated by fine particles. Consequently, the inlet change from $PM_1$ to $PM_{10}$ at
KPS had probably a minor effect on SAE. Finally, two stations (MSA and MAD) changed the inlet
from $PM_{2.5}$ diameter cut-off to $PM_{10}$. For these two Southern European stations the inlet change
may have had an effect on SAE especially during Saharan dust outbreaks, which are however
sporadic events. Thus, despite the differences in the particle diameter cut-off the comparison
between the different stations in terms of SAE seems feasible.
The SAE shows a huge variability across the geographical sectors (Fig. 4). On average, the
highest median SAE, around 1.8 – 2.0, are observed at all central and eastern European
observatories (cf. Table S5). These values are quite high indicating clearly the predominance of
fine particles at these two geographical locations. Moreover, high $PM_{2.5}/PM_{10}$ ratios, indicative of
presence of small particles, are typical for rural lowland sites in central Europe (e.g. Spindler et al.,
2010; EMEP, 2008). Figure S3 also shows that at central and eastern sites the SAE data have
very similar unimodal delta-like distributions. Exceptions are CMN, JFJ and BEO mountain sites,





where left-tailed distributions of SAE are observed likely due to the reduced effect of fine particles
from the PBL in winter and an increase in the relative importance of coarse mineral dust or sea salt
particles as well as aged aerosols compared to lower altitude stations in the same geographical
sector.
On average, the SAE is lower at all other geographical sectors compared to central and eastern
Europe even though some exceptions are observed. For example, at CBW (western Europe) the
median SAE reaches values around 2.1. Indeed, both polluted air masses from industrialized
zones of the Benelux countries and clean air masses from the sea contribute to the presence of
aerosol particles at this site (Crumeyrolle, et al., 2010). Moreover, CBW is surrounded by several
large cities at a distance of about 20 to 40 km from the station, which may have contributed to the
high SAE measured in this geographical location. Asmi et al. (2011) have also shown that
background particle number concentrations at CBW are much higher than for example at BIR.
Median SAE close to one or lower, indicative of the fact that $\sigma_{sp}$ is dominated by large particles, are
observed at more remote sites such as MHD, IZO, ZEP, and TRL. Low SAE at MHD was already
reported by Vaishya et al. (2011, 2012) and justified by the frequent presence mainly in winter of
coarse mode sea-salt particles, since mineral dust particles can be ruled out. In fact, air masses
originating from dust sources are not frequent at these sites. Similarly, the low SAE observed at
ZEP and TRL can be associated with the presence of coarse sea-salt particles. Conversely, the
SAE obtained at IZO is mainly due to the frequent presence of mineral dust particles from African
deserts (e.g. Rodríguez et al., 2011). Very similar bi-modal frequency distributions are observed at
MHD and IZO showing a pronounced left peak indicating the high probability of measuring coarse
particles at these sites. BIR and PLA also show an enhanced left peak in the SAE frequency
distributions.
Differently from $\sigma_{sp}$, the SAE does not show any clear gradient when moving from mountain to
regional/urban sites. For example, at mountain sites the median SAE ranges between around 0.7
at IZO to values higher than two at JFJ and CMN. As reported by Zieger et al. (2012) a SAE value
around 2 prevails for most of the time at JFJ and can be regarded as the typical background under
non-dusty conditions. Thus, the SAE values at JFJ and CMN can be considered as representative
of central Europe free troposphere and especially in winter when the PBL emissions at these sites
are reduced. This high variability of SAE at mountain sites was also reported by Andrews et al.
(2011). Andrews et al. (2011) reported SAE values from 11 mountaintop stations worldwide
ranging from less than one to more than two. Moreover, Bourcier et al. (2012) have shown that at
mountain sites coarse particles are transported more efficiently at high altitude by higher wind
speed thus probably also contributing to the observed variability of SAE at mountain sites. Also at
coastal sites (PLA and MHD), the SAE shows large variability with higher SAE measured at PLA
compared to MHD confirming a higher effect of anthropogenic emissions at PLA compared to
MHD. Less variability in median SAE is on average observed at regional sites, with the exception
of OPE where a lower SAE is observed probably due to the influence of agricultural practices in the
vicinity. Among the urban sites, MAD registers the lowest median SAE (1.47) compared to UGR



(1.69) and DEM (1.60). The lower SAE at MAD could be explained, as already noted, by the
reduced formation of secondary inorganic aerosols during the available measurement period.
Moreover, resuspended dust from vehicles could also explain the lower SAE observed at MAD
observatory.
**3.3 Variability of *g***
The asymmetry parameter is widely used in radiative transfer models because it provides
information about how much radiation is scattered back compared to the amount of radiation
scattered in the forward direction. Figure 5 shows the box-and-whiskers plots of *g* calculated at the
different stations. Table S6 and Figure S4 in the Supporting Material report the statistics of *g* and
frequency and cumulative frequency distributions, respectively. Given that *g* is calculated from BF
using Equation 2 (Section 2.2.3), we report in Figure S5 in the Supporting material the box-and-
whiskers plots of BF whereas Table S7 reports the statistics of BF. Figure 5 and Figure S5 are
symmetrical, thus the lower BF the higher is *g*. As already observed for SAE, the *g* varies
considerably among the different stations ranging between median values around 0.49 (CMN) to
around 0.7 (TRL). Higher *g* median values are in some cases observed at mountain sites
compared to regional or urban environments. This is the case for example for IZO compared to
MSY, UGR and MAD in the southwestern European sector or HPB and JFJ compared to IPR, MPZ
and KOS in central Europe. However, exceptions are observed for example for CMN where the
median *g* value (only 2 years available) is the lowest in the central European sector and among the
lowest observed in this study. On average, *g* values range between 0.49 to 0.64 at mountain sites
with a mean value of 0.58±0.05. This value is consistent with the mean value of 0.61±0.05 reported
by Andrews et al. (2011) at the mountain sites included in their work. Figure S6 in the Supporting
material reports the mean SAE (ordered from low to high values in each geographical location) and
*g* at each station used in this work and the SAE-*g* scatter plot. Figure S6 shows that no clear
relationship between *g* and SAE can be observed. For example, TRL and MHD observatories
register among the highest *g* observed in the network which is consistent with the very low SAE
measured at these stations because of the frequent presence of coarse mode sea-salt particles (cf.
Fig. 4). However, *g* values similar to TRL and MHD are also observed at stations such as PLA,
BIR, JFJ, and DEM, which are dominated on average by fine aerosol particles (SAE similar or
higher than 1.5). However, there are geographical locations (e.g. Nordic and Baltic, western and
southwestern Europe) where SAE increases and correspondingly the *g* decreases from one station
to another indicating a shift toward finer particles. However, this is not a general rule. In fact, the
same relationship is not observed for example in central or eastern Europe (cf. Fig. S6).
Differences in the shape of the particle number size distribution, particle shape and chemical
composition (e.g. refractive index, RI) are factors likely contributing to explain the poor relationship
observed between *g* and SAE. The Mie theory of polydisperse spherical particles predicts that BF
is lower and *g* correspondingly higher for coarse mode aerosol particles (for which the SAE will be
low) compared to fine mode particles. However, some studies deploying integrating nephelometer





have found that BF can be higher for coarse mode aerosol particles (such as mineral dust) than for
fine mode aerosol particles (Carrico et al., 2003; Doherty et al., 2005). Doherty et al. (2005)
suggested that an under-correction for the $\sigma_{sp}$ truncation of the forward-scattered radiation (which
is relatively larger for coarse particles) could bias the calculated BF high. Moreover, the shape of
particle number size distribution is another factor affecting BF and SAE. Thus, differences in the
relative fractions of the fine and coarse modes could also drive the BF-SAE relationship. In fact,
the SAE is most sensitive to the presence of coarse mode aerosol particles compared to BF which
is most sensitive to small accumulation mode particles (Delene and Ogren, 2002; Collaud Coen et
al., 2007). Thus, depending on the shape of the particle number size distribution, BF and SAE
might or might not correlate. Moreover, the refractive index (RI), which is strongly related to the
chemical composition of the particles, is another important variable, that can affect $g$ (e.g. Marshall
et al., 1995). In the work from Hansen and Travis (1974; Fig. 12) the authors showed that for a
given particle diameter the $g$ parameter did non linearly decreased with increasing real RI. Thus,
coarse mode particles with a given RI could have an asymmetry parameter similar or lower to that
of fine particles with lower RI. Recently, Obiso et al. (2017) confirmed the findings by Hansen and
Travis (1974) showing also that a perturbation in RI of 20% has a higher effect on $g$ compared to
similar relative perturbation of particle shape. On the other side, Obiso et al. (2017) showed that a
variation of RI for coarse particles can have a small effect on the mass scattering efficiency of the
particle and its spectral dependence and consequently on SAE.
**3.4 Relationships between $\sigma_{sp}$ and intensive optical properties**
Figure 6 shows the relationships between $\sigma_{sp}$ and SAE and between $\sigma_{sp}$ and $g$ at each station.
Mean SAE and $g$ are calculated for each $\sigma_{sp}$ bin and the bin size at each station is calculated
following the Freedman – Diaconis rule:
Bin size $= 2\frac{IQR(x)}{\sqrt[3]{n}}$ (Eq. 3)
where IQR(x) is the interquartile range of the data and $n$ is the number of observations in the
sample $x$. This kind of graphs helps in understanding which aerosol type on average dominates the
particle light scattering, depending on the amount of scattering measured. It should be noted that in
Figure 6 the number of samples available at each station is not evenly distributed among the
considered bins. Figure S7 in the Supplementary Material shows for some stations the SAE-$\sigma_{sp}$
pairs colored by the number of samples in each bin to highlight how samples are distributed among
the bins.
**3.4.1 $g$-$\sigma_{sp}$ relationships**
The asymmetry parameter $g$ shows the lowest values under very low $\sigma_{sp}$ suggesting the
predominance of small fine mode particles. Andrews et al. (2011) reported similar $g$-$\sigma_{sp}$





relationships at different mountain sites and suggested that the removal of large particles by cloud
scavenging or by deposition during transport could explain the observed low *g* under a clean
atmosphere. They also suggested that the formation of new particles followed by
condensation/coagulation could generate small but optically active particles. Here, we show that
this behavior of BF or *g* as a function of $\sigma_{sp}$ was observed at all sites, not only at mountain sites.
The parameter *g* then increases with increasing $\sigma_{sp}$ indicating a shift of the particle number size
distribution towards the larger end of the accumulation mode. Delene and Ogren (2002), Andrews
et al. (2011) and Pandolfi et al. (2014) showed that BF tends to decrease with increasing aerosol
loading, consistent with the observed increase of *g*. For comparison with previous works, Figure S8
in the Supplementary Material shows the BF-$\sigma_{sp}$ relationships for all observatories evidencing the
aforementioned BF decrease with increasing $\sigma_{sp}$.
The shift of the particle number size distribution towards the large end of the fine mode with
increasing $\sigma_{sp}$ is probably the main reason causing the observed increase of *g* (and the decrease
of BF, cf. Fig. S8). A possible explanation for this shift could be a progressive aging of atmospheric
aerosol particles. Then, at the majority of stations, the variation of *g* is less pronounced under
periods of high particle mass concentrations suggesting changes mostly in the coarse aerosol
particles mode rather than in the fine mode.
**3.4.1 SAE-$\sigma_{sp}$ relationships**
As reported in Figure 6, at some stations the SAE progressively increases with $\sigma_{sp}$ in the $\sigma_{sp}$ range
where the *g* parameter increases as well. The increase of both *g* and SAE with $\sigma_{sp}$, observed for
example at the Nordic and Baltic, central and eastern European observatories, could be related to
the different effects that different particle sizes have on SAE and *g*. A progressive increase of SAE
with $\sigma_{sp}$ would suggest an increasing relative importance of fine aerosol particles. The origin of
these fine particles is probably different depending on the location of the measuring site. For the
remote PAL site, for example, Lihavainen et al. (2015b) observed an increase of both $\sigma_{sp}$ and SAE
with increasing temperature due to increasing formation of BSOA (biogenic secondary organic
aerosols) with increasing ambient temperatures, thus likely driving the $\sigma_{sp}$-SAE relationships
reported in Fig. 6 for PAL. The BSOA from gas-to-particle formation over regions substantially
lacking in anthropogenic aerosol sources such as the European boreal region (Tunved et al., 2006)
are probably strongly contributing to the $\sigma_{sp}$-SAE relationships observed at other Nordic and Baltic
sites such as SMR. At polluted sites such as those located in central and eastern Europe the
anthropogenic aerosol emission and the active secondary aerosol production in the region (e.g. Ma
et al., 2014) are probably driven the $\sigma_{sp}$-SAE relationships reported in Fig. 6. For higher $\sigma_{sp}$, the
$\sigma_{sp}$-SAE relationships changed and a progressive shift toward relatively larger particles is on
average observed with increasing $\sigma_{sp}$. However, at the majority of northwestern, central and
eastern European stations, the SAE keeps values around or higher 1.5 under high particle load
indicating that high $\sigma_{sp}$ is dominated by fine particles. An exception is MHD where SAE increases





with increasing $\sigma_{sp}$ keeping values on average lower than 1.4 under high particle load (cf. Fig. 6).
As already observed, low SAE at MHD is mainly due to the predominance of sea-salt coarse
particles at this site (Vaishya et al., 2011). Conversely, at some sites in South Europe (e.g. MSA,
MSY, IZO, DEM) the SAE reaches values around one or lower under high particle load indicating
that at these stations high $\sigma_{sp}$ is dominated by mineral dust coarse particles mainly from African
deserts. Exceptions are two urban sites in Southwestern Europe (UGR and MAD) where fine
particle likely mostly from traffic (and also from biomass burning at UGR) on average dominate the
highest measured $\sigma_{sp}$. Similar $\sigma_{sp}$-SAE relationships, as those reported in Fig. 6, were observed by
Andrews et al. (2011) at mountain sites and by Delene and Ogren (2002) at marine sites. Among
the lowest SAE are observed at IZO, the station closest to the African continent. Interestingly, at
IZO the SAE shows the highest gradient for $\sigma_{sp}$ coefficients in the range of 0-50 Mm$^{-1}$ whereas the
gradient is much lower for $\sigma_{sp}$ higher than 50 Mm$^{-1}$ being the SAE almost constant for $\sigma_{sp}$ higher
than 100 Mm$^{-1}$. IZO station is often in the free troposphere and high loading at this station are only
registered under Saharan dust events, thus almost only mineral dust is measured at IZO. Normally
the long-rang transport mineral dust particle don't have a significant fraction above 10 µm because
of the short lifetime, thus likely explaining the constant SAE observed at IZO under high aerosol
loading.
**3.5 Seasonal variability**
Figures 7, 8 and 9 present the annual cycles of $\sigma_{sp}$, SAE and $g$, respectively, at each site. Overall,
strong seasonal cycles of $\sigma_{sp}$ and intensive aerosol particle optical parameters are observed at the
majority of the stations even if exceptions are observed. Given the important role that the station
placement plays in the seasonal cycles of aerosol parameters, the analysis is presented below
separately for mountain observatories ad for low altitude observatories.
**3.5.1 Seasonal variability at mountain observatories**
At the mountain stations (PUY, HPB, JFJ, CMN, BEO, MSA, and IZO), $\sigma_{sp}$ peaks in spring/summer
whereas lower $\sigma_{sp}$ values are measured in autumn/winter. Similar findings were for example
already reported by Nyeki et al. (1998) for JFJ and summarized by Andrews et al. (2011) for many
mountain top stations worldwide and by Pandolfi et al. (2014) for MSA station. Different factors
contribute to the $\sigma_{sp}$ increase in spring/summer at the mountaintop observatories, such as the
increase of the boundary layer height and stronger upslope winds during the warmest months.
Moreover, specific events such as Saharan mineral dust outbreaks, may contribute to the
increased $\sigma_{sp}$ observed at mountain stations in spring/summer, and especially in southern Europe
(e.g. Pey et al., 2013; Pandolfi et al., 2014; Rodríguez et al., 2011). At IZO, $\sigma_{sp}$ peaks strongly in
July-August because of the very high influence of African mineral dust at this station during these




months (e.g. Alastuey et al., 2005; Diaz et al., 2006). At the mountaintop CHC observatory, $\sigma_{sp}$
progressively increases during the dry season, from May to October, reaching lower values during
the rainy season (from December to April). Moreover, during the dry season the new particle
formation events, taking place at CHC with one of the highest frequency reported in the literature
so far (Rose et al. 2015), can introduce very small particles that grow to the nucleation and Aitken
mode. At the mountain stations, both SAE and $\sigma_{sp}$ are on average higher in summer compared to
the winter period, thus suggesting a higher anthropogenic influence at these sites during the
warmest months. The summer SAE increase is more evident at some mountain stations, e.*g.* HPB,
CMN, and BEO, compared to other mountain stations such as JFJ and MSA. Less pronounced
SAE seasonal variation at JFJ was related by Bukowiecki et al. (2016) to the rather constant
composition of the JFJ aerosol. At the southern station of MSA the observed less pronounced
seasonal cycle of SAE could be related with the Saharan dust outbreaks which contrast the PBL
transport of fine particles observed at other mountain sites. At IZO, the SAE reaches the lowest
values during July-August being the Saharan dust outbreaks very intense at this site during this
period.

Overall, the *g* parameter shows opposite seasonal cycles compared to SAE at almost all mountain
stations with the exception of JFJ and BEO where *g* slightly increases with SAE in summer. At
almost all mountain stations, the seasonal variations of SAE and *g* are less pronounced compared
to the seasonal variation of $\sigma_{sp}$ indicating larger seasonal variation in the extensive aerosol optical
properties than in the intensive properties. For example, the median $\sigma_{sp}$ values at MSA increase by
around 800% during summer (JJA) compared to winter (DJF), whereas SAE and *g* increase by
around 5-7%. Similar relative increases are observed at JFJ (660%, 16% and 11% for $\sigma_{sp}$, SAE
and *g*, respectively) whereas the relative increases are much higher at BEO, especially for $\sigma_{sp}$
(around 1300%) and SAE (26%). At CMN, the median $\sigma_{sp}$ value increases by around 400% from
winter to summer, whereas SAE and *g* increase and decrease, respectively, by around 46% and
6%, respectively. At CHC, the SAE decreases as the $\sigma_{sp}$ increases moving from wet to dry season,
indicating an increasing effect of coarse particles on $\sigma_{sp}$ during the dry season. At PUY, $\sigma_{sp}$ peaks
from March to September and this increase is accompanied by a small SAE increase. Venzac et
al. (2009) and Boulon et al. (2011) have shown that PUY is more often influenced by the free
troposphere or residual layers in winter and spring compared to the summer season.

**3.5.1 Seasonal variability at low altitude observatories**

At some of the low altitude observatories, the seasonal variation of particle scattering is opposite
compared to the variations observed at mountain sites, $\sigma_{sp}$ being higher in winter and lower in
summer. MHD, CBW and SIR in the western sector, IPR, MPZ and KOS in central, KPS in eastern
and UGR in south-western Europe show such increase in particle mass concentration in
wintertime. The reasons causing these marked seasonal cycles are probably different depending
on the geographical sector considered.



### 3.5.1.1 Central and eastern Europe

Central and eastern European observatories show marked seasonal cycles of both extensive and intensive aerosol particles optical properties. In these regions, less horizontal and vertical pollutant dispersion in winter, due to a higher frequency of stagnant conditions and temperature inversions, play an important role in accumulating aerosols. As a consequence, as reported in Figure 7, the $\sigma_{sp}$ is much higher in winter compared to summer. SAE and $g$ also show marked season cycles in these regions, being the SAE ($g$) higher (lower) in summer compared to winter (cf. Fig. 8). Ma et al. (2014) have shown that at MPZ an increased SAE in summer is mainly explained by the variation of the particle number size distribution. Thus, high concentrations in spring and summer of small particles during new particle formation and subsequent growth cause the observed increase of SAE during warmest months.

### 3.5.1.2 Nordic and Baltic regions

At the Nordic and Baltic sites, the monthly variation of $\sigma_{sp}$ is on average less pronounced compared to the central or eastern European stations and especially at BIR, SMR and PAL. This is likely due to the placement of these stations located in remote areas with different meteorology (e.g. less pronounced PBL variations) and where on average much lower $\sigma_{sp}$ values are measured compared to other European sites. Moreover, this could also indicate the importance of anthropogenic sources like domestic heating in central and eastern Europe in winter. However, the monthly variation of SAE and $g$ is rather pronounced at these Nordic and Baltic observatories: SAE ($g$) increases (decreased) in summer compared to winter indicating the predominance of relatively smaller particles during the warmest months. Similar findings were reported for the SMR and PAL observatories by Virkkula et al. (2011) and Lihavainen et al. (2015a), respectively. The observed seasonal variations in intensive aerosol optical properties were related to both the transport of different air masses at these remote sites depending on the season and the enhanced formation of BSOA in summer (e.g. Lihavainen et al., 2015a). Lihavainen et al. (2015a) and Virkkula et al. (2011) also reported a lower single scattering albedo in winter compared to summer at PAL and SMR, respectively, frequently dropping below 0.7 at SMR due to a significant contribution from light absorbing carbon, mostly from residential wood combustion. Thus, they have shown that aerosol particles observed in summer at SMR and PAL had the potential to cool the atmosphere more efficiently than those observed during winter. Similar intensive optical properties season cycles were observed at BIR.

### 3.5.1.3 Western Europe

Similarly to the Nordic and Baltic regions, differences in aerosol sources and sinks are the likely reasons explaining the seasonal variation of $\sigma_{sp}$, SAE and $g$ observed in western Europe. Marked $\sigma_{sp}$ seasonal cycles are observed at all low altitude western European observatories, with higher values measured in winter compared to summer. On average, at these sites, SAE ($g$) is higher (lower) in summer compared to winter. O'Connor et al. (2008) and Vaishya et al. (2011, 2012)





showed that the background marine aerosol measured at MHD contains a strong and significant
seasonal cycle with sea-salt dominating in winter and biogenic organic aerosol dominating the
submicron sizes in summer. This is consistent with the observed season cycles of SAE and *g*
reported here for MHD.
**3.5.1.4 South Europe**
Among the southern European observatories, marked seasonal variation for $\sigma_{sp}$ is observed
especially at UGR, MSY and FKL. At the urban UGR site, the mean aerosol type is very different in
winter compared to summer. As evidenced by the seasonal cycles of SAE and g, aerosol particles
are generally finer in winter at UGR compared to the summer season as already observed for
example by Lyamani et al. (2010; 2012) and Titos et al. (2012). This is likely due to the
accumulation of fine particles, mainly from traffic, domestic heating and biomass burning, favored
by stagnant conditions and atmospheric inversions during winter. In summer, the higher frequency
of Saharan mineral dust outbreaks at this site increases the mean size of the particles during the
warmest months. At the MSY regional site, the higher efficiency of the sea breeze in transporting
pollutants from the urbanized/industrialized coastline toward regional inland areas during the
warmer season mainly explains the summer increase in aerosol particle mass concentration
observed at this site (e.g. Pandolfi et al., 2011). Moreover, the enhanced formation of secondary
sulfate and organic matter in summer together with frequent Saharan mineral dust outbreaks,
strongly contribute to the observed seasonal cycle for $\sigma_{sp}$ and intensive properties at MSY site. The
$\sigma_{sp}$ peak observed at MSY in March is due to the winter pollution episodes typical of the western
Mediterranean basin (WMB) (e.g. Pandolfi et al., 2014a and references therein). During these
episodes, the accumulation of pollutants close to the emission sources is favored by anticyclonic
conditions coupled with strong atmospheric inversions. During such conditions, pollutants
accumulate in the PBL and can subsequently reach the station when PBL height increases. On
average, at MSY low SAE are measured in April and October likely due to the occurrence of
Saharan dust outbreaks during these months. At FKL no intensive optical aerosol properties are
available. The high $\sigma_{sp}$ in summer at this site is also associated with mineral dust storm events as
for example reported by Vrekoussis et al. (2005). However, mineral dust storms in the
Mediterranean are not the only reason for the observed increased $\sigma_{sp}$ in summer. In fact, as for
example reported by Kalivitis et al. (2011) for FKL and Pandolfi et al. (2011) for MSY, ammonium
sulfate and particulate organic matter, whose concentrations increase in summer in the
Mediterranean Basin, were assumed as important contributors to $\sigma_{sp}$ during the warm season. At
the DEM urban observatories, the high $\sigma_{sp}$ measured in spring are linked to Saharan dust
outbreaks as also supported by the seasonal cycles of SAE and *g* which showed the lowest and
highest, respectively, values in spring.



## 3.6 Trends

Trends of $\sigma_{sp}$, SAE and BF are studied for those stations having more than 8 years of data (13 observatories). Generally, it is recommended to have more than 10 years of data for trend studies. Among the ACTRIS stations, PAL, SMR, MHD, HPB, IPR, JFJ, and UGR have more than 10 yr of data, whereas at PUY, MPZ, CMN, BEO, KPS, and IZO, 8 or 9 years are available. These stations are included in order to improve the spatial coverage, similarly as in Collaud Coen et al. (2013). The Theil Sen statistical estimator (Theil, 1950; Sen, 1968) is used here to determine the regression parameters of the data trends, including slope, uncertainty in the slope and p-value. The Theil Sen method provides similar results as the Mann-Kendall test and it is implemented for example in the Openair Package available for R space (Carslaw, 2012; Carslaw and Ropkins, 2012). The applied method yields accurate confidence intervals even with non-normal data and it is less sensitive to outliers and missing values (Hollander and Wolfe, 1999). Monthly means are used for trend analysis and the data are deseasonalized. The data coverage of $\sigma_{sp}$ is higher than 70% at all stations included in trend analyses with the exception of IZO where the $\sigma_{sp}$ data coverage is 55%. For SAE, the data coverage is higher than 65% at all sites with the exception of PAL (54%), PUY (59%), and IZO (52%). For BF, the data coverage is higher than 65% with the exception of PAL (26%), PUY (43%), BEO (47%) and IZO (27%). At the remote (PAL) or mountain stations (PUY, BEO, and IZO), the percentage for the intensive aerosol particle optical properties is lower because of a higher probability of measuring $\sigma_{sp}$ lower than the threshold (0.8 Mm$^{-1}$) selected for the calculation of SAE and BF. Table 2 reports the trends observed for $\sigma_{sp}$, SAE and BF at the thirteen observatories included in this analysis. Magnitude and statistical significance of the trends for these parameters are reported in Table S8 in the Supporting Material. It should be noted that changes in particle size cut-off reported for PAL and KPS (cf. Table 1) may have affected the reported trend analyses at these stations, but estimating the impact of these changes in the observed trend is not simple. However, as already noted, Lihavainen et al. (2015a) reported that at PAL the inlet changes had minor effects on scattering, because the number concentration of coarse particles is very low at this observatory. KPS is dominated by very fine particles and the change from PM$_1$ to PM$_{10}$ had probably a minor effect on $\sigma_{sp}$, SAE and BF. Moreover, at KPS the inlet was changed in April 2008, less 1.5 years after the beginning of the measurements thus likely having a minor effect in the trend analysis performed at this site over the period 2006 – 2014. The FKL observatory was removed from trend analysis because the inlet was changed from whole air to PM$_{10}$ in 2009, from PM$_{10}$ to PM$_1$ in 2011, and again from PM$_1$ to PM$_{10}$ in 2013 (cf. Table 1), thus likely having a major effect on the measured particle optical properties.

In Table 2, a comparison with previous trends analysis results presented by Collaud Coen et al. (2013) for aerosol particle optical properties and by Asmi et al. (2013) for particle number concentrations is also reported.


### 3.6.1 Trends of $\sigma_{sp}$

Overall, $\sigma_{sp}$ decreases at the majority of the stations included in this work. Significantly decreasing trends for $\sigma_{sp}$ are observed at: the two Nordic and Baltic observatories (PAL for the period 2000 – 2010 and SMR); at two observatories (HPB and IPR) out of five observatories in central Europe; and at the two observatories in southwestern Europe (IZO and UGR). The trends are not statistically significant in western (MHD and PUY) and eastern (BEO and KPS) Europe. The highest magnitude of $\sigma_{sp}$ trend [Mm$^{-1}$/yr] (cf. Table S8 in the Supplementary Material) is observed at the polluted IPR observatory. Conversely, the lowest magnitude is observed at the remote PAL observatory. For the periods considered in this work, the total reductions (TR) for $\sigma_{sp}$ range between around 30% (SMR) and 60% (IZO). The high TR observed at IZO might be affected by the intensity and frequency of Saharan dust outbreaks at this site. However, estimating the effects of these events at IZO is beyond the scope of this study. Overall, the observed decreasing trends of $\sigma_{sp}$ are consistent with the uniform decrease in aerosol optical depth observed in Europe (AERONET data in Li et al., 2014). A statistically significant decreasing trend of $\sigma_{sp}$ at IPR was also reported by Putaud et al. (2014) for the period 2002 – 2010. As reported in Table 2 statistically significant decreasing trend for $\sigma_{sp}$ is observed at around 50% of the stations considered here. Overall, the observed statistically significant decreasing trends of $\sigma_{sp}$ are consistent with the demonstrated reduction of PM concentration in the atmosphere in Europe in these last decades thanks to the implementation of European/national/regional/local mitigation strategies. These decreasing trends are also consistent with trends of aerosol chemistry derived from observations in urban environments in Europe (e.g. EEA, 2013; Barmpadimos et al., 2011; Titos et al., 2014; Pandolfi et al., 2016), regional and remote environments in western Mediterranean (Cusack et al., 2012; Pandolfi et al., 2016) and in general with derivation of trends for aerosol chemistry across Europe (Tørseth et al., 2012). Recently, Collaud Coen et al. (2013) showed that trends in $\sigma_{sp}$ are observed at most of the US continental sites and that these trends are generally consistent with the strong $SO_2$ and PM reductions observed in the US (Asmi et al., 2013; EPA, 2011). Conversely, in Europe the strong decreasing trend observed for $SO_2$ (e.g. Tørseth et al., 2012; Henschel et al., 2013) and, with a lower spatial homogeneity and statistical significance, for $PM_{2.5}$ (e.g. EEA, 2016) is not observed for aerosol optical properties. As reported in Collaud Coen et al. (2013) the reasons why at some of the European sites no significant trends are observed, might be related to the spatial inhomogeneities and under-representation of continental Europe PBL sites (e.g. Laj et al., 2009) and/or the timing of the $SO_2$ and PM trends for the US and Europe. In Europe the emissions reductions were greater for the period 1980–2000 compared to the period 2000 – 2010 (e.g. Colette et al., 2016; Tørseth et al., 2012; Manktelow et al., 2007), thus the measurements of optical particle properties in Europe may not go back far enough to reflect the time period with the largest emission reductions. Tørseth et al. (2012) reported average reductions for ambient sulfate and nitrate mass concentrations in Europe of -12% and -1%, respectively, during 2000 – 2009 compared to -24% and -7%, respectively, during 1990 – 2000. They also reported statistically




significant decreases of $PM_{10}$ and $PM_{2.5}$ mass concentrations at around 50% of European sites
with total reductions of -18% and    -27%, for $PM_{10}$ (24 sites) and $PM_{2.5}$ (13 sites), respectively,
during 2000 – 2009. A direct comparison between the stations included in this work and those
included in Tørseth et al. (2012) is not possible because of the different timing of reported $\sigma_{sp}$ and
PM mass concentration measurements. At those stations where a significant decreasing trend for
$\sigma_{sp}$ is observed and considering a period of 10 yr (even if not coincident for all stations), the total
reduction for $\sigma_{sp}$ in Europe is around -35% (cf. Table S8) consistent with the trend reported by
Tørseth et al. (2012) for PM in Europe. Quite good agreement, even though again likely biased by
the different timings, is also observed comparing PM mass concentration and $\sigma_{sp}$ trends by
geographical sectors. A significant total reduction around -40 ÷ -30% was reported for $PM_{10}$ and
$PM_{2.5}$ in the Nordic and Baltic sector by Tørseth et al. (2012; cf. Fig. 7 in Tørseth et al. (2012)) in
close agreement with the statistically significant total decrease of $\sigma_{sp}$ around -34% reported for PAL
during 2000 – 2010 (cf. Table S8). In the Western sector (MHD) the decreasing trend for $PM_{2.5}$
during 2000 – 2009 was insignificant (-10 ÷ 0%) as reported here for $\sigma_{sp}$ during the period 2001 –
2010. In the Central sector statistically significant decreases for $PM_{2.5}$ and $PM_{10}$ mass
concentrations ranged between -20% and -40% during a 10 yr period (2000 – 2009) and the total
reduction for $\sigma_{sp}$ ranged between -38% (HPB) and around -48% (IPR). In the Southwestern
European sector the total reduction for $\sigma_{sp}$ is around -32% (at UGR) and -60% (at IZO), whereas
Tørseth et al. (2012) reported around -20 ÷ -40% decrease for the PM10 mass concentration. To
further confirm the observed close agreement between PM trends reported in literature and the
trends of $\sigma_{sp}$ in this work, Table S9 in the Supporting Material reports the comparison between $\sigma_{sp}$
and $PM_{10}$ and/or $PM_{2.5}$ mass concentration trends calculated at those stations where simultaneous
$\sigma_{sp}$ and PM mass concentration measurements are available. As reported in Table S9 both the
observed total reductions and the statistical significance of the trends are very similar for $\sigma_{sp}$ and
$PM_{10}$.
**3.6.2 Trends of SAE and BF**
The trends for SAE are estimated for three different quantities, namely: the SAE calculated as
linear fit using three wavelengths (b-g-r), using the blue and the green wavelengths (b-g) and using
the green and red wavelengths (g-r). For the periods considered in this work (in bold in Table 2),
the SAE calculated using the three wavelengths (b-g-r) shows statistically significant trends at five
sites. At PAL (Nordic and Baltic), PUY (western Europe) and BEO (eastern Europe) decreasing
trends are observed, whereas increasing trends are observed at HPB (central Europe) and UGR
(southwestern Europe). Uniform negative trends of columnar Ångström exponent from AERONET
data were reported by Li et al. (2014) across Europe and these trends were ascribed to reduced
fine-mode anthropogenic emission. The positive SAE trend observed at HPB and UGR would
suggest a shift of the accumulation mode particles towards smaller sizes and/or a change in the



coarse aerosol mode. For example, the SAE increase at UGR could be probably explained by a
progressive relative importance of fine particles emissions driven by a progressive reduction of
coarse particles for example from construction/demolition works due to the economic crisis which
affected Spain from 2008 (e.g. Lyamani et al., 2011; Querol et al., 2014; Pandolfi et al., 2016). In
fact, Titos et al. (2014) reported statistically significant decreasing trend for $PM_{10}$ fraction during the
period 2006 – 2010 whereas no trend was observed for $PM_1$ fraction. Moreover, at UGR,
statistically significant increasing trend is also observed for the SAE calculated using the green and
red wavelengths (g-r), likely more sensitive to the coarser particle mode, whereas the trend was
non-statistically significant for the SAE b-g. The possible change in the coarse aerosol mode at
UGR is likely also causing the observed statistically significant increasing trend of BF (cf. Table 2),
given that a positive trend of BF would be consistent with a shift of the accumulation mode
particles towards smaller sizes. Similarly, statistically significant increasing trends for both SAE and
BF are also observed at SMR (SAE b-g) and HPB. Statistically significant increasing trends of BF
are also observed at the other Nordic and Baltic stations (PAL) and at PUY (western Europe),
where the SAE shows statistically significant decreasing trends, and at IPR (central Europe) where
the trend of SAE was insignificant. Thus, overall, the trends of BF are positive at all stations where
BF measurements are available. The opposite sign of the trends of SAE and BF at PAL and PUY
could be due to different effects that different particle sizes have on SAE and $g$ or a progressive
change in the mean diameter of the fine mode aerosol. Further research involving for example size
distribution data and Mie calculation could help in understanding the differences observed in some
cases between SAE and BF (or $g$). Recently, Korras-Carraca et al. (2015) have shown that the
column integrated $g$ from Modis-Terra had widely statistically significant positive trends (2002-
2010) with stronger increases observed in the eastern and southern Black Sea, as well as over the
Baltic and Barents seas. Moreover, both Modis-Terra and Modis-Aqua produce positive trends of $g$
in the eastern Mediterranean Sea and the eastern coast of the Iberian Peninsula. Positive trends
for $g$ would correspond to negative trends for BF. The difference observed with our work could be
due to the different variability often observed between near-surface measurements and column
integrated measurements which can confound the relationship between surface and column optical
properties (e.g. Bergin et al., 2000; Lyamani et al., 2010). Although, it was shown that mid altitude
station might be globally representative of the whole atmopheric column (Chauvigne et al., 2016).

### 3.6.3 Comparison with previous trend analyses

Table 2 shows the comparison, over the same periods, between the trend analyses performed in
this work and the analyses presented by Collaud Coen et al. (2013) for aerosol particle optical
properties and by Asmi et al. (2013) for particle number concentrations ($N_{LDL-500}$, $N_{20-500}$ and $N_{100-}$
$_{500}$). An agreement with the results from Collaud Coen et al. (2013) is observed for JFJ where
consistent insignificant trends are detected for the three periods reported in Collaud Coen et al.
(2013). For MHD, we calculated a non-significant increasing trend for $\sigma_{sp}$ during 2001 – 2010,



whereas Collaud Coen et al. (2013) reported a statistically significant increasing trend for the same
period. At PAL, non-statistically significant trend for $\sigma_{sp}$ is observed here and in Collaud-Coen et al.
(2013) for the period 2001 – 2010, whereas we observe a statistically significant decreasing trend
for the period 2000 – 2010. Moreover, at PAL, we observe statistically significant decreasing trend
for SAE during the two common periods which were insignificant in Collaud Coen et al. (2013). It
should be noted that Collaud Coen et al. (2013) reported insignificant SAE trend at PAL using the
Mann Kendall test whereas they reported statistically significant decreasing trends using the
GLS/ARB and LMS methods, consistent with our work. These differences are thus likely due to the
relative short period used in these trend analyses and the different sensitivity of the methods used
to the presence of missing values or outliers especially at PAL where $\sigma_{sp}$ is very low (cf. Fig. 2). For
example, in this work the SAE calculated for PAL during the year 2007 was removed from the
trend analysis due to the presence of too many extreme high SAE values, thus also likely
explaining the difference observed for SAE with the work from Collaud Coen et al. (2013).
Moreover, here we use de-seasonalized monthly means for trend analyses whereas Collaud-Coen
et al. (2013) used de-seasonalized medians with different time granularity (3 days) thus likely
affecting the comparison, especially over relatively short periods.
A comparison of trends analysis results between $\sigma_{sp}$ and the particle number concentration is not
straightforward as the $\sigma_{sp}$ measurements are more sensitive to the particle number concentration in
the upper end of the fine mode than to smaller particles. For example, Asmi et al. (2013) reported
that, globally, no strong similarities were observed between $\sigma_{sp}$ and N trends and that the N trends
are controlled by particles in the larger range of the Aitken mode and smaller range of the
accumulation mode, e.g. ca. 50–150 nm diameter. In this work, as reported in Table 2, the
statistically significant decreasing trend reported for N during the period 2001 – 2010 is not
observed for $\sigma_{sp}$. However, differences are also observed at PAL between N20 and N100 mainly
because DMPS measurements at PAL had long gaps during periods with unusually low
concentrations thus effectively removing low concentrations from the trend analysis (Asmi et al.,

27  2013).

**3.6.4 Daytime and nighttime trend analyses at mountain sites**
Finally, the analysis of the trends during daytime (08:00 – 16:00 GMT) and nighttime (21:00 –
05:00 GMT) by season at mountain stations are also analyzed (Table 3). This analysis could
provide information about changes in $\sigma_{sp}$ when the mountain stations are likely affected by the PBL
(e.g. daytime and/or summer) or by the residual layer (e.g. nighttime in summer) or when these are
representative of the free troposphere (e.g. nighttime in winter). Consistently with what reported in
Table 2 for $\sigma_{sp}$, the trends are insignificant at JFJ, PUY CMN, and BEO irrespective of the time of
the day or season. The decreasing trends observed at HPB, also reported in Table 2, are





statistically significant only during autumn, irrespective of the time of the day. Conversely, the trend
observed for $\sigma_{sp}$ at IZO reported in Table 2, is not observed by splitting the analysis by time of the
day and/or season.
**Conclusions**
This investigation presented the near-surface in-situ $\sigma_{sp}$ (aerosol particle light scattering), SAE
(Scattering Ångström exponent), BF (backscatter fraction), and *g* (asymmetry parameter)
measurements obtained over the past decade at 28 measuring atmospheric observatories which
are part of the ACTRIS Research Infrastructure and most of them belong to the GAW network.
Results show a large variability of both extensive and intensive aerosol particle optical properties
across the network, which is consistent with the previously reported variability observed for other
aerosol particle properties such as particle mass concentration, particle number concentration and
chemical composition. Main findings can be summarized as follows:
-   Overall, the highest $\sigma_{sp}$ are measured at low altitude observatories in central and eastern

17       Europe and at some urban sites in south Europe whereas, the lowest $\sigma_{sp}$ are observed at some

18       mountain stations and at two Arctic and Antarctic sites. Low $\sigma_{sp}$ levels, comparable with those

19       measured at mountain sites, are also observed at the majority of the regional/continental

20       Nordic and Baltic observatories. The $\sigma_{sp}$ values in Western Europe are on average higher

21       compared to those measured in the Nordic and Baltic regions and lower compared to those

22       measured at regional level in south Europe. Some exceptions to these general features are

23       however observed.

24   -   In central and eastern Europe, independently from the station placement, the SAE *(g)* is among

the highest (lowest) observed across the network indicating a large predominance of fine

particles. In these regions, the SAE *(g)* is even higher (lower) in summer compared to winter

suggesting the shift toward the small end of the aerosol particle size distribution likely linked to

new particle formation events during the warmest months. On average SAE *(g)* is lower

(higher) in the Nordic and Baltic, western and southern sectors compared to central and

eastern Europe.

-   Seasonal cycles for $\sigma_{sp}$ are observed in all geographical sectors. These are especially marked

at regional level in central and eastern Europe where wintertime episodes linked with stable air

and thermal inversions favor the accumulation of pollutants. Clear annual cycles are also

observed at mountain sites where $\sigma_{sp}$ is higher in summer because of the enhanced boundary

layer influence. In some cases, SAE (*g*) is also high (low) in summer at mountain sites

indicating a higher PBL anthropogenic influence during the warmer months. In the Nordic and

Baltic regions, the seasonal variation of $\sigma_{sp}$ is less pronounced compared to central and

eastern Europe likely due to different meteorology and less pronounced PBL variations.

Despite the relatively small $\sigma_{sp}$ seasonal cycles in the Nordic and Baltic regions, SAE (*g*)



increases (decreases) in these regions in summer compared to the winter period likely due to a seasonal-dependent transport of air masses at these remote sites and an enhanced formation of secondary organic aerosols previously observed at these sites during the warmest months. At coastal sites in northwestern Europe, the presence of sea-salt particles in winter also contributes to the observed pronounced seasonal cycles of SAE and *g*.

- The analysis of the systematic variability of SAE and *g* as a function aerosol loading ($\sigma_{sp}$) reveals some common patterns. At all stations, *g* shows the lowest values under very low $\sigma_{sp}$ likely because the formation of new particles in a clean atmosphere followed by condensation/coagulation with consequence generation of small but optically active particles. The *g* then sharply increases with increasing $\sigma_{sp}$ indicating the shift of the particle number size distribution toward the larger and of the accumulation mode. Then, under periods of high particle mass concentrations, the variation of *g* is less pronounced at the majority of the stations contrary to the SAE which increases or decreases suggesting changes mostly in the coarse aerosol particles mode rather than in the fine mode.

- The analyses of the trends reported in this investigation provide evidence that both extensive and intensive aerosol optical properties have significantly changed at some of the locations include here over the last 10 and 15 years. The $\sigma_{sp}$ decreasing trends reported here are statistically significant at 5 out of 13 stations included in the analysis. These 5 stations are located in the Nordic and Baltic, central and southwestern sectors. Conversely, $\sigma_{sp}$ decreasing trends are not statistically significant in western and eastern Europe. Statistically significant decreasing trends of SAE are observed at 3 out of 10 observatories included in the analysis: one site the Nordic and Baltic sector and two mountain sites in the western and eastern sectors. These negative trends could be ascribed to reduced fine mode anthropogenic emission as already observed in literature for columnar SAE in Europe. Conversely, at two stations (one mountain site in central Europe and one urban site in southwestern Europe), the SAE shows statistically significant increasing trend suggesting a shift of the accumulation mode particles towards smaller sizes and/or a change in the coarse aerosol mode. At the remaining 5 observatories the reported SAE trends are not statistically significant. The backscatter fraction shows statistically significant increasing trend at 6 out of 9 sites where BF measurements are available. At three stations (the mountain site in central Europe, the urban site in southwestern Europe and one of the two sites in the Nordic and Baltic sector), both BF and SAE increase suggesting consistent evidence of a shift of the accumulation mode particles towards smaller size. Conversely, at the other site in the Nordic and Baltic sector and at one mountain site in the western sector BF increases whereas SAE decreases.

In conclusion, this investigation provides a clear and useful picture of the spatial and temporal variability of the surface in-situ aerosol particle optical properties in Europe. The results presented here give a comprehensive view of the particle optical properties and provide a reliable analysis of aerosol optical parameters for model constraints. In addition, the analysis presented here suggests



findings that may need additional investigation. For example, the fact that at some of the stations the trend of $\sigma_{sp}$ changes in terms of both statistically significance and sign depending on the period used, suggests that trend analyses are necessary in the future when longer-duration records will be available. Moreover, the fact that at some sites BF and SAE show different sign in the trends suggests that further analysis is needed to better understand how other aerosol parameters such as particle size distribution and mean diameter affect the relationships between BF and SAE.

**Acknowledgments**

This project has received funding from the European Union's Horizon 2020 research and innovation programme under grant agreement No 654109, ACTRIS (project No. 262254), ACTRIS-PPP (project No 739530). We also thank the International Foundation High Altitude Research Stations Jungfraujoch and Gornergrat (HFSJG), which made it possible to carry out the experiments at the High Altitude Research Station at the Jungfraujoch and the support by MeteoSwiss within the Swiss program of the Global Atmosphere Watch (GAW) of the WMO. MAD station is co-financed by the PROACLIM ( CGL2014-52877-R) project. SMR station acknowledges BACCHUS (project No. 603445), CRAICC (project No. 26060) and Academy of Finland (project No. 3073314). UGR station is co-financed by the Spanish Ministry of Economy and Competitiveness through project CGL2016-81092-R. Measurements at Montseny and Montsec stations were supported by the MINECO (Spanish Ministry of Economy and Competitiveness) and FEDER funds under the PRISMA project (CGL2012-39623-C02/00), by the MAGRAMA (Spanish Ministry of Agriculture, Food and Environment) and by the Generalitat de Catalunya (AGAUR 2014 SGR33 and the DGQA). Measurements at Izaña were supported by AEROATLAN project (CGL2015-17 66229-P), co-funded by the Ministry of Economy and Competitiveness of Spain and the European Regional Development Fund. Station Košetice is supported by Ministry of Education, Youth and Sports of the Czech Republic within project for support of national research infrastructure ACTRIS – participation of the Czech Republic (ACTRIS-CZ – LM2015037). Measurements at Puy de Dôme were partly supported by CNRS-INSU, University Clermont-Auvergne, OPGC and the french CLAP program. PAL station acknowledges KONE Foundation, Academy of Finland (project No. 269095 and No. 296302). CHC station received support from Institut de Recherche pour le Développement (IRD) under both Jeune Equipe program attributed to LFA and support to ACTRIS-FR program. CHC received grants from Labex OSUG@2020 (Investissements d'avenir – ANR10 LABX56). Marco Pandolfi is funded by a Ramón y Cajal Fellowship (RYC-2013-14036) awarded by the Spanish Ministry of Economy and Competitiveness. The authors would like to express their gratitude to D. C. Carslaw and K. Ropkins for providing the OpenAir software used in this paper (Carslaw and Ropkins, 2012; Carslaw, 2012).



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





1    **Tables**
2
3    **Table 1:** List of ACTRIS observatories providing aerosol particle scattering measurements

| Observatory name | Country | Observatory code | Lat, Long | Altitude [m a.s.l.] | Placement (from EBAS metadata) | Inlet | Nephelometer model | Period (a) |
|---|---|---|---|---|---|---|---|---|
| *nordic and Baltic* | | | | | | | | |
| Birkenes II **(BIR)** | Norway | NO0002R | 58.3885 N, 8.252 E | 219 | regional | PM$_{10}$ | TSI3563 | 07/2009 −12/2015 |
| Hyytiälä **(SMR)** | Finland | FI0050R | 61.85N, 24.2833 E | 181 | regional | PM$_{10}$ | TSI3563 | 05/2006 −12/2015 |
| Pallas **(PAL)** | Finland | FI0096G | 67.97 N, 24.12 E | 565 | continental | PM$_5$; PM$_{2.5}$; PM$_{10}$ (*b*) | TSI3563 | 02/2000 −12/2015 |
| Vavihill **(VHL)** | Sweden | SE0011R | 56.0167 N, 13.15 E | 175 | continental | PM$_{10}$ | ECOTECH Aurora3000 | 03/2008 −04/2014 |
| Preila **(PLA)** | Lithuania | LT0015R | 55.35 N, 21.0667 E | 5 | coastal/marine | PM$_{10}$ | TSI3563 | 12/2012 −04/2014 |
| | | | | | | | | |
| *western* | | | | | | | | |
| Mace Head **(MHD)** | Ireland | IE0031R | 53.3258 N, -9.8994 E | 5 | coastal/marine | whole air | TSI3563 | 07/2001 −12/2013 |
| Cabauw **(CBW)** | The Netherlands | NL0011R | 51.9703 N, 4.9264 E | 1 | regional | PM$_{10}$ | TSI3563 | 01/2008 −12/2012 |
| SIRTA **(SIR)** | France | FR0020R | 48.7086 N, 2.1589 E | 162 | suburban | whole air | ECOTECH M9003 | 07/2012 −12/2013 |
| Observatory Perenne **(OPE)** | France | FR0022R | 48.5622 N, 5.505555 E | 392 | regional | whole air; PM$_{10}$ (*c*) | ECOTECH Aurora3000 | 09/2012 −12/2015 |
| Puy de Dome **(PUY)** | France | FR0030R | 45.7667 N, 2.95 E | 1465 | mountain | whole air | TSI3563 | 01/2007 −12/2014 |
| | | | | | | | | |
| *central* | | | | | | | | |
| Hohenpeissenberg **(HPB)** | Germany | DE0043G | 47.8 N, 11.0167 E | 985 | mountain | PM$_{10}$ | TSI3563 | 01/2006 −12/2015 |
| Ispra **(IPR)** | Italy | IT0004R | 45.8 N, 8.6333 E | 209 | semi regional | PM$_{10}$ | TSI3563 | 01/2004 −12/2014 |
| Melpitz **(MPZ)** | Germany | DE0044R | 51.53 N, 12.93 E | 86 | regional | PM$_{10}$ | TSI3563 | 01/2007 −12/2015 |
| Jungfraujoch **(JFJ)** | Switzerland | CH0001G | 46.5475 N, 7.985 E | 3578 | mountain | whole air | TSI3563 | 07/1995 −12/2015 |
| Mt. Cimone **(CMN)** | Italy | IT0009R | 44.1833 N, 10.7 E | 2165 | mountain | whole air | ECOTECH Aurora M9003; TSI 3563 (*d*) | 05/2007 −12/2015 |
| Košetice **(KOS)** | Czech Republic | CZ0007R | 49.58333N, 15.0833 E | 534 | regional | PM$_{10}$ | TSI3563 | 03/2013 − 12/2015 |
| | | | | | | | | |
| *eastern* | | | | | | | | |
| Beo Moussala **(BEO)** | Bulgaria | BG0001R | 42.1667 N, 23.5833 E | 2971 | mountain | whole air | TSI3563 | 03/2007 −12/2015 |
| K-Puszta **(KPS)** | Hungary | HU0002R | 46.9667 N, 19.5833 E | 125 | regional | PM$_1$; PM$_{10}$ (*e*) | TSI3563 | 05/2006 −12/2014 |
| | | | | | | | | |
| *south-western* | | | | | | | | |
| Montsec **(MSA)** | Spain | ES0022R | 42.0513 N, 0.44 E | 1570 | mountain | PM$_{2.5}$; PM$_{10}$ (*f*) | ECOTECH Aurora3000 | 01/2013 − 12/2015 |
| Izaña **(IZO)** | Spain | ES0018G | 28.309 N, -16.4994 E | 2373 | mountain | PM$_{10}$ | TSI3563 | 03/2008 − 12/2015 |
| Granada **(UGR)** | Spain | ES0020U | 37.164 N, -3.605 E | 680 | urban | whole air | TSI3563 | 01/2006 −12/2015 |
| Montseny **(MSY)** | Spain | ES1778R | 41.7667 N, 2.35 E | 700 | regional | PM$_{10}$ | ECOTECH Aurora3000 | 01/2010 −12/2015 |
| Madrid **(MAD)** | Spain | ES1778R | 40.4627 N, -3.717 E | 669 | sub-urban | PM$_{2.5}$; PM$_{10}$ (*g*) | ECOTECH Aurora3000 | 01/2014 − 12/2014 |
| | | | | | | | | |
| *south-eastern* | | | | | | | | |
| Finokalia **(FKL)** | Greece | GR0002R | 35.3167 N, 25.6667 E | 250 | coastal/marine | whole air; PM$_1$; PM$_{10}$ (*h*) | RR M903; Ecotech Aurora1000 (*i*) | 04/2004 −12/2015 |
| Athens **(DEM)** | Greece | GR0100B | 37.9905 N, 23.8095 E | 270 | sub-urban | PM$_{10}$ | ECOTECH Aurora3000 | 01/2012 −12/2015 |
| | | | | | | | | |
| *Arctic* | | | | | | | | |



| | | | | | | | | |
|---|---|---|---|---|---|---|---|---|
| Zeppelin **(ZEP)** | Svalbard (Norway) | NO0042G | 78.9067 N, 11.8883 E | 474 | arctic environment | PM$_{10}$ | TSI3563 | 07/2010 –12/2014 |
| | | | | | | | | |
| *Antarctic* | | | | | | | | |
| Troll **(TRL)** | Antarctica | NO0058G | -72.0167 N, 2.5333 E | 1309 | antarctic environment | whole air; PM$_{10}$ (*j*) | TSI3563 | 02/2007 –12/2015 |
| | | | | | | | | |
| *South America* | | | | | | | | |
| Mt. Chacaltaya **(CHC)** | Bolivia | BO0001R | -16.2000 N, -68.09999 E | 5240 | mountain | whole air | ECOTECH Aurora3000 | 01/2012 – 12/2015 (*k*) |

(a) Start-end of measurements; Total aerosol particle scattering was used as reference for measurement period; (b) PM$_5$ (2000-2005), PM$_{2.5}$ (2005-2008) and PM$_{10}$ (2008-2015); (c) whole air (2012-2013) and PM$_{10}$ (2014-2015); (d) ECOTECH Aurora M9003 during 2007-2013 and TSI 3563 (2014-2015); (e) PM$_1$ (2006-04/2008) and PM$_{10}$ (05/2008-2014); (f) PM$_{2.5}$ (2013) and PM$_{10}$ (2014-2015); (g) PM$_{10}$ from 03/2014; (h) whole air (2004-2008), PM$_{10}$ (2009-2011), PM$_1$ (2011-2012), PM$_{10}$ (2013-2015); (i) RR M903 during 2004-2011, Ecotech AURORA1000 during 2012-2015; (j) whole air (2007-2009) and PM$_{10}$ (2010-2015); (k) Only measurements performed during the year 2012 were used in this investigation.





**Table 2:** Trends of aerosol particle scattering coefficient ($\sigma_{sp}$), scattering Ångström exponent (SAE),
and backscatter fraction (BF). Three trends for SAE are reported: SAE calculated as linear fit using
three wavelengths (b-g-r); using the blue and the green wavelengths (b-g) and using the green and
red wavelengths (g-r). Trend results are reported for the whole period available at each station until
2015 (bold) and for the periods reported in Collaud Coen et al. (2013) and in Asmi et al. (2013).
Trends are considered as statistically significant if p-value < 0.05. Statistically significant increasing
or decreasing trends are highlighted with up ( ↟ ) and down ( ↡ ) red and green arrows, respectively.
Non-statistically significant increasing or decreasing trends are highlighted with up (↑) and down (↓)
grey arrows, respectively. Grey colored table cells highlight stations included in this work but not
included in the works from Collaud Coen et al. (2013) or Asmi et al. (2013). $: parameters removed
in this work and in the work from Collaud Coen et al. (2013) because of measurement gaps, low
data coverage or break points for one or more wavelengths.  #: Only available for 2014-2015; ± not
available.

| Station | period | Trend (This work) | | | | | MK Trend (Collaud Coen et al., 2013) | | | | | MK Trend (Asmi et al., 2013) | | |
|---|---|---|---|---|---|---|---|---|---|---|---|---|---|---|
| | | $\sigma_{sp}$ | SAE | | | BF | $\sigma_{sp}$ | SAE | | | BF | Particle number | | |
| | | | b-g-r | b-g | g-r | | | b-r | b-g | g-r | | N | N20 (20-500 nm) | N100 (100-500 nm) |
| *Nordic and Baltic* | | | | | | | | | | | | | | |
| PAL | **2000 - 2015** | ↑ | ↓(green) | ↓ | ↓(green) | ↟(red) | | | | | | | | |
| | 2000 - 2010 | ↓(green) | ↓(green) | $ | $ | ↑ | ↓ | ↑ | $ | $ | ↑ | | | |
| | 2001 - 2010 | ↓ | ↓(green) | $ | $ | ↑ | ↓ | ↑ | $ | $ | ↑ | ↓(green) (10-500 nm) | ↔ | ↟(red) |
| | 1996 - 2010 | | | | | | | | | | | ↓ (10-500 nm) | | |
| SMR | **2006 - 2015** | ↓(green) | ↑ | ↟(red) | ↑ | ↟(red) | | | | | | | | |
| | 1996 - 2011 | | | | | | | | | | | | ↓(green) | ↓(green) |
| | 2001 - 2010 | | | | | | | | | | | | ↓(green) | ↓(green) |
| *western* | | | | | | | | | | | | | | |
| MHD | **2001 - 2013** | ↓ | $ | $ | $ | $ | | | | | | | | |
| | 2000 - 2010 | | | | | | | | | | | ↓(green) (3-500 nm) | | |
| | 2001 - 2010 | ↑ | $ | $ | $ | $ | ↟(red) | $ | $ | $ | $ | ↑ (3-500 nm) | | |
| PUY | **2007 - 2014** | ↓ | ↓(green) | ↓(green) | ↓(green) | ↟(red) | | | | | | | | |
| *central* | | | | | | | | | | | | | | |
| HPB | **2006 - 2015** | ↓(green) | ↟(red) | ↟(red) | ↑ | ↟(red) | | | | | | | | |
| | 2001 - 2010 | | | | | | ↑ | $ | $ | $ | $ | | | |
| | 2002 - 2010 | | | | | | ↓ | $ | $ | $ | $ | | | |
| | 1995 - 2011 | | | | | | | | | | | ↑ (15-500 nm) | | |
| IPR | **2004 - 2014** | ↓(green) | ↑ | ↑ | ↑ | ↟(red) | | | | | | | | |
| MPZ | **2007 - 2015** | ↓ | ↓ | ↓ | ↓ | ↑ | | | | | | | | |
| | 1997 – 1998 and 2004 - 2010 | | | | | | | | | | | | ↑ | ↑ |
| JFJ | **1995 - 2015** | ↓ | $ | $ | $ | $ | | | | | | | | |
| | 1995 - 2010 | ↑ | $ | $ | $ | $ | ↑ | $ | $ | $ | $ | | | |
| | 1996 - 2010 | ↑ | $ | $ | $ | $ | ↑ | $ | $ | $ | $ | | | |
| | 2001 - 2010 | ↓ | $ | $ | $ | $ | ↓ | $ | $ | $ | $ | ↓(green) (10-500 nm) | | |
| | 1997 - 2010 | ↑ | $ | $ | $ | $ | | | | | | ↑ (10-500 nm) | | |
| CMN | **2007 - 2015** | ↓ | # | # | # | # | | | | | | | | |
| *eastern* | | | | | | | | | | | | | | |
| BEO | **2007 - 2015** | ↓ | ↓(green) | ↓(green) | ↓(green) | ↓ | | | | | | | | |



| KPS | 2006 - 2014 | ↑ | ↓ | ↓ | ↑ | ↑ | | | | | | | |
|-----|-------------|---|---|---|---|---|--|--|--|--|--|--|--|
| *south-western* | | | | | | | | | | | | | |
| IZO | 2008 - 2015 | ↓ | ↑ | ↑ | ↑ | $ | | | | | | | |
| UGR | 2006 - 2015 | ↓ | ↑ | ↑ | ↑ | ↑ | | | | | | | |

**Table 3:** Daytime (08:00 – 16:00 GMT) and nighttime (21:00 – 05:00 GMT) of $\sigma_{sp}$ trends by season calculated for the periods considered in this work. Sp: Spring; Su: Summer; Au: Autumn; Wi: Winter. Trends are considered as statistically significant if p-value < 0.05. Statistically significant increasing or decreasing trends are highlighted with up ( ↑ ) and down ( ↓ ) red and green arrows, respectively. Non statistically significant increasing or decreasing trends are highlighted with up ( ↑ ) and down ( ↓ ) grey arrows, respectively.

| | | SCATTERING | | | | | |
|---------|-------------|----|----|----|----|----|----|
| | | *daytime* | | *nighttime* | | *24h* | |
| **Station** | **period** | *Sp* | *Su* | *Sp* | *Su* | *Sp* | *Su* |
| | | *Au* | *Wi* | *Au* | *Wi* | *Au* | *Wi* |
| JFJ | 1995 - 2015 | ↓ | ↓ | ↓ | ↓ | ↓ | ↓ |
| | | ↑ | ↓ | ↑ | ↓ | ↑ | ↓ |
| HPB | 2006 - 2015 | ↓ | ↓ | ↓ | ↓ | ↓ | ↓ |
| | | ↓ | ↓ | ↓ | ↓ | ↓ | |
| PUY | 2006 - 2014 | ↓ | ↓ | ↓ | ↓ | ↓ | ↓ |
| | | ↓ | ↓ | ↓ | ↓ | ↓ | ↓ |
| CMN | 2007 - 2015 | ↓ | ↑ | ↓ | ↓ | ↓ | ↓ |
| | | ↓ | ↓ | ↓ | ↓ | ↓ | ↓ |
| BEO | 2007 - 2015 | ↓ | ↓ | ↓ | ↑ | ↓ | ↓ |
| | | ↓ | ↓ | ↓ | ↑ | ↓ | ↑ |
| IZO | 2008 - 2015 | ↓ | ↓ | ↓ | ↓ | ↓ | ↓ |
| | | ↑ | ↓ | ↑ | ↓ | ↑ | ↓ |





**Figures**

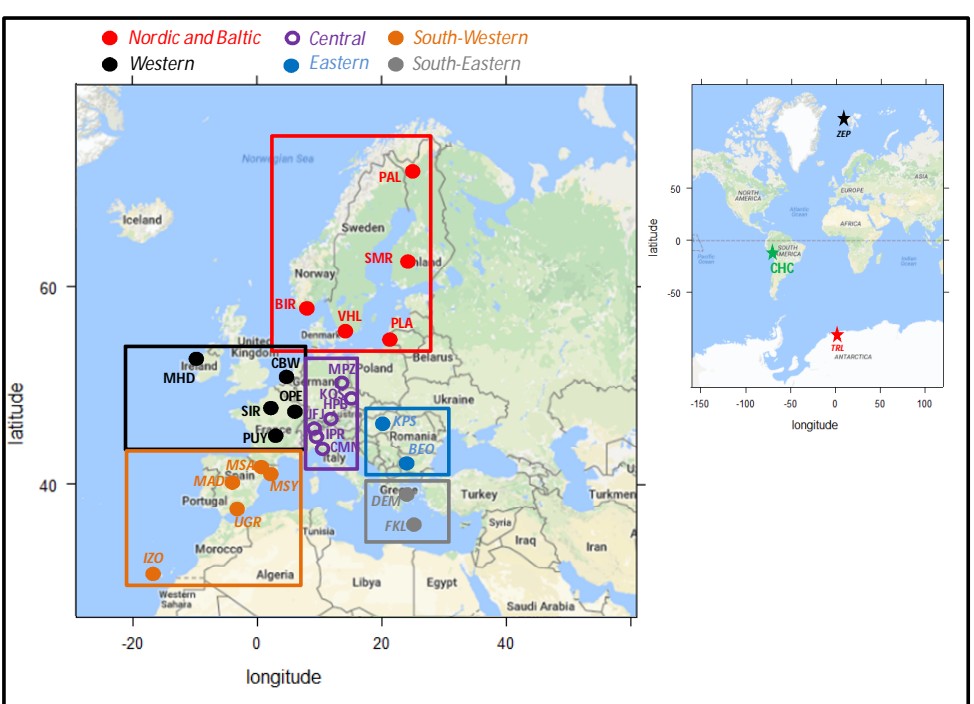

**Figure 1:** Location of the 28 ACTRIS stations included in this work.



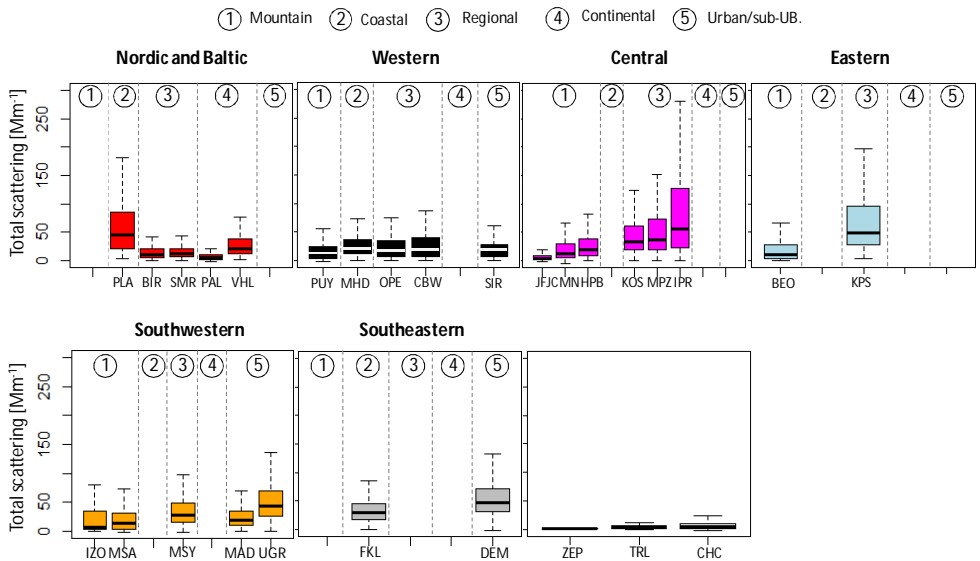

**Figure 2:** Total aerosol scattering coefficients in the green divided by geographical location. At SIR aerosol scattering was available only at 450 nm. Medians (horizontal lines in the boxes), percentiles 25th and 75th (lower and upper limits of the boxes, respectively) and percentiles 5th and 95th (lower and upper limits of the vertical dashed lines) are reported. Hourly data were used for the statistic. For each location data are ordered from mountain sites (1) to urban/sub-urban sites (5).

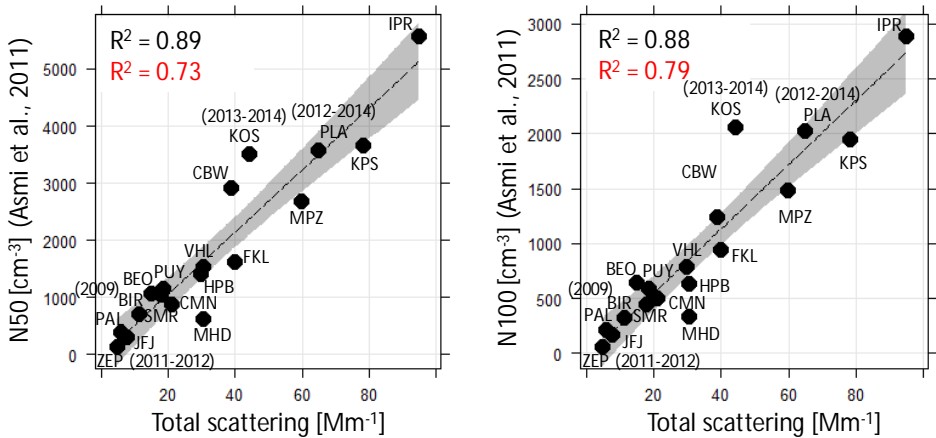

**Figure 3:** Relationship between N50 (mean particle number concentration between 50 nm and 500 nm) and N100 (mean particle number concentration between 100 nm and 500 nm) and mean aerosol particle scattering coefficient averaged over the period 2008 – 2009. For ZEP, BIR, KOS and PLA aerosol particle scattering measurements were not available during 2008 – 2009 and different period were used. $R^2$ highlighted in red were obtained using the median values.

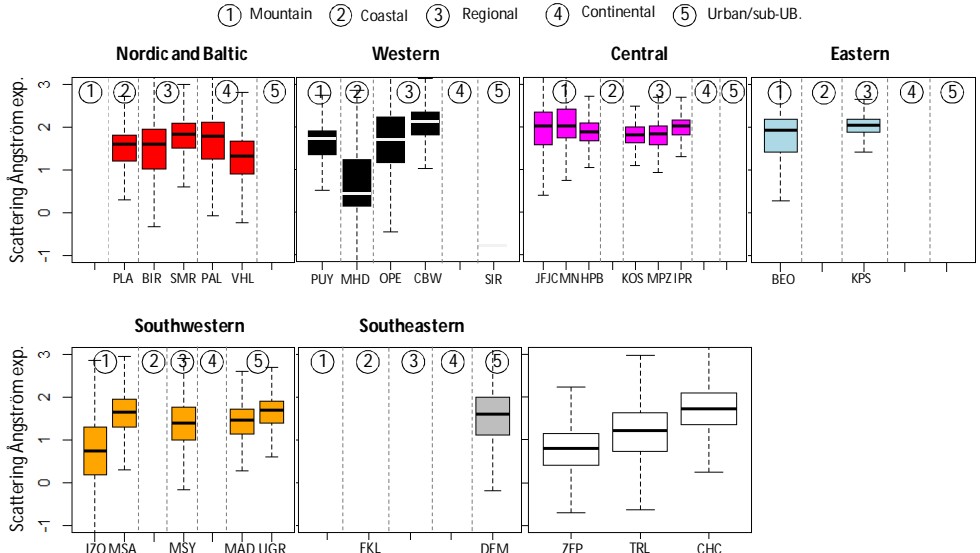

**Figure 4:** Scattering Ångström exponent divided by geographical location. Medians (horizontal lines in the boxes), percentiles 25$^{th}$ and 75$^{th}$ (lower and upper limits of the boxes, respectively) and percentiles 5$^{th}$ and 95$^{th}$ (lower and upper limits of the vertical dashed lines) are reported. For each location data are ordered from mountain sites to urban/sub-urban sites. At CHC, the SAE was calculated using the blue and green wavelengths.

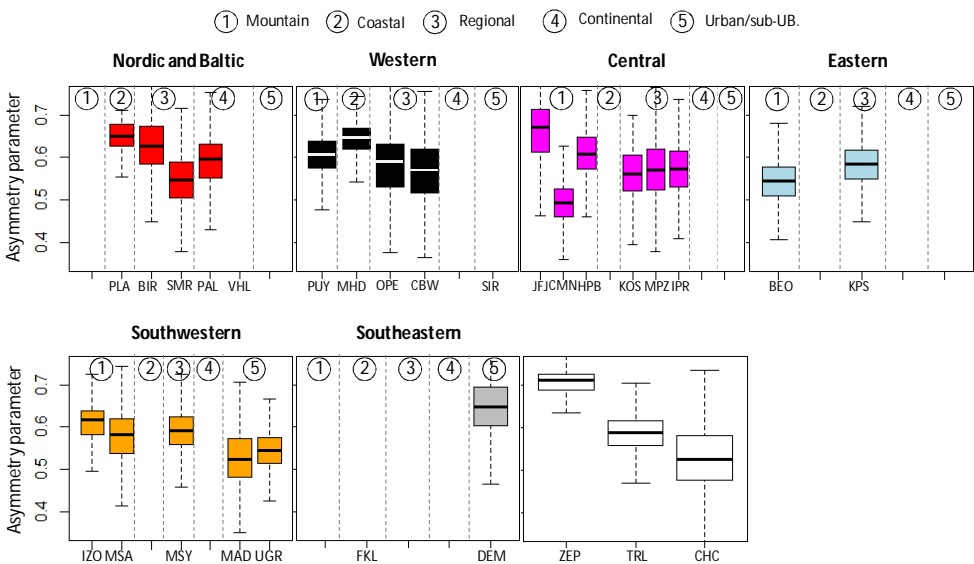

**Figure 5:** Asymmetry parameter in the green divided by geographical location. Medians (horizontal lines in the boxes), percentiles 25$^{th}$ and 75$^{th}$ (lower and upper limits of the boxes, respectively) and percentiles 5$^{th}$ and 95$^{th}$ (lower and upper limits of the vertical dashed lines) are reported. For each location data are ordered from mountain sites to urban/sub-urban sites.



**Figure 6**: Scatterplots between $\sigma_{sp}$ (x-axes) and SAE (right y-axes; red lines) and $g$ (left y-axes; black lines). Dashed lines represent median $\sigma_{sp}$ values at each station. At CHC, SAE was calculated using the blue and the green wavelengths.




**Figure 7**: Seasonal cycles of $\sigma_{sp}$ [Mm$^{-1}$] measured in the green nephelometer wavelength.




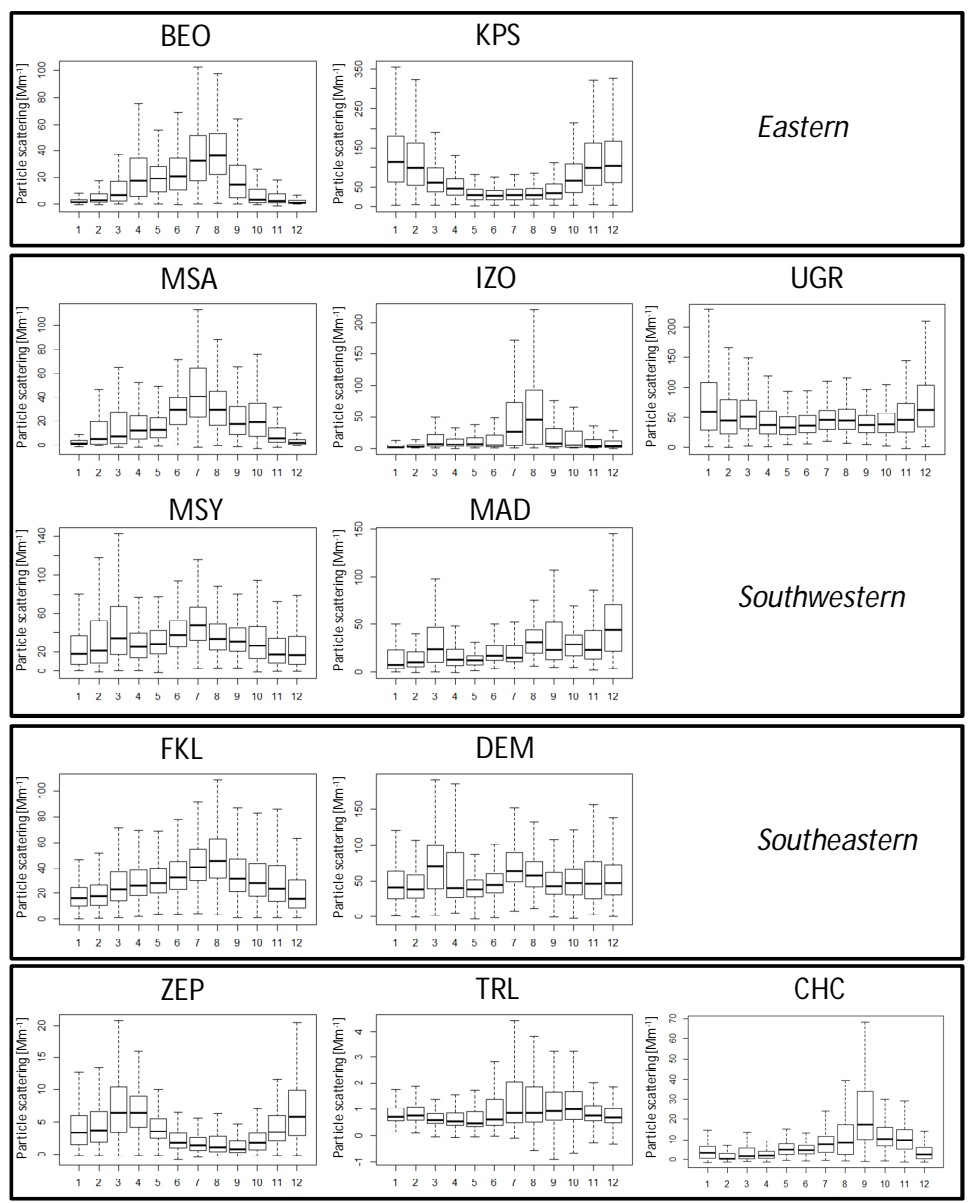

**Figure 7**: (Continued) Seasonal cycles of $\sigma_{sp}$ [Mm$^{-1}$] measured in the green nephelometer wavelength.





**Figure 8**: Seasonal cycles of SAE (calculated as linear fit using three nephelometer wavelengths)





**Figure 8**: (Continued) Seasonal cycles of SAE (calculated as linear fit using three nephelometer wavelengths).
At CHC the SAE was calculated using the blue and the green wavelengths.




**Figure 9**: Seasonal cycles of *g* (calculated for the green wavelength).





**Figure 9**: (Continued) Seasonal cycles of *g* (calculated for the green wavelength).