# Peer review of "A European aerosol phenomenology-6: Scattering properties of atmospheric aerosol particles from 28 ACTRIS sites"

_Atmospheric Chemistry and Physics, 2017_

## Referee Comment (RC1) · Anonymous Referee #1 · 17 Nov 2017

Review of Pandolfi et al 2017 – ACTRIS nephelometer measurements
This manuscript summarizes nephelometer measurements (in terms of monthly climatology, co-variance of scattering-related optical properties and trends) across the ACTRIS network in Europe. The manuscript also includes a few non-European sites supported by ACTRIS.  A lot of work went into bringing these data sets together and summarizing station differences (instrumentation, size cut, corrections…).

**General/Technical comments**

The manuscript includes a lot of information and clearly a lot of work was involved in getting that information into useable form for analysis, but, as written, it's really hard to digest.  Some of that would be helped by more editing (e.g., shorter paragraphs as described in my editorial comments below).

Why not compare scattering seasonality with Zanatta seasonality of absorption where possible?

Why not present SSA seasonality/trends at the same time? I realize an absorption measurement is needed to do so – but Zanatta's ACTRIS paper means those data exist for some sites and should be in some sort of consistent form.

I understand you want to include all ACTRIS supported light scattering measurements, but the non-European sites (TRL and CHC) are a distraction. They seem to only be included because they are part of ACTRIS but not for any strong scientific reason and are barely discussed in the text.  I think it is fine to include these stations in the supplemental materials if you must, but they should be eliminated from the main manuscript.

Page 6, Line 23 – the nomenclature 'regional' and 'continental' is a bit confusing. Perhaps change continental to remote or rural? Or combine to 'continental' and note that some are more polluted than others?  This would make sense as I think they are usually discussed together in the manuscript.

Page 7 – somewhere in Section 2 (under data treatment) you should explain where the data came from, i.e., was it downloaded directly from EBAS (if so, what level (presumably level 2)).  Or was it provided by data providers?  Or some combination of the two?  You should also say whether you performed additional quality checks on the data or did you take them "as is" from EBAS/data providers?  I would hesitate to assume that even Level 2 data downloaded from EBAS it is OK to use without further review (particularly if you are working with backscatter values of any wavelength and blue or red scattering values. Personal experience with the Level 2 nephelometer data in EBAS suggests that much of the data provider QC focuses primarily on the green scattering and the other nephelometer parameters may not get as much attention.  A further issue is that it seems that often data QC is done by different individuals in different years

and they don't look at years before and after their year to see if there are obvious differences.

Page 7, Line 15-18 – Later on you note that there are differences in operations from the GAW protocols – the measurement RH is one obvious difference from GAW protocols at many sites, so the statement 'the nephelometer instruments are run following the ACTRIS/GAW standards (WMO-GAW Report, 2016)' is not strictly true. Please revise.

Page 7, Line 29 – 'Due to the non-homogeneity of the light source' this should be made clearer. Was the light source causing measurement problems and is there a reference report describing this? Do you use the data from before the light source change? Why were different replacement light sources used at SIR and CMN and does this have implications for the data?

Page 8, Line 15-16 – "However, for SSA < 0.8 a correction scheme based on particle number size distribution should be used" Was this type of correction necessary for any of the stations? It looks from the table like it may have been performed for some stations, but was it necessary for those stations?

Page 8, Line 19-20 - Only at SIR, FKL, and CMN, $\sigma_{sp}$ data are not corrected for truncation because $\sigma_{sp}$ at these observatories was measured at one wavelength. Anderson and Ogren suggest corrections for the nephelometer when the SAE is not available. Doesn't the Mueller scheme also provide an option for correction when no SAE is present? I would expect FKL (a coastal site presumably with large sea salt aerosol) will be quite sensitive to not being corrected for truncation and uncorrected scattering will significantly underestimate the actual scattering.

Page 9, Line 10-11 – "However, the scattering enhancement due to a change in RH between 40% and 50% should be small." This very much depends on the aerosol type. See hygroscopicity work at several of these ACTRIS sites by Paul Zieger as well as more recent (2017) work on sea salt. How much do the results change if RH<40% is chosen instead of RH<50%? That would (a) be a useful finding and (b) indicate whether the statement above is reasonable (though it would not prove it one way or the other).

Page 9, Line 31 – is 1.5 a reasonable Angstrom exponent for a marine site? Seems a bit high to me

Page 10, Line 28 – I think it would be worthwhile to explain why the blue and red scattering have less coverage. Are the consistent problems with the wavelengths in particular nephelometers or are there data QC issues? (I realize some sites only have green measurements but it's unclear from your statement if those are the sites that don't have red/blue coverage (it wouldn't be expected!) or if it is other sites that do have those measurements but for whatever reason they aren't available. My experience with Ecotech neph data suggests the red wavelength is pretty unstable at many sites within the ACTRIS network and that

has implications for SAE climatologies and trends.

Page 12, Lines 4-10 – it would be good to include the physical interpretation of the skewness as well as the mathematical

Page 11, Line 34 - PAL is often considered an Arctic site (see e.g., Backman et al., AMT, 2017) so it may make more sense to group it with ZEP)

Page 14, Lines 19-21 – you have the data and can determine whether the inlet size cut had an effect on the Angstrom exponent.  Could plot monthly medians of SAE before and after the inlet was changed.

Page 15, Line 22 – why do BIR and PLA have a large particle peak?

Page 20, Lines 20-26 – these percentage comparisons are misleading – scattering Angstrom exponent is constrained between the values of approximately 0 and 3 and therefore you will not see anywhere close to the same percentage change as you would see for scattering which is unconstrained. I would remove lines 20-26

Page 25, Line 32 – PAL trends in SAE – you may want to discuss with John Backman – the Level 2 PAL data in EBAS has some suspicious red values for several years.

Page 27, Line 29 – are there day/night differences at coastal sites due to onshore/offshore flow?

**Editorial comments**
It would be good to have a native English speak read and edit the paper before resubmission. It would also be really helpful to the reader to organize the discussion better – the paragraphs are really long and it's occasionally hard to follow the arguments because of that.  I've suggested some places where the really long paragraphs could be broken into smaller paragraphs but some transitional sentences may be needed.

Break into smaller paragraphs: line 11 – start new paragraph, line 19 – start new paragraph, line 32 – start new paragraph

line 15 – there are evidences → there is evidence

line 17 – phrasing of 'would eventually unmask the global warming'

line 22-23 – phrasing of 'Several international projects are providing in the last decades important information on the atmospheric particle properties worldwide'

line 25 completed →complemented

line 26 - 27 USA or EMEP → USA and EMEP

line 30 – define RTD

Line 36 – You should be careful here. (1) EBAS also includes data from the IMPROVE network nephelometers which are operated outside at ground level and at ambient conditions with no size cut. These IMPROVE data aren't really comparable to the ACTRIS data sets discussed here. (2) Additionally there are other sites making nephelometer measurements that aren't providing the data to EBAS (see comment below for page 5, lines 20-22).

Line 7 'that decreasing or' → that a decreasing or

Lines 13-22 – should cite Sherman et al 2015 – it is an updated version of Delene and Ogren 2002

Lines 20-22 – this statement only relates to multi-station studies although Andrews et al included 3 mountain sites in Asia (WLG (China), PYR (Nepal), LLN(Taiwan)). Please rephrase to make clear that there are measurements outside of Europe and the US, but that many of those measurements have primarily been written about in isolation (e.g., not in the context of other sites). For example, FMI has reported on long-term optical properties in Saudi Arabia and South Africa. Paolo Laj et al have reported on aerosol optical properties in Nepal, and Paolo Artaxo in Brazil. Note: this manuscript doesn't really change this since it is focused on European sites as well and I think you should remove the non-European sites you do include to improve the discussion and flow.

Line 26 - 'related with aerosol phenomenology' → related to aerosol phenomenology

Line 31 – 'Zanatta' → 'and Zanatta'

Lines 1-13 should be in methods section 2.2.3, not introduction

Line 1 – delete 'In fact,'

Line 1 – add the word 'spectral' in front of $\sigma sp$

Line 5 – particles → particle

Line 7 – associated to → associated with

Line 7  - course → coarse

Line 11 – a better reference is the Andrews et al. 2006 reference which you also cite later.  Ogren 2006 is gray (not peer-reviewed) literature and the Andrews paper is the peer-reviewed version of it

Line 18 – performed → characterized

Line 19 – observatories → observatory

Line 19 – measurements →measurement

Line 20 – divided in -→ divided into

Line 22 – coastal – how close to the sea coast?

Page 6 -lines 26-36 and Page 7 lines 1-4 - Delete and say the categories for each station are given in table 1.  No need to have in paragraph form also.

Line 5 - Earlier you say mountain sites are higher than 1km, but then you characterize HPB at 985 m as a mountain.  Should change previous statement and say mountain sites are at or higher than 985 m to be consistent.

Line 9 – investigation → investigations

Lines 11-15 – delete - you say this in the trends section.

Line 25 - 'Most used nephelometer models are the'  → The most common nephelometers in the ACTRIS program are the

Line 27 - 'Other used models'→ Other models used

Line 37 – 'guarantee the quality and comparability of the data.'  This is a strong statement. There are RH control issues, wavelength, differences, data QC issues and many of the systems operate with different inlets/size cuts (e.g., whole air, pm2.5 pm 10).  Do stations have different procedures for dealing with negative values close to zero? How do they deal with local contamination?  There are unfortunately a lot of issues and while the nephs themselves are probably quite comparable when they are operated side-by-side as happens at a Leipzig workshop the operating conditions at the stations will vary and make the measurements less comparable.  What happens if a nephelometer fails a performance check – is the data prior to that time invalidated?  You discuss some of these issues below in the sections below but I recommend changing the text as follows:

{Recommended quality assurance procedures during on-site operation as described in GAW (WMO/GAW, 2016), guarantee the quality and comparability of the data. Moreover, most of the integrating nephelometers involved in ACTRIS have undergone performance checks at scheduled times at the World Calibration Center for Aerosol Physical properties of ACTRIS/GAW.}

Change text in red to text in blue→

{Recommended quality assurance procedures during on-site operation as described in GAW (WMO/GAW, 2016), help to ensure the quality and comparability of the data. Additionally, most of the integrating nephelometers involved in ACTRIS have undergone performance checks at scheduled times at the World Calibration Center for Aerosol Physical properties of ACTRIS/GAW.

2.2.2 Data treatment
The $\sigma$sp and $\sigma$bsp data reported to EBAS and used in this work are referenced to standard T (273.15 ºC) and P (1013 hPa) conditions. There are however station-to-station differences (e.g., sizecut, RH control, wavelength, data processing, etc) which are addressed in the sections below.

2.2.2.1}

Line 30 – should say here in the first sentence that the GAW (and ACTRIS?) protocol is RH<40%.  You say it later in the paragraph but it should be at the very beginning of the paragraph.

Lines 14-16 - this should be moved elsewhere – it is not related to the RH discussion. See my suggestion above which puts it after section heading 2.2.2

Line 20 – other used wavelengths → other wavelengths used

Line 23 – most used → most common

Line 24 – following Sections → following sections,

Line 27 – (and at CMN → (or at CMN

Line 28-29 - measured at different wavelength than 550 nm →measured at additional wavelengths to 550 nm

Line 6 - delete (with $\lambda_1 > \lambda_2$): this is not a necessary condition for the SAE equation you provide - you can flip things around so long as you are consistent,

e.g., if ssp550=40 and ssp700=30, then SAE =  -log(30/40)/log(700/550) = -log(40/30)/log(550/700) = 1.19

Line 8-9 – 'Here, the SAE is calculated as linear estimation of $\sigma_{sp}$ measured at the three available wavelengths.'  This statement is inconsistent with the equation describing Angstrom exponent (eq 1).  Further the relationship is not linear – that's why the equation has logs in it.  Please clarify what was meant.

Line 14 – given radiation → given direction

Line 18 – see previous comment about the Ogren 2006 citation

Section 3.1 – two points: (a) Why no separate section for Arctic sites? Could move discussion of ZEP and PAL to an arctic section (as suggested above I would remove discussion of TRL from main manuscript). (b) Why not order the discussion of sites from typically cleanest (arctic) →mountain→coastal→…→to typically dirtiest (urban).  That would make the discussion easier to follow I think.

Line 20 - placements → locations

Line 32 – because their → because of their

Line 35 – since these are →because they are

Line 27 – low scattering are → low scattering is (or low scattering value are)

Page 11, line 16 – page 12, line 10 – Split the paragraph into several smaller paragraphs: Start a new paragraph at page 11 line 25 for discussion of figure 3. Start a new paragraph at line 29.  Start a new paragraph page 12 line 4.

Lines 29 to page 12, line 4 – This discussion is hard to follow – in part because the site types as defined earlier aren't used consistently.

Line 13 – placements →environments

Line 30 – here you combine regional and continental stations for discussion – I do think it makes sense to call them clean and polluted continental stations – it would make the discussion simpler.

Line 31 – present → exhibit

Line 33-34 – I think 'linked to strong stable air with thermal inversion' this could be better phrased: linked to stable air due to strong thermal inversions

Line 33-34 – delete 'On the other side,'

Line 2 – instead of continental should say Nordic/Baltic continental sites because those are the subset of sites VHL is being compared to rather than all continental sites.

Section 3.2 – maybe would be easier to read if had subsections by station type or could have two subsections: (a) by geography (east to west) (b) by type (coastal, mountain, etc).  Right now they are a bit intertwined.

Line 16 – start new paragraph

Line 37 – the SAE data → the frequency plot of the SAE data

Line 13 – start new paragraph

Line 18 – cite work by Zieger et al 2010 at ZEP (their figure 4) which shows presence of sea salt at ZEP.

Line 24 – 'Differently than $\sigma$sp,…' →Unlike $\sigma$sp, …

Line 34 – start a new paragraph with 'Also at…'

Section 3.3 – maybe would be easier to read if had subsections by station type or could have two subsections: (a) by geography (east to west) (b) by type (coastal, mountain, etc).  Right now they are a bit intertwined.

Line 14 - start new paragraph

Line 14 – "..thus the lower BF the higher is *g..*" → change to "…with lower BF corresponding to higher g…"

Line 16-17 – "Higher *g* median values are in some cases observed at mountain sites compared to regional or urban environments" →At some mountain sites higher median g values are observed relative to the g values obtained at regional or urban locations.

Line 18 – "…European sector or HPB…" → …European sector and for HPB…

Line 19 – "However, exceptions are observed for example for CMN…"
→However, exceptions are observed.  For example, at CMN, …

Line 23 – start a new  paragraph

Line 35 – start a new paragraph

Line 10 – start a new paragraph

Line 10 – "Moreover, the refractive…" →The refractive …

Line 13 – "…did non linearly…" →…non-linearly decreased…

Line 16 – higher → larger

Line 17 – "On the other side, Obiso et al. (2017) showed…" → Obiso et al. (2017)
also showed…

Line 20 – "This kind of…" →These kind of …

Line 36 – retitle section BF and g vs scattering relationships.  Both are talked
about in the section so the title is a little confusing

Line 7 – Sherman et al., ACP 2015 also.
www.atmos-chem-phys.net/15/12487/2015/

Line 37 "or higher 1.5" →or higher than 1.5

Page 18, Line 20 – page 19, line 17 – this paragraph should be better organized
and split into two smaller paragraphs.  It's a bit hard to follow in its current form.

Section 3.4.1 – general comment – I would recommend first discussing the
SAE/scattering relationships and then relating it to the g/BF relationships.  Move
lines 20-23 to a paragraph at the end of the section.  Also, perhaps it would make
sense to split this section into aerosol types 'marine/dust' and 'anthropogenic'.

Section 3.5 – my personal suggestion would be to switch the order of sections
3.4 and 3.5.  Sections 3.1-3.3 talk about overall variability of the different
parameters on an annual basis.  It seems logical to talk next about the seasonal
variability and tie it into annual variability.  Once the individual parameter
variability has been discussed then it makes sense to look at how those
parameters co-vary (i.e., 3.5).  Note: that's also the order those topics are
presented in the conclusions.

Line 25 - ad → and

Line 6 – start new paragraph

Lines 11-13 – "At the southern station of MSA the observed less pronounced seasonal cycle of SAE could be related with the Saharan dust outbreaks which contrast the PBL transport of fine particles observed at other mountain sites" →it's unclear what is being said here; is MSA not impacted by dust?

Line 14 - "July-August being the Saharan dust outbreaks very"→July-August. The Saharan dust are very…

Line 7 – "being the SAE" →with the SAE

Split the southern Europe paragraph into several paragraphs. Line 15 –start new paragraph. Line 27 start new paragraph.

Line 3 – citation for recommendation about having more than 10 years of data for trend analysis?

Line 20 – start new paragraph

Page 24-25
This section (3.6.1) needs to be broken into smaller paragraphs! Line 9 – start new paragraph. Line 17 start new paragraph.

Line 14-15 – "A statistically significant decreasing trend of $\sigma$sp at IPR was also reported by Putaud et al. (2014) for the period 2002 – 2010."→ delete from text and put as footnote to Table 2.

Line 16-16 – "As reported in Table 2 statistically significant decreasing trend for $\sigma$sp is observed at around 50% of the stations considered here." →delete from text – put the 50% number in the first sentence of the first paragraph on line 2 of this page.

Line 8 – start new paragraph

Line 36 – start new paragraph

Line 9 start new paragraph about BF trends

Line 21 start new paragraph

Tables and Figures
Table 1 – Explain that the 'observatory code' is ACTRIS' code (or EBAS?) (or GAW?). Note where necessary any differences between GAW ids and station ids (e.g., Finokalia's GAW id is FIK and there may be others).

Figure 2 – you could put this on log scale

Figures 2, 4 and 5 – how would this look if you had panes for different site types and then organized by geographical region? For example, you would have a pane for mountain sites and then sections for Nordic (empty), western (puy), central (jfj, cmn, hpb), etc. I think doing that would make it easier to see the east west shift and also commonalities among site types. You could keep the boxes colored by geographic region within each pane. You could do a similar thing with figure 6.

Supplemental materials
Table S1 – how is this table organized? It's not alphabetical or by geography. Or by instrument type.

Table S2 - should provide number of points with RH>40% for each station (i.e., how many points are above the GAW and ACTRIS protocol value) as well as the number of points with RH>50%. Caption should state that table is organized by decreasing number of points with RH>50%.

Figure S1 – should make the x-axes cover the same range (0-80%?) and draw a vertical line at 40% (GAW protocol value) and 50% (value chosen for this paper). Lines could be added indicating the median RH distribution as a function of season. Explain why different widths of bars on distribution plots (presumably because different numbers of data points). I think it would make more sense to use consistent widths for the distribution plots.

Table S3 – state in caption what colors lambda1, lambda2, and lambda3 correspond with. (not the wavelength because that obviously changes with instrument and time period)

Figure S2 – remove map – that's already a figure in the manuscript. Explain why different widths of bars on distribution plots (presumably because different numbers of data points). I think it would make more sense to use consistent widths for the distribution plots.

Figure S3 – remove map – that's already a figure in the manuscript. Explain why different widths of bars on distribution plots (presumably because different

numbers of data points). I think it would make more sense to use consistent widths for the distribution plots.

Figure S4 – remove map – that's already a figure in the manuscript.  Explain why different widths of bars on distribution plots (presumably because different numbers of data points). I think it would make more sense to use consistent widths for the distribution plots.

Figure S6b – explain the different colors of dots in the figure caption.  Perhaps you could make the bars in figure S6a the same color as the dots in figure S6b and then use some other color for the dots in figure S6a

Figure S8 – has MTC instead of CMN

---

## Referee Comment (RC2) · Anonymous Referee #2 · 9 Feb 2018

**GENERAL REMARKS**

The manuscripts approaches a European aerosol phenomenology on scattering properties of atmospheric aerosol particles from 28 ACTRIS sites, based on measurements with various types of integrating nephelometers. The manuscript focuses exclusively on ACTRIS sites in Europe, with the addition of Arctic and Antarctic stations and one mountain station in Bolivia, operated jointly by ACTRIS and local partners. The efforts for bringing this extensive data set together are huge and the richness of data is the major contribution of this work to an important scientific discussion on the long-term evolution of extensive and intensive aerosol properties in industrialized and rural re-

gions of the world. However, in its current version, the manuscript is very difficult to read and it is hard to follow any story line of the paper. Major revisions are required to make this manuscript acceptable for publication in ACP.

Major concerns refer to the organization of the manuscript, presentation of results and lacking of joint in-depth analyses of the observations in combination with existing publications on the long-term evolutions of aerosol optical properties.

Before resubmission of the manuscript, the following concerns should be considered:

1. Organization of the manuscript and the presentation of results:

- The abstract is far too long and requires substantial shortening.

- The results sections in Chapter 3 require a substructure to become more readable. Currently, paragraphs are too long and there is no line of arguments the reader can follow. Instead the paragraphs are highly descriptive and do not point at the key messages of the data analyses. In its current form, the reader likely misses a large part of the information contained in this manuscript.

- The conclusions section is not well structured and a lot of information may get lost. Sharpening and shortening of this section is recommended. Finally, what are the key points of the presented work? This should be clearly expressed.

2. Consideration of published results on the long-term evolution of aerosol optical properties:

Although previous studies on aerosol scattering properties (see, e.g., Andrews et al. (2011), Collaud Coen et al., (2013) and Zanatta et al., (2016)) are mentioned in the text, they have been included into the discussion only briefly. A discussion is missing (or got lost due to the current organization of the manuscript) whether the results presented here are in agreement with the published results, or whether they provide new findings, and then the question would be: where do differences come from? Finally, the discussion of the evolution of aerosol light scattering properties together with light

absorption properties published in Zanatta et al. (2016) is completely missing.

MINOR COMMENTS

1. In section 2.2.2.1, the applied truncation correction is described. For light absorbing aerosol (single scattering albedo < 0.8) a method proposed by Müller et al. (2011) is applied. However, Massoli et al. (2009) presented a correction scheme particularly for light-absorbing aerosols measured with the TSI Model 3563 Integrating Nephelometer. It should be briefly discussed why this approach has not been applied at the stations running TSI Model 3563 instruments.

2. In section 2.2.2.3, wavelength adjustments were conducted for sites where multiple-wavelengths data do not exist. For that purpose SAE values were prescribed (1.5 for FKL and SIR and 2.0 for CMN). The choice of these SAE values should be justified.

3. Before resubmission, checking of the language by a native speaker is highly recommended.

REFERENCES

Andrews, E., Ogren, J. A., Bonasoni, P., Marinoni, A., Cuevas, E., Rodriguez, S., Sun, J. Y., Jaffe, D. A., Fischer, E. V., Baltensperger, U., Weingartner, E., Coen, M. C., Sharma, S., Macdonald, A. M., Leaitch, W. R., Lin, N. H., Laj, P., Arsov, T., Kalapov, I., Jefferson, A., and Sheridan, P.: Climatology of aerosol radiative properties in the free troposphere, Atmos. Res., 102, 365-393, doi: 10.1016/j.atmosres.2011.08.017, 2011.

Collaud Coen, M., Andrews, E., Asmi, A., Baltensperger, U., Bukowiecki, N., Day, D., Fiebig, M., Fjaeraa, A. M., Flentje, H., Hyvärinen, A., Jefferson, A., Jennings, S. G., Kouvarakis, G., Lihavainen, H., Lund Myhre, C., Malm, W. C., Mihapopoulos, N., Molenar, J. V., O'Dowd, C., Ogren, J. A., Schichtel, B. A., Sheridan, P., Virkkula, A., Weingartner, E., Weller, R., and Laj, P.: Aerosol decadal trends – Part 1: In-situ optical measurements at GAW and IMPROVE stations, 13, 869-894, doi: 10.5194/acp-13-869-2013, 2013.

Massoli, P., Murphy, D. M., Lack, D. A., Baynard, T., Brock, C. A., and Lovejoy, E. R.: Uncertainty in light scattering measurements by TSI Nephelometer: Results from Laboratory studies and implications for ambient measurements, Aerosol Sci. Technol., 43, 1064–1074, 2009.

Müller, T., Laborde, M., Kassell, G., and Wiedensohler, A.: Design and performance of a three-wavelength LED-based total scatter and backscatter integrating nephelometer, Atmos. Meas. Tech., 4, 1291-1303, doi: https://doi.org/10.5194/amt-4-1291-2011, 2011.

Zanatta, M., Gysel, M., Bukowiecki, N., Muller, T., Weingartner, E., Areskoug, H., Fiebig, M., Yttri, K. E., Mihalopoulos, N., Kouvarakis, G., Beddows, D., Harrison, R. M., Cavalli, F., Putaud, J. P., Spindler, G., Wiedensohler, A., Alastuey, A., Pandolfi, M., Sellegri, K., Swietlicki, E., Jaffrezo, J. L., Baltensperger, U., and Laj, P.: A European aerosol phenomenology-5: Climatology of black carbon optical properties at 9 regional background sites across Europe, Atmos. Environ., 145, 346-364, doi: 10.1016/j.atmosenv.2016.09.035, 2016.

---

## Author Comment (AC1) · 14 Mar 2018

Dear editor, dear Referees, We would like to thank you for all your comments. This input has allowed us to refine the manuscript by adding more thorough detailed explanations, to correct some points and to improve in a large sense the manuscript.

Below the answers to the Referee's comments

*Review of Pandolfi et al 2017 – ACTRIS nephelometer measurements. This manuscript summarizes nephelometer measurements (in terms of monthly climatology, co-variance of scattering-related optical properties and trends) across the ACTRIS network in Europe. The manuscript also includes a few non European sites supported by ACTRIS. A lot of work went into bringing these data sets together and summarizing station differences (instrumentation, size cut, corrections…).*

*General/Technical comments*
*The manuscript includes a lot of information and clearly a lot of work was involved in getting that information into useable form for analysis, but, as written, it's really hard to digest. Some of that would be helped by more editing (e.g., shorter paragraphs as described in my editorial comments below).*

1) *Why not compare scattering seasonality with Zanatta seasonality of absorption where possible?*

Figure 3 was modified to include a comparison of total scattering coefficients with the absorption coefficients reported in Zanatta et al. (2016; Table 8). The text describing Figure 3 has been accordingly modified.

"The observed variation is consistent with the differences in particulate matter (PM) mass concentrations, PM chemical composition, particle number concentration and absorption coefficients observed across Europe, as described for example by Putaud et al. (2010), Asmi et al. (2011) and Zanatta et al. (2016).
Figures 3a and 3b show the relationship between the mean particle number concentration measured at different stations during 2008 to 2009 (and reported in Asmi et al. (2011)) and the mean $\sigma sp$ measured over the same period (where available). As reported in Figure 3, good correlations are observed between N50 (Figure 3a: mean/median particle number between 50 nm and 500 nm) and N100 (Figure 3b: mean/median particle number between 100 nm and 500 nm) and mean $\sigma sp$. Figure 3c shows the relationship (for some stations) between absorption coefficients reported in Zanatta et al. (2016) and the total scattering. The good correlations reported in Figure 3c (especially high for the winter and autumn periods) suggest an increase of both scattering and absorption coefficients with increasing aerosol loading."

[Figure]

**Figure 3:** Relationship between: (a) N50 (mean particle number concentration between 50 nm and 500 nm), (b) N100 (mean particle number concentration between 100 nm and 500 nm), (c) absorption coefficient and mean aerosol particle total scattering coefficient. (a) and (b): data averaged over the period 2008 to 2009. For ZEP, BIR, KOS and PLA aerosol particle scattering measurements were not available during 2008 to 2009 and different periods were used. R2 values, highlighted in red, were obtained using the median values. (c) Data averaged as in Zanatta et al. (2016).

*2) Why not present SSA seasonality/trends at the same time? I realize an absorption measurement is needed to do so – but Zanatta's ACTRIS paper means those data exist for some sites and should be in some sort of consistent form.*

Certainly, absorption coefficient measurements are available at the majority of the observatories included in the present manuscript. However, a manuscript presenting SSA climatology and trends at ACTRIS sites is already under preparation. Moreover, we think that adding SSA (and consequently absorption) would make the manuscript too long and difficult to read. For this reason and with the permission of the Reviewer we prefer not to include SSA in the present manuscript.

*3) I understand you want to include all ACTRIS supported light scattering measurements, but the non-European sites (TRL and CHC) are a distraction. They seem to only be included because they are part of ACTRIS but not for any strong scientific reason and are barely discussed in the text. I think it is fine to include these stations in the supplemental materials if you must, but they should be eliminated from the main manuscript.*

Following the Reviewer suggestion, the figures related to the two non-European sites TRL and CHC were moved to the supporting material. The Figure is reported below:

[Figure]

**Figure S3**: Total scattering, Scattering Angstrom Exponent, Asymmetry parameter and Backscatter fraction for the non-European stations TRL and CHC.

*4) Page 6, Line 23 – the nomenclature 'regional' and 'continental' is a bit confusing. Perhaps change continental to remote or rural? Or combine to 'continental' and note that some are more polluted than others? This would make sense as I think they are usually discussed together in the manuscript.*

We thank the Reviewer for this comment which considerably simplifies the discussion. As suggested by the Reviewer the PAL station was included in the "Arctic" category in the revised version of the

manuscript. Thus, following the EBAS definition, the only continental station is VHL. Consequently, the category "continental" was removed in the revised version of the manuscript and VHL was included in the new "regional/rural" category.

**5)** ***Page 7 – somewhere in Section 2 (under data treatment) you should explain where the data came from, i.e., was it downloaded directly from EBAS (if so, what level (presumably level 2)). Or was it provided by data providers? Or some combination of the two? You should also say whether you performed additional quality checks on the data or did you take them "as is" from EBAS/data providers? I would hesitate to assume that even Level 2 data downloaded from EBAS it is OK to use without further review (particularly if you are working with backscatter values of any wavelength and blue or red scattering values. Personal experience with the Level 2 nephelometer data in EBAS suggests that much of the data provider QC focuses primarily on the green scattering and the other nephelometer parameters may not get as much attention. A further issue is that it seems that often data QC is done by different individuals in different years and they don't look at years before and after their year to see if there are obvious differences.***

In order to take into account the Reviewer comments #5 and #53, the section 2 was modified.
Thus, the following sentences were added at the end of the section 2.2.1 and beginning of section 2.2.2:

[revised manuscript text omitted]

6) **Page 7, Line 15-18 – Later on you note that there are differences in operations from the GAW protocols – the measurement RH is one obvious difference from GAW protocols at many sites, so the statement 'the nephelometer instruments are run following the ACTRIS/GAW standards (WMO-GAW Report, 2016)' is not strictly true. Please revise.**

In the revised version of the manuscript we removed the sentence at Pag. 7, Line 15-18:

"The stations included in this work report the data to ACTRIS and GAW/EMEP, consequently the data are quality assured given that the nephelometer instruments are run following the ACTRIS/GAW standards (WMO-GAW Report, 2016) and regularly inter-compared."

We removed the sentence because it is a repetition: The sentence is moved at the end of the Paragraph 2.2.1 and modified as follow (see also the replies to the Reviewer comments #5 and #54):

"Recommended quality assurance procedures during on-site operation, as described in GAW (WMO-GAW Report, 2016), help to ensure the quality and comparability of the data. The nephelometers included in this investigation are regularly calibrated using span gas and are zero adjusted using particle-free air. Additionally, most of the integrating nephelometers employed in ACTRIS have undergone a schedule of performance checks at the World Calibration Center for Aerosol Physics of ACTRIS/GAW."

"2.2.2 Data treatment
Data used in this investigation include hourly averaged Level 2 aerosol particle scattering data downloaded from the ACTRIS/EBAS Data Centre web portals (www.actris.nilu.no; www.ebas.nilu.no; last downloads August 2017). The σsp and σbsp data reported to EBAS and used in this work are referenced to standard T (273.15 °C) and P (1013 hPa) conditions. Data consistency is critical when comparing many years' worth of data from different stations. In this work, the Level 2 scattering data were further reviewed in order to ensure a high quality of the data presented. There are however station-to-station differences (e.g. sizecut, RH control, wavelength, data processing, etc.) which are addressed in the sections below."

7) **Page 7, Line 29 – 'Due to the non-homogeneity of the light source' this should be made clearer. Was the light source causing measurement problems and is there a reference report describing this? Do you use the data from before the light source change? Why were different replacement light sources used at SIR and CMN and does this have implications for the data?**

The problems with the angular intensity function of the nephelometer Ecotech model M9003 have been discussed by Muller et al. (2009). This reference was added to the sentence at Pag. 7, Line 29. Yes, in the manuscript we also used the scattering data collected before the change of the light source. We do not know why different replacements were used for SIR and CMN, but both data providers confirmed (personal communication) that both nephelometers performed well after the change of the light source. Unfortunately, the reports of these intercomparisons are not available in the ECAC (European Center for Aerosol Calibration) webpage.

The sentence was modified as follow:

"Due to the non-homogeneity of the angular distribution of light intensity of model M9003 (cf. Müller et al., 2009), the light source was changed at SIR in 2013 with the AURORA3000 light source and at CMN in 2009 with an opal glass light source. After the change of the light sources, both nephelometers were examined at the World Calibration Center for Aerosol Physical properties in Leipzig and both performed very well (personal communication from CMN and SIR data providers)."

8) *Page 8, Line 15-16 – "However, for SSA < 0.8 a correction scheme based on particle number size distribution should be used" Was this type of correction necessary for any of the stations? It looks from the table like it may have been performed for some stations, but was it necessary for those stations?*

To the best of our knowledge al stations involved in this investigation applied the Angstrom-based correction scheme to correct the nephelometer scattering data. In the revised version of the manuscript, the sentence was modified as follow:

"These schemes consist of a simple linear correction based on the scattering Ångström exponent (SAE) determined from the raw nephelometer data to take account of the size-distribution-dependent truncation error. It has been demonstrated that these simple correction schemes are accurate for a wide range of atmospheric aerosols and that the uncertainties in the corrections are not expected to be larger than 2% for an aerosol particle population with a single scattering albedos (SSA) greater than 0.8 (Bond et al., 2009)."

9) *Page 8, Line 19-20 - Only at SIR, FKL, and CMN, σsp data are not corrected for truncation because σsp at these observatories was measured at one wavelength. Anderson and Ogren suggest corrections for the nephelometer when the SAE is not available. Doesn't the Mueller scheme also provide an option for correction when no SAE is present? I would expect FKL (a coastal site presumably with large sea salt aerosol) will be quite sensitive to not being corrected for truncation and uncorrected scattering will significantly underestimate the actual scattering.*

During the discussion phase further feedback was needed with data providers regarding the correction of scattering data for some stations. For BIR, MPZ, PUY and TRL the information provided in the metafiles was that scattering data were NOT corrected for truncation, whereas data submitted to EBAS were actually corrected by data providers. Table S1 has been accordingly modified and Figures and Tables changed in order to take into account for this error in the metafiles.

Concerning the Reviewer comment, our opinion is that the correction of the single-wavelength scattering measurements at SIR, FKL and CMN could introduce additional, undesired errors. This is very probably the reason why scattering data provided to EBAS by these three stations were not corrected for truncation.

At CMN and SIR the Ecotech model M9003 was used. Correction schemes for this nephelometer model were provided by Müller et al. (2009). At FKL two nephelometers were used, namely the RR M903 (until 2011) and the Ecotech 1000 (from 2012).

Concerning the Ecotech model M9003, Figure 4 from Müller et al. (2009) shows that the correction factor for this nephelometer model is around 0.97 – 1-00 for SAE around 1.5 – 2.0. The mean SAE at CMN (from TSI measurements performed during 2014-2015; cf. Table S5 in supporting information of this manuscript) was around 2. We do not have an estimation of SAE for SIR observatory which is located in a suburban area 20 km south of Paris. However, the mean SAE at other stations located in West of Europe (PUY, OPE and CBW) was around 1.6-2.0 (after excluding the coastal/marine MHD observatory also located in the Western European sector). Thus, we suppose that the SAE at SIR is probably not low. Thus, the correction factor for both SIR and CMN is close to one. For these reasons we prefer not to correct scattering data from CMN and SIR.

The correction scheme for the RR M903 (deployed at FKL) was also provided by Müller et al. (2009). However, to the best of our knowledge no correction scheme was provided for the Ecotech 1000 which was used at FKL from 2012. Moreover, at FKL the inlet was changed many times (cf. Table 1: whole air (2004-2008), PM10 (2009-2011), PM1 (2011-2012), PM10 (2013-2015)) and the correction factors

provided in literature are also function of the size cut-off used. For these reasons, we prefer not to correct scattering data collected at FKL.

With the permission of the Reviewer, we prefer not to correct the 1-λ scattering data from CMN, SIR and FKL. Thus, in the revised version of the manuscript all scattering data presented are corrected for truncation with the exception of CMN, SIR and FKL. The following sentence was added at the end of the Paragraph 2.2.2.1:

"Data from the integrating nephelometers used here are corrected for non-ideal illumination of the light source (deviation from a Lambertian distribution of light) and for truncation of the sensing volumes in the near-forward (around 0-10°) and near-backward (around 170-180°) directions (Müller et al., 2009 and Anderson and Ogren, 1998). Correction schemes have been provided by Müller et al. (2009; 2011) for the RR M903 and Ecotech models M9003 and AURORA3000, and by Anderson and Ogren (1998) for the TSI3563. These schemes consist of a simple linear correction based on the scattering Ångström exponent (SAE) determined from the raw nephelometer data to take account of the size-distribution-dependent truncation error. It has been demonstrated that these simple correction schemes are accurate for a wide range of atmospheric aerosols and that the uncertainties in the corrections are not expected to be larger than 2% for an aerosol particle population with a single scattering albedos (SSA) greater than 0.8 (Bond et al., 2009).
The majority of the $\sigma sp$ data in the EBAS database are corrected for non-ideal illumination and for truncation by the data providers. Exceptions are the scattering data submitted for KOS, MHD, PLA, CMN, FKL and SIR. Scattering data from KOS, MHD and PLA were corrected in this work using the correction scheme provided by Anderson and Ogren (1998) (cf. Table S1 of the Supporting Material). The $\sigma sp$ data collected at CMN, FKL and SIR are not corrected because the nephelometers deployed at these three stations provide scattering only at one wavelength, thus preventing the estimation of the SAE. Given that the nephelometer correction factors vary as a function of SAE, the assumption of a constant correction factor to correct the 1-λ scattering data could introduce undesired noise. Moreover, at SIR and CMN, the $\sigma sp$ is measured with the single wavelength Ecotech nephelometer model M9003 (until 2013 at CMN). The correction curve from Müller et al. (2009; Figure 4) provides a correction factor of around 0.97 to 1.0 for the M9003 for a SAE of around 1.5 to 2. Using the TSI3563 scattering measurements performed at CMN during 2014-2015, we estimated a mean SAE of around 2 for CMN (cf. Table S5). Thus, given the rather small effect of the correction factor estimated for the Ecotech M9003, scattering data from CMN and SIR were not corrected in this work. At FKL the nephelometer models RR M903 (until 2011) and Ecotech 1000 (from 2012) were used (cf. Table 1). To the best of our knowledge, no correction scheme has been provided for the Ecotech 1000. Moreover, at FKL, the inlet was changed many times (cf. Table 1) and the correction factors provided in the literature are a strong function of the size cut-off used. For these reasons, scattering data collected at FKL are not corrected in this investigation."

10) **Page 9, Line 10-11 –"However, the scattering enhancement due to a change in RH between 40% and 50% should be small." This very much depends on the aerosol type. See hygroscopicity work at several of these ACTRIS sites by Paul Zieger as well as more recent (2017) work on sea salt. How much do the results change if RH<40% is chosen instead of RH<50%? That would (a) be a useful finding and (b) indicate whether the statement above is reasonable (though it would not prove it one way or the other).**

In order to take into account the Reviewer comment the following sentence:

"Estimating the aerosol particle light scattering enhancement due to an increase of RH from 40% to 50% is difficult using the data available here because $\sigma sp$ measurements at RH>40% are not evenly distributed over the measurement periods. In fact, at the majority of the stations RH higher than 40% is registered mostly in summer. However, the scattering enhancement due to the change in RH between 40% and 50% should be small and will not exceed around 3-5% even for more hygroscopic particles (e.g. Fierz-Schmidhauser et al., 2010a,b)."

was removed and replaced with the following sentence:

"Estimating the aerosol particle light scattering enhancement due to an increase of RH from 40% to 50% is difficult using the data available here because the $\sigma sp$ measurements at a RH>40% are not evenly distributed over the measurement periods, with the majority of the stations registering a RH higher than 40% during the summer. Moreover, the chemical composition of atmospheric aerosol particles is an important factor determining the magnitude of the scattering enhancement due to water uptake, which can then change from one site to another (e.g. Fierz-Schmidhauser et al., 2010a,b; Zieger et al., 2014, 2017). However, the scattering

enhancement due to a change in RH between 40% and 50% should be small and will not exceed few percent even for more hygroscopic particles (e.g. Fierz-Schmidhauser et al., 2010a,b). Table S2 in the Supporting Material reports the percentage of hourly σsp values collected in the range 40%<RH<50% whereas the frequency distributions of the measured RH are shown in Figure S1."

Following the Reviewer comment we performed a sensitivity study (Figures reported below) to understand the effect of using data at RH<50 (rather than RH<40%) on aerosol optical properties. However, we confirm what already stated in the manuscript, i.e. that "Estimating the aerosol particle light scattering enhancement due to an increase of RH from 40% to 50% is difficult using the data available here because σsp measurements at RH>40% are not evenly distributed over the measurement periods. In fact, at the majority of the stations RH higher than 40% is registered mostly in summer."

The following figures show the difference between σsp, SAE, BF and g using data collected at RH<50% and RH<40%. We performed this test for the observatories where the percentage of measurements at RH>50 is high (cf. Table S2), namely: IPR, CMN and SIR. However, we have also performed the same test using two stations (MSA and MPZ) where the RH never exceeded the 40% threshold. For MSA observatory, we compared the measurements performed at RH<40% with those performed at RH<30%. For the MPZ observatory, we compared the measurements performed at RH<30% with those performed at RH<20%. The objective of using MSA and MPZ is demonstrating that the differences in σsp, SAE, BF and g cannot be only ascribed to differences in RH, but that removing data during specific periods of the year also strongly contributes to the observed differences.

In general, differences up to -30% ÷ +35% were observed for IPR, SIR and CMN for scattering. However, the differences for the intensive optical properties (SAE, BF and g) were much smaller and never exceeding -7% ÷ +3%
The differences (%) are calculated in such a way these are positive when scattering at RH<50% exceeds the scattering at RH<40%. However, for CMN, the difference is negative for example during the month of February and September, thus this difference cannot ascribed to the scattering enhancement due to water uptake. Rather, this difference is due to the fact that we are removing non-simultaneous data in order to perform the calculations.

Moreover, we observed that also for MSA and MPZ the differences for scattering can be high (-20% ÷ +50%) despite the lower RH taken into account at these two stations.

With the permission of the Reviewer, we would like to use data collected at RH<50% in the revised version of the manuscript. Moreover, we do not think that adding the Figures below to the Supporting Material can add useful information to the reader.

[Figure]

[Figure]

[Figure]

[Figure]

**11) Page 9, Line 31 – is 1.5 a reasonable Angstrom exponent for a marine site? Seems a bit high to me.**

Surely, a value of 1.5 is probably high for a marine site. In the revised version of the manuscript we provide a range of SAE value and we calculated the scattering at 550 using SAE of 1 and 1.5.
The sentence:

" At FKL and SIR, where SAE is not available and assuming a SAE of 1.5, the difference by adjusting to 550 nm is 4.9% at FKL and 26% at SIR, respectively. The higher difference at SIR is due to the fact that measurements at this station are performed at 450 nm."

Was modified as follow:

"At FKL and SIR, where the SAE is not available, and assuming a reasonable SAE range between 1.5 and 1.0, the difference due to the adjustment to 550 nm is 4.9-3.0% at FKL and 26-18% at SIR. The higher difference at SIR is due to the fact that measurements at this station are performed at 450 nm."

**12) Page 10, Line 28 – I think it would be worthwhile to explain why the blue and red scattering have less coverage. Are the consistent problems with the wavelengths in particular nephelometers or are there data QC issues? (I realize some sites only have green measurements but it's unclear from your statement if those are the sites that don't have red/blue coverage (it wouldn't be expected!) or if it is other sites that do have those measurements but for whatever reason they aren't available. My experience with Ecotech neph data suggests the red wavelength is pretty unstable at many sites within the ACTRIS network and that has implications for SAE climatologies and trends.**

As reported in Table S3, the data coverage for scattering at the three wavelengths is very similar at all observatories. Only at CMN the blue and red wavelengths have much lower data coverage because the three wavelengths nephelometer was implemented starting from 2014 (only green measured until 2014). At all other observatories the data coverage of scattering is very similar for the three wavelengths.

In order to clarify this point the sentence:

"Exception are σsp measurements in the blue (450 nm) and in the red (700 nm) and σbsp measurements at CMN where the three wavelengths nephelometer was implemented starting from 2014.",

was replaced with the following sentence:

"Exceptions are the σsp measurements at CMN in the blue (450 nm) and red (700 nm) wavelengths which have much less data coverage compared to the green wavelength because the three wavelength nephelometer was implemented at CMN in 2014. Consequently, also the SAE and g have low data coverage at CMN."

Moreover, we agree with the Reviewer that the red wavelength is especially critical for the Ecotech nephelometer. For this reason, for example, the red was not used at CHC station because of wrong calibrations which effect was especially evident in the red. At all other stations, the red wavelength of Ecotech seemed acceptable. We decided to include the red wavelengths of the majority of Ecotech nephelometers after inspecting the differences between SAE calculated between blue and green and between green and red.

**13) Page 12, Lines 4-10 – it would be good to include the physical interpretation of the skewness as well as the mathematical**

Following the reviewer comment the sentence:

"Positive skewness is usually observed for positive defined parameters having a frequency distribution with a pronounced right tail indicating the presence of high positive values."

was replaced with the following sentence:

"The skewness is defined as the third standardized moment of a probability distribution and it is a measure of the asymmetry of the probability distribution. Its value can be positive or negative. Positive skewness is usually observed for parameters which are defined to be positive and it indicates that the tail on the right side of the distribution is longer or fatter than that on the left side. Thus, for a right-skewed distribution, the mass of the distribution is concentrated on the left, and there is a higher probability of measuring a high value compared to a left-skewed distribution. Figure S2 in the Supporting Material shows the frequency and cumulative frequency distributions for σsp for each station, evidencing the presence of these right-skewed tails."

**14) Page 11, Line 34 - PAL is often considered an Arctic site (see e.g., Backman et al., AMT, 2017) so it may make more sense to group it with ZEP)**

Following the Reviewer suggestion the category "Arctic" has been included in the revised version of the manuscript. The PAL and ZEP stations have been included in the new category "Arctic".

**15) Page 14, Lines 19-21 – you have the data and can determine whether the inlet size cut had an effect on the Angstrom exponent. Could plot monthly medians of SAE before and after the inlet was changed.**

As suggested by the reviewer we plotted the SAE monthly medians (attached below) before and after inlet change for KPS [$PM_1$ (2006-04/2008) and $PM_{10}$ (05/2008-2014)], PAL [$PM_5$ (2000-08/2005), $PM_{2.5}$ (08/2005-2007) and $PM_{10}$ (2008-2015)], TRL [whole air (2007-2009) and $PM_{10}$ (2010-2015)], and MSA [$PM_{2.5}$ (2013-03/2014) and $PM_{10}$ (04/2014-2015)]. For these stations we calculated the %diff of SAE for different periods: 1) before and after the inlet change (red lines in Figure B), and 2) for periods without inlet changes (black lines in Figure B).
As reported in Figures A and B, the natural variability of aerosol particles over time (black lines in Fig.B) can cause changes in SAE of the same order of those possibly caused by the inlet change (red lines in Fig.B).

Thus, it is very difficult to evaluate the effect of the inlet change on SAE given that measurements performed with different cutoffs are not simultaneous. For this reason, we also report in the figures below the same analysis for MPZ and MSA where the inlet was not changed during the sampling period. The aim of reporting the comparison of SAE for different periods for MPZ and MSA is confirming that natural variability also can considerably affect the variability of SAE.
At MSA the %diff of SAE between the two selected periods is similar to the %diff observed for PAL or TRL (before and after the inlet change). At MPZ the %diff is smaller suggesting more homogeneity in some microphysical properties of atmospheric particles such as size (cf. Figures S4 in Supporting Material). From Fig. S4 we can say that also at KPS more homogeneity in particles size can be expected (very narrow SAE frequency distribution as for MPZ). In fact, at KPS the %diff is rather small compared to the other stations.

In conclusion, we think that the analysis of SAE before and after the inlet change cannot be used to determine how much the inlet change affected the calculated SAE.
Moreover, the analysis of the trends of scattering and SAE for PAL has been already reported in literature for the period 2000-2010 (Collaud Coen et al., 2013). Furthermore, Lihavainen et al. (2015a) assumed that the inlet changes at PAL had only minor effects on scattering, because the number concentration of coarse particles is very low at this observatory.

In conclusion, we do not think that adding the Figures below to the Supporting Material can help understanding the effects of the inlet change. Moreover, given the huge amount of data provided by many of the stations involved in this investigation, some latitude in data processing amongst stations should be deemed acceptable. Thus, despite the possible effect of the inlet changes, a detailed picture of scattering properties of surface aerosol particles is clearly provided with the data used here.

[Figure]

**Figure A**

[Figure]

**Figure B**

**16) Page 15, Line 22 – why do BIR and PLA have a large particle peak?**

These peaks with low SAE at BIR and PLA are also probably due to the presence of marine aerosols at these sites.
The sentence was modified as follow:

"BIR and PLA also show an enhanced left peak in the SAE frequency distributions likely due to the presence of coarse marine aerosols at these sites."

**17) Page 20, Lines 20-26 – these percentage comparisons are misleading – scattering Angstrom exponent is constrained between the values of approximately 0 and 3 and therefore you will not see anywhere close to the same percentage change as you would see for scattering which is unconstrained. I would remove lines 20-26.**

Yes. We agree. The sentence was removed in the revised version of the manuscript.

**18) Page 25, Line 32 – PAL trends in SAE – you may want to discuss with John Backman – the Level 2 PAL data in EBAS has some suspicious red values for several years.**

Yes. It is true that Level 2 EBAS PAL data has some suspicious red values. We detected such problem during the double check we performed on the data. In the manuscript we already removed data collected at PAL during: the whole year 2017; July and August 2002; September and October 2003; May-June-July 2015.

**19) Page 27, Line 29 – are there day/night differences at coastal sites due to onshore/offshore flow?**

Onshore/offshore flows surely have important effects at coastal sites. However, we do not know if these flows have any effect at the mountain sites considered for the day/night analysis presented in the manuscript.

**Editorial comments:**

**20) It would be good to have a native English speak read and edit the paper before resubmission.**

Following the Reviewer suggestion, the manuscript has been revised and edited by a native English speak.

**21) It would also be really helpful to the reader to organize the discussion better – the paragraphs are really long and it's occasionally hard to follow the arguments because of that. I've suggested some places where the really long paragraphs could be broken into smaller paragraphs but some transitional sentences may be needed.**

Following the Reviewer suggestions, smaller paragraphs were added in the revised version of the manuscript

**22) Page 4**

**Break into smaller paragraphs: line 11 – start new paragraph, line 19 – start new paragraph, line 32 – start new paragraph**

Done

**23) line 15 – there are evidences -> there is evidence**

Done

**24) line 17 – phrasing of 'would eventually unmask the global warming'**

Following the Reviewer comment, the sentence was modified as follow:

"In fact, there is evidence suggesting that the observed (and projected) decrease in emissions of anthropogenic aerosol particles in response to air quality policies will eventually exert a positive aerosol effective radiative forcing at the top of the atmosphere (Rotstayn et al., 2013). Thus, current emission controls could both enhance climate warming while improving air quality (e.g. Stohl et al., 2015)."

**25) line 22-23 – phrasing of 'Several international projects are providing in the last decades important information on the atmospheric particle properties worldwide'**

Following the Reviewer comment, the sentence was modified as follow:

"In recent decades, several international projects have provided important information on atmospheric particle properties worldwide."

**26) line 25 completed ->complemented**

Done

**27) line 26 - 27 USA or EMEP -> USA and EMEP**

Done

**28) line 30 – define RTD**

The sentence has been modified as follow:

"... and from short-term RTD (Research and Technological Development) projects such as EUCAARI (European Integrated Project on Aerosol Cloud Climate and Air Quality Interactions; http://www.cas.manchester.ac.uk/resprojects/eucaari/)."

**29) Line 36 – You should be careful here. (1) EBAS also includes data from the IMPROVE network nephelometers which are operated outside at ground level and at ambient conditions with no size cut. These IMPROVE data aren't really comparable to the ACTRIS data sets discussed here. (2) Additionally there are other sites making nephelometer measurements that aren't providing the data to EBAS (see comment below for page 5, lines 20-22).**

We agree with the Reviewer that ACTRIS and IMPROVE nephelometer data are not really comparable. The following sentence has been added:

"However, EBAS also includes data from the IMPROVE network nephelometers, which latter are operated at ambient conditions with no size cut, as a result of which these IMPROVE data are not directly comparable to the ACTRIS dataset discussed in this investigation."

**Page 5**

**30) Line 7 'that decreasing or' -> that a decreasing or**

Done

**31) Lines 13-22 – should cite Sherman et al 2015 – it is an updated version of Delene and Ogren 2002**

The reference Sherman et al. (2015) was added to the text and the bibliography.

**32) Lines 20-22 – this statement only relates to multi-station studies although Andrews et al included 3 mountain sites in Asia (WLG (China), PYR (Nepal), LLN(Taiwan)). Please rephrase to make clear that there are measurements outside of Europe and the US, but that many of those measurements have primarily been written about in isolation (e.g., not in the context of other sites). For example, FMI has reported on long-term optical properties in Saudi Arabia and South Africa. Paolo Laj et al have reported on aerosol optical properties in Nepal, and Paolo Artaxo in Brazil. Note: this manuscript doesn't really change this since it is focused on European sites as well and I think you should remove the non-European sites you do include to improve the discussion and flow.**

We agree with the Referee that this manuscript doesn't really change the fact that all the multi-sites studies presenting aerosol particle optical properties mostly focus on Europe and US. For this reason the following sentence:

"Thus, the number of papers reporting aerosol particle optical properties measured at different sites is rather scarce and unfortunately almost inexistent outside Europe and the United States."

was removed in the revised version of the manuscript.
Moreover, as suggested by the Reviewer, the non European sites have been removed from the manuscript and presented mostly in the Supporting Material.

**33) Line 26 - 'related with aerosol phenomenology' -> related to aerosol Phenomenology**

Done

**34) Line 31 – 'Zanatta' -> 'and Zanatta'**

Done

**Page 6**

**35) Lines 1-13 should be in methods section 2.2.3, not introduction**

Lines 1-13 have been moved to the Section 2.2.3, as follow:

"2.2.3 Calculation of aerosol particle intensive optical properties
Starting from the spectral $\sigma_{sp}$ measurements performed at the ACTRIS observatories, three intensive aerosol particle optical parameters can be estimated, namely; the scattering Ångström exponent (SAE), the backscattering fraction (BF) and the asymmetry parameter (g). These intensive properties do not depend on the PM mass concentration and are directly related to aerosol particle properties such as size, shape, size distribution and chemical composition. The SAE can be considered as a proxy for the aerosol particle size range with a higher (lower) SAE associated with predominance of fine (coarse) aerosol particles (e.g. Seinfeld and Pandis, 1998; Esteve et al., 2012; Valenzuela et al., 2015 among others). The BF and g parameters are calculated quantities that influence the variability of the radiative forcing efficiency and that represent the angular light scattering of aerosol particles. For computational efficiency, the angular light scattering is often represented by a single value (BF, $\sigma_{sp}/\sigma_{bsp}$ or g) (Andrews et al., 2006)."

**36) Line 1 – delete 'In fact,'**

Done

**37) Line 1 – add the word 'spectral' in front of $\sigma_{sp}$**

Done

**38) Line 5 – particles -> particle**

Done

**39) Line 7 – associated to -> associated with**

Done

**40) Line 7 - course -> coarse**

Done

**41) Line 11 – a better reference is the Andrews et al. 2006 reference which you also cite later. Ogren 2006 is gray (not peer-reviewed) literature and the Andrews paper is the peer-reviewed version of it**

Done

**42) Line 18 – performed -> characterized**

Done

**43) Line 19 – observatories -> observatory**

Done

**44) Line 19 – measurements ->measurement**

Done

**45) Line 20 – divided in -> divided into**

Done

**46) Line 22 – coastal – how close to the sea coast?**

The sentence has been modified as follow:

"coastal: includes observatories located close to the coast (<1-4 km);"

**47) Page 6 -lines 26-36 and Page 7 lines 1-4 - Delete and say the categories for each station are given in table 1. No need to have in paragraph form also.**

Done

**Page 7**

**48) Line 5 - Earlier you say mountain sites are higher than 1km, but then you characterize HPB at 985 m as a mountain. Should change previous statement and say mountain sites are at or higher than 985 m to be consistent.**

The sentence has been modified as follow:

"mountain: includes those observatories located at more than 985 m above sea level (the lowest altitude among the mountain observatories included here);".

**49) Line 9 – investigation -> investigations**

Done

**50) Lines 11-15 – delete - you say this in the trends section**

Done

**51) Line 25 - 'Most used nephelometer models are the' -> The most common nephelometers in the ACTRIS program are the**

Done

**52) Line 27 - 'Other used models' -> Other models used**

Done

**53) Line 37 – 'guarantee the quality and comparability of the data.' This is a strong statement. There are RH control issues, wavelength, differences, data QC issues and many of the systems operate with different inlets/size cuts (e.g., whole air, pm2.5 pm 10). Do stations have different procedures for dealing with negative values close to zero? How do they deal with local contamination? There are unfortunately a lot of issues and while the nephs themselves are probably quite comparable when they are operated side-by-side as happens at a Leipzig workshop the operating conditions at the stations will vary and make the measurements less comparable. What happens if a nephelometer fails a performance check – is the data prior to that time nvalidated? You discuss some of these issues below in the sections below but I recommend changing the text as follows:**
**{Recommended quality assurance procedures during on-site operation as described in GAW (WMO/GAW, 2016), guarantee the quality and comparability of the data. Moreover, most of the integrating nephelometers involved in ACTRIS have undergone performance checks at scheduled times at the World Calibration Center for Aerosol Physical properties of ACTRIS/GAW.}**
**Change text in red to text in blue ·**

*{Recommended quality assurance procedures during on-site operation as described in GAW (WMO/GAW, 2016), help to ensure the quality and comparability of the data. Additionally, most of the integrating nephelometers involved in ACTRIS have undergone performance checks at scheduled times at the World Calibration Center for Aerosol Physical properties of ACTRIS/GAW. 2.2.2 Data treatment The σsp and · σbsp data reported to EBAS and used in this work are referenced to standard T (273.15 °C) and P (1013 hPa) conditions. There are however station-to-station differences (e.g., sizecut, RH control, wavelength, data processing, etc) which are addressed in the sections below. 2.2.2.1}*

Following the Reviewer comment the sentence was changed as follow:

"Recommended quality assurance procedures during on-site operation, as described in GAW (WMO-GAW Report, 2016), help to ensure the quality and comparability of the data. The nephelometers included in this investigation are regularly calibrated using span gas and are zero adjusted using particle-free air. Additionally, most of the integrating nephelometers employed in ACTRIS have undergone a schedule of performance checks at the World Calibration Center for Aerosol Physics of ACTRIS/GAW."

"2.2.2 Data treatment
Data used in this investigation include hourly averaged Level 2 aerosol particle scattering data downloaded from the ACTRIS/EBAS Data Centre web portals (www.actris.nilu.no; www.ebas.nilu.no; last downloads August 2017). The σsp and σbsp data reported to EBAS and used in this work are referenced to standard T (273.15 °C) and P (1013 hPa) conditions. Data consistency is critical when comparing many years' worth of data from different stations. In this work, the Level 2 scattering data were further reviewed in order to ensure a high quality of the data presented. There are however station-to-station differences (e.g. sizecut, RH control, wavelength, data processing, etc.) which are addressed in the sections below."

*Page 8*
**54) Line 30 – should say here in the first sentence that the GAW (and ACTRIS?) protocol is RH<40%. You say it later in the paragraph but it should be at the very beginning of the paragraph.**

Done

*Page 9*
**55) Lines 14-16 - this should be moved elsewhere – it is not related to the RH discussion. See my suggestion above which puts it after section heading 2.2.2**

The sentence was moved to the beginning of the section 2.2.2.

**56) Line 20 – other used wavelengths -> other wavelengths used**

Done

**57) Line 23 – most used · most common**

Done

**58) Line 24 – following Sections -> following sections,**

Done

**59) Line 27 – (and at CMN -> (or at CMN**

Done

**60)        Line 28-29 - measured at different wavelength than 550 nm -> measured at additional wavelengths to 550 nm**

Done

**Page 10**

**61)        Line 6 - delete (with λ1 > λ2): this is not a necessary condition for the SAE equation you provide - you can flip things around so long as you are consistent, e.g., if ssp550=40 and ssp700=30, then SAE = -log(30/40)/log(700/550) = -log(40/30)/log(550/700) = 1.19**

Done

**62) Line 8-9 – 'Here, the SAE is calculated as linear estimation of σsp measured at the three available wavelengths.' This statement is inconsistent with the equation describing Angstrom exponent (eq 1). Further the relationship is not linear – that's why the equation has logs in it. Please clarify what was meant.**

The sentence was changed as follow:

"Here, the SAE is derived from a multispectral log linear fit based on the three nephelometer wavelengths."

**63) Line 14 – given radiation -> given direction**

Done

**64) Line 18 – see previous comment about the Ogren 2006 citation**

Done

**Page 11**

**65) Section 3.1 – two points: (a) Why no separate section for Arctic sites? Could move discussion of ZEP and PAL to an arctic section (as suggested above I would remove discussion of TRL from main manuscript). (b) Why not order the discussion of sites from typically cleanest (arctic) ->mountain ->coastal ->...-> to typically dirtiest (urban). That would make the discussion easier to follow I think.**

We agree with the Reviewer. A separate section for Arctic sites (ZEP and PAL) has been introduced in the revised version of the manuscript. The TRL station (together with CHC) was moved in the Supporting material. Sections 3.1 and 3.5 were modified accordingly to the Reviewer comment with the discussion presented from cleanest sites to urban sites. Figures and Tables were also accordingly modified.

**66) Line 20 - placements -> locations**

Done

**67) Line 32 – because their -> because of their**

Done

**68) Line 35 – since these are -> because they are**

Done

**69) Line 27 – low scattering are -> low scattering is (or low scattering value are)**

Done

70) **Page 11, line 16 – page 12, line 10 – Split the paragraph into several smaller paragraphs: Start a new paragraph at page 11 line 25 for discussion of figure 3. Start a new paragraph at line 29. Start a new paragraph page 12 line 4.**

Done

**71) Lines 29 to page 12, line 4 – This discussion is hard to follow – in part because the site types as defined earlier aren't used consistently.**

Following the Reviewer comments the presentation of the results is organized differently in the revised version of the manuscript. The sentence (from pag. 11, line 29 to pag.12, line 4) was moved at the beginning of section 3.1 as follow:

"3.1 Variability of $\sigma$sp
Figure 2 shows the box-and-whiskers plots of $\sigma$sp measured at the stations included in this investigation. In Figure 2, the observatories are grouped based on their placement and ordered according to their geographical location. Table S4 and Figure S2 in the Supplementary Material report, respectively, the statistics of $\sigma$sp (mean, standard deviation, minimum and maximum values and 5th, 25th, 50th, 75th, and 95th percentiles) and frequency and cumulative frequency distributions.
In each geographical sector, an increasing gradient of $\sigma$sp is generally observed when moving from mountain to regional and to urban sites. Thus, the $\sigma$sp values measured at mountain sites are lower than the measurements made at other locations (coastal to urban) even if exceptions are observed in some sectors.
A large range of $\sigma$sp coefficients is observed across the network, ranging from median values lower than 10 Mm-1 to values higher than 40 Mm-1. Overall, the lowest $\sigma$sp is on average measured at remote stations because of either: a) their altitude, for example JFJ is located in Central Europe at more than 3500 m a.s.l. and CHC in Bolivia is at around 5300 m a.s.l. (cf. Figure S3), or b) because of their large distance from pollution sources, for example the Arctic ZEP and PAL stations, TRL station (cf. Figure S3) and some regional sites in the Nordic and Baltic sector such as BIR and SMR. Higher $\sigma$sp values (medians > 40 Mm-1) are on average registered at more polluted sites, such as some urban sites in Southern Europe (UGR and DEM), some regional sites in Eastern and Central Europe (KPS and IPR, respectively) and one coastal site in the Nordic and Baltic sector (PLA). The observed variation is consistent with the differences in particulate matter (PM) mass concentrations, PM chemical composition, particle number concentration and absorption coefficients observed across Europe, as described for example by Putaud et al. (2010), Asmi et al. (2011) and Zanatta et al. (2016)."

**Page 12**
**72) Line 13 – placements ->environments**

Done

**73) Line 30 – here you combine regional and continental stations for discussion – I do think it makes sense to call them clean and polluted continental stations – it would make the discussion simpler.**

Following the Reviewer comments the PAL station is included among the Arctic stations together with ZEP. Thus, the only other continental station included in this work is VHL. For this reason, the category continental was removed in the revised version of the manuscript. The category Regional/rural was added/modified.
See also the reply to the Reviewer comment #4: "We thank the Reviewer for this comment which considerably simplifies the discussion. As suggested by the Reviewer the PAL station was included in the "Arctic" category in the revised version of the manuscript. Thus, following the EBAS definition, the only continental station is VHL. Consequently, the category "continental" was removed in the revised version of the manuscript and VHL was included in the new "regional/rural" category."

**74) Line 31 – present -> exhibit**

Done

**75) Line 33-34 – I think 'linked to strong stable air with thermal inversion' this could be better phrased: linked to stable air due to strong thermal inversions**

Done

**76) Line 33-34 – delete 'On the other side,'**

Done

*Page 13*
**77) Line 2 – instead of continental should say Nordic/Baltic continental sites because those are the subset of sites VHL is being compared to rather than all continental sites.**

In the revised version of the manuscript the category "continental" was removed and the continental VHL station included in the new category "Regional/rural". Please, see reply to Reviewer comment #4.

*Page 14*
**78) Section 3.2 – maybe would be easier to read if had subsections by station type or could have two subsections: (a) by geography (east to west) (b) by type (coastal, mountain, etc). Right now they are a bit intertwined.**

Following the Reviewer suggestion two subsections were added to the section 3.2, namely:

3.2.1 Variability of SAE by geographical sector, and
3.2.2 Variability of SAE by station type

**79) Line 16 – start new paragraph**
Done

*80) Line 37 – the SAE data -> the frequency plot of the SAE data*
Done

*Page 15*
*81) Line 13 – start new paragraph*
Done

*82) Line 18 – cite work by Zieger et al 2010 at ZEP (their figure 4) which shows presence of sea salt at ZEP.*
Done

*83) Line 24 – 'Differently than σsp,...' -> Unlike σsp, ...*
Done

*84) Line 34 – start a new paragraph with 'Also at...'*
Done

*Page 16*
*85) Section 3.3 – maybe would be easier to read if had subsections by station type or could have two subsections: (a) by geography (east to west) (b) by type (coastal, mountain, etc). Right now they are a bit intertwined.*

Following the Reviewer suggestion two subsections were added to the section 3.2, namely:

3.3.1 Variability of *g* by geographical sector, and
3.3.2 Variability of *g* by station type

*86) Line 14 - start new paragraph*
Done

*87) Line 14 – "..thus the lower BF the higher is g.." · change to "...with lower BF corresponding to higher g..."*
Done

*88) Line 16-17 – "Higher g median values are in some cases observed at mountain sites compared to regional or urban environments" -> At some mountain sites higher median g values are observed relative to the g values obtained at regional or urban locations.*
Done

*89) Line 18 – "...European sector or HPB..." -> ...European sector and for HPB...*
Done

*90) Line 19 – "However, exceptions are observed for example for CMN..." -> However, exceptions are observed. For example, at CMN, ...*
Done

*91) Line 23 – start a new paragraph*
Done

*92) Line 35 – start a new paragraph*
Done

*Page 17*
*93) Line 10 – start a new paragraph*
Done

*94) Line 10 – "Moreover, the refractive…" • The refractive …*
Done

*95) Line 13 – "…did non linearly…" • …non-linearly decreased…*
Done

*96) Line 16 – higher -> larger*
Done

*97) Line 17 – "On the other side, Obiso et al. (2017) showed…" • Obiso et al. (2017) also showed…*
Done

*98) Line 20 – "This kind of…" • These kind of …*
Done

*99) Line 36 – retitle section BF and g vs scattering relationships. Both are talked about in the section so the title is a little confusing*
Done

*Page 18*
*100)    Line 7 – Sherman et al., ACP 2015 also www.atmos-chem-phys.net/15/12487/2015/*

The reference has been added to the main text and to the bibliography.

*101)    Line 37 "or higher 1.5" • or higher than 1.5*
Done

*102)    Page 18, Line 20 – page 19, line 17 – this paragraph should be better organized and split into two smaller paragraphs. It's a bit hard to follow in its current form.*

Please, see reply to the comment below (#103)

*103)    Section 3.4.1 – general comment – I would recommend first discussing the SAE/scattering relationships and then relating it to the g/BF relationships. Move lines 20-23 to a paragraph at the end of the section.*
*Also, perhaps it would make sense to split this section into aerosol types 'marine/dust' and 'anthropogenic'.*

Reply to comments #102 and #103: With the permission of the Reviewer we would like to discuss first the *g*-scattering relationship and then the SAE-scattering relationship. This because the *g*-scattering relationships are very similar at all sites. The *g* always increases with increasing σsp. Then, we comment the variations of SAE-scattering relationships among the observatories in function of the *g*-scattering relationships.

Moreover, we do not find a clear way to split the "SAE-σsp relationships" section. Surely, the presence of coarse dust aerosols o sea salt can lead to a reduction of SAE with increasing σsp. This is observed for example at IZO and PLA observatories which are affected by dust and sea salt, respectively. However, the same SAE vs. σsp relationship is not observed at MHD observatory which is also affected by sea salt. Moreover, the reduction of SAE with increasing σsp is also observed at sites not affected by dust such as CBW, KPS or MPZ for example.
We think that the analysis of the aerosol particles size distribution measured at the different observatories could probably help in properly separating the section. However, presenting the aerosol particle size distributions is far behind the scope of the present work.
In order to improve the readability of the section, the section was divided in different paragraphs.

**Page 19**
**104) Section 3.5 – my personal suggestion would be to switch the order of sections 3.4 and 3.5. Sections 3.1-3.3 talk about overall variability of the different parameters on an annual basis. It seems logical to talk next about the seasonal variability and tie it into annual variability. Once the individual parameter variability has been discussed then it makes sense to look at how those parameters co-vary (i.e., 3.5). Note: that's also the order those topics are presented in the conclusions.**

As suggested by the Reviewer the order of the sections 3.4 and 3.5 was changed in the revised version of the manuscript.

**105)     Line 25 - ad -> and**

Done

**Page 20**
**106)     Line 6 – start new paragraph**
Done

**107)     Lines 11-13 – "At the southern station of MSA the observed less pronounced seasonal cycle of SAE could be related with the Saharan dust outbreaks which contrast the PBL transport of fine particles observed at other mountain sites" it's unclear what is being said here; is MSA not impacted by dust?**

As suggested by the Reviewer the following sentence:

"At the southern station of MSA the observed less pronounced seasonal cycle of SAE could be related with the Saharan dust outbreaks which contrast the PBL transport of fine particles observed at other mountain sites."

was replaced with the following sentence:

"At MSA in Southwestern Europe, the observed less pronounced seasonal cycle of SAE could be due to the contribution of Saharan dust in spring/summer, which contrasts with the PBL transport of fine particles observed at other mountain sites during the warm season."

**108)** *Line 14 - "July-August being the Saharan dust outbreaks very" · July-August. The Saharan dust are very...*

The sentence was modified as follow:

"At IZO, the SAE reaches its lowest values during July-August in conjunction with the peak frequency of dust events (Rodríguez et al., 2015)."

*Page 21*

**109)** *Line 7 – "being the SAE" · with the SAE*
Done

*Page 22*

**110)** *Split the southern Europe paragraph into several paragraphs. Line 15 –start new paragraph. Line 27 start new paragraph.*
Done

*Page 23*

**111)** *Line 3 – citation for recommendation about having more than 10 years of data for trend analysis?*

This recommendation was based on a recent work from the Task Force on Measuring and Modelling (TFMM – CLRTAP). In this work the Chemical Coordinating Centre (CCC; http://www.nilu.no/projects/ccc/) presented a sensitivity study on Mann Kendall (MK) test demonstrating that the chances that the MK methodology detects a significant trend in a time series with 11 years of data are very small when the actual trend is of the order of 1 %/y. Furthermore, they show that with a trend of 2 %/y and even 3 %/y one could only be fairly certain to find a trend when the natural (inter-annual) variability is small (5 %) [https://wiki.met.no/_media/emep/emep-experts/mannkendall_note.pdf.]
However, this sensitivity study was performed analyzing 11yr and 23yr of data which were the periods selected for the EMEP Report 1/2016 on air pollution trends in the EMEP region between 1990 and 2012 (Colette et al., 2016; http://www.unece.org/fileadmin/DAM/env/documents/2016/AIR/Publications/Air_pollution_trends_in_the_EMEP_region.pdf).
Thus, this sensitivity study was not meant to provide general recommendations.
For this reason we prefer to remove the sentence from the manuscript.

**112)** *Line 20 – start new paragraph*

Done

*Page 24-25*

**113)** *This section (3.6.1) needs to be broken into smaller paragraphs! Line 9 – start new paragraph. Line 17 start new paragraph.*

Done

**114)** *Line 14-15 – "A statistically significant decreasing trend of $\sigma sp$ at IPR was also reported by Putaud et al. (2014) for the period 2002 – 2010."-> delete from text and put as footnote to Table 2.*
Done

**115)** *Line 16-16 – "As reported in Table 2 statistically significant decreasing trend for σsp is observed at around 50% of the stations considered here." -> delete from text – put the 50% number in the first sentence of the first paragraph on line 2 of this page.*
Done

**Page 25**
**116)** *Line 8 – start new paragraph*
Done

**117)** *Line 36 – start new paragraph*
Done

**Page 26**
**118)** *Line 9 start new paragraph about BF trends*
Done

**119)** *Line 21 start new paragraph*
Done

**Tables and Figures**
**120)** *Table 1 – Explain that the 'observatory code' is ACTRIS' code (or EBAS?) (or GAW?). Note where necessary any differences between GAW ids and station ids (e.g., Finokalia's GAW id is FIK and there may be others).*

The following sentence was added as footnote in Table 1:

(1) Observatory codes from EBAS; (2) GAW code: FIK; (3) GAW code: VAV; (4) GAW code: CES; (5) GAW code: MEL;"

**121)** *Figure 2 – you could put this on log scale*

Done

**122)** *Figures 2, 4 and 5 – how would this look if you had panes for different site types and then organized by geographical region? For example, you would have a pane for mountain sites and then sections for Nordic (empty), western (puy), central (jfj, cmn, hpb), etc. I think doing that would make it easier to see the east west shift and also commonalities among site types. You could keep the boxes colored by geographic region within each pane. You could do a similar thing with figure 6.*

Following the suggestion of the Reviewer Figures 2, 4, 5 have been modified. In the revised version of the manuscript the panes report different site types (Arctic, mountain, coastal, regional/rural and urban/sub-urban) and the boxes are colored in order to distinguish among the different geographic regions. For simplicity the ZEP station was included among the Nordic and Baltic stations.
Figure 6 and Figures 7, 8, 9 and Table 1 have been modified accordingly to the new selection of station categories.

Moreover, Figures and Tables in the Supporting material have been also accordingly modified. All Tables where modified the observatories ordered as a function of the station setting: from Arctic to Urban/sub-urban sites.

***Supplemental materials***

*123)     Table S1 – how is this table organized? It's not alphabetical or by geography.  Or by instrument type.*

The table was organized in alphabetical order in the revised version of the manuscript.

*124)     Table S2 - should provide number of points with RH>40% for each station (i.e., how many points are above the GAW and ACTRIS protocol value) as well as the number of points with RH>50%. Caption should state that table is organized by decreasing number of points with RH>50%.*

Following the Reviewer comment we changed the Table S2 in the revised version of the manuscript so that it presents the percentage of σsp data (not RH data) collected at 40%<RH<50%.
In the ACPD version of the manuscript, the Table S2 reports the percentage of RH data higher than 50%. This information is misleading given that often σsp data collected at RH>50% (or 40%) are removed by data providers or flagged as non valid because of instrument failure, calibration periods, unspecified contamination or local influence, etc.
Nevertheless, the % of σsp data collected at 40%<RH<50% is not small at some stations. This is clearly represented in Table S2.

In the table S3 in supporting material we report the percentage [%] of data coverage at the 28 ACTRIS stations included in this study. Percentages are calculated as the ratio between the number of scattering (backscattering) data used in this investigation and the total number of hours during the sampling period at each station. Removed data include data flagged as non valid by data providers (instrument failure, calibration periods, unspecified contamination or local influence, etc) or obtained at RH higher than 50%.

Consequently, in the revised version of the manuscript the Table S2 in supporting material reports the number of σsp hourly data used in this investigation and the % of hourly σsp collected at 40%<RH<50%.

The text in the manuscript describing Table S2 has been accordingly modified.

*125)     Figure S1 – should make the x-axes cover the same range (0-80%?) and draw a vertical line at 40% (GAW protocol value) and 50% (value chosen for this paper). Lines could be added indicating the median RH distribution as a function of season. Explain why different widths of bars on distribution plots (presumably because different numbers of data points). I think it would make more sense to use consistent widths for the distribution plots.*

We changed the Figure S1 in the revised version of the manuscript to include vertical lines for 40% RH, 50% RH, and medians RH values by season. X-axes was set to 0-80%. Consistent widths of bars are now used.

*126)     Table S3 – state in caption what colors lambda1, lambda2, and lambda3 correspond with. (not the wavelength because that obviously changes with instrument and time period)*
Done

*127)     Figure S2 – remove map – that's already a figure in the manuscript. Explain why different widths of bars on distribution plots (presumably because different numbers of*

*data points). I think it would make more sense to use consistent widths for the distribution plots.*

Figure was modified so that the width of the bars is equal for all stations. The map was removed. Colors were also added to represent the geographical location of each station.

**128)** *Figure S3 – remove map – that's already a figure in the manuscript. Explain why different widths of bars on distribution plots (presumably because different numbers of data points). I think it would make more sense to use consistent widths for the distribution plots.*

Figure was modified so that the width of the bars is equal for all stations. The map was removed. Colors were also added to represent the geographical location of each station.

**129)** *Figure S4 – remove map – that's already a figure in the manuscript. Explain why different widths of bars on distribution plots (presumably because different numbers of data points). I think it would make more sense to use consistent widths for the distribution plots.*

Figure was modified so that the width of the bars is equal for all stations. The map was removed. Colors were also added to represent the geographical location of each station.

**130)** *Figure S6b – explain the different colors of dots in the figure caption. Perhaps you could make the bars in figure S6a the same color as the dots in figure S6b and then use some other color for the dots in figure S6a*

Colors were added to the figure to represent the geographical location of each station.

**131)** *Figure S8 – has MTC instead of CMN*

Done. Moreover, the figure was modified and the color codes for different geographical locations were added.

---

## Author Comment (AC2) · 14 Mar 2018

Dear Editor, dear Referees, We would like to thank you for all your comments. This input has allowed us to refine the manuscript by adding more thorough detailed explanations, to correct some points and to improve in a large sense the manuscript.

Below the answers to the Referee's comments

*GENERAL REMARKS*
*The manuscripts approaches a European aerosol phenomenology on scattering properties of atmospheric aerosol particles from 28 ACTRIS sites, based on measurements with various types of integrating nephelometers. The manuscript focuses exclusively on ACTRIS sites in Europe, with the addition of Arctic and Antarctic stations and one mountain station in Bolivia, operated jointly by ACTRIS and local partners. The efforts for bringing this extensive data set together are huge and the richness of data is the major contribution of this work to an important scientific discussion on the long-term evolution of extensive and intensive aerosol properties in industrialized and rural regions of the world.*

*However, in its current version, the manuscript is very difficult to read and it is hard to follow any story line of the paper. Major revisions are required to make this manuscript acceptable for publication in ACP.*

*Major concerns refer to the organization of the manuscript, presentation of results and lacking of joint in-depth analyses of the observations in combination with existing publications on the long-term evolutions of aerosol optical properties.*
*Before resubmission of the manuscript, the following concerns should be considered:*

*Organization of the manuscript and the presentation of results:*

*1) The abstract is far too long and requires substantial shortening.*

Following the Reviewer comment, the Abstract (reported below) was slightly shortened in the revised version of the manuscript. However, given the amount of information provided by this work, it was difficult to further shorten the Abstract. Now the Abstract is less than one page (28 lines, 376 words).

"Abstract

This paper presents the light scattering properties of atmospheric aerosol particles measured over the past decade at 28 ACTRIS observatories which are located mainly in Europe. The data include particle light scattering ($\sigma_{sp}$) and hemispheric backscattering ($\sigma_{bsp}$) coefficients, scattering Ångström exponent (SAE), backscatter fraction (BF) and asymmetry parameter (g). An increasing gradient of $\sigma_{sp}$ is observed when moving from remote environments (Arctic/mountain) to regional and to urban environments. At regional level in Europe, $\sigma_{sp}$ also increases when moving from Nordic and Baltic countries and Western Europe to Central/Eastern Europe whereas no clear spatial gradient is observed for other station environments. The SAE does not show a clear gradient as a function of the placement of the station. However, a West to East increasing gradient is observed for both regional and mountain placements suggesting a lower fraction of fine-mode particle in Western/Southwestern Europe compared to Central and Eastern Europe where the fine-mode particles dominate the scattering. The g does not show any clear gradient by station placement or geographical location reflecting the complex relationship of this parameter with the aerosol particles physical properties. Both the station placement and the geographical location are important factors affecting the intra-annual variability. At mountain sites, higher $\sigma_{sp}$ and SAE values are measured in the summer due to the enhanced boundary layer influence and/or new particles formation episodes. Conversely, the lower horizontal and vertical dispersion during winter leads to higher $\sigma_{sp}$ values at all low altitude sites in Central and Eastern Europe compared to summer. These sites also show SAE maxima in the

summer (with corresponding g minima). At all sites, both SAE and g show a strong variation with aerosol particle loading. The lowest values of g are always observed together with low $\sigma_{sp}$ values, indicating a larger contribution from particles in the smaller accumulation mode. During periods of high $\sigma_{sp}$ values, the variation of g is less pronounced whereas the SAE increases or decreases, suggesting changes mostly in the coarse aerosol particle mode rather than in the fine mode. Statistically significant decreasing trends of $\sigma_{sp}$ are observed at 5 out of the 13 stations included in the trend analyses. The total reductions of $\sigma_{sp}$ are consistent with those reported for PM2.5 and PM10 mass concentrations over similar periods across Europe."

**2)** ***The results sections in Chapter 3 require a substructure to become more readable. Currently, paragraphs are too long and there is no line of arguments the reader can follow. Instead the paragraphs are highly descriptive and do not point at the key messages of the data analyses. In its current form, the reader likely misses a large part of the information contained in this manuscript.***

Following the Reviewer comment, Section 3 has been changed in the revised version of the manuscript. The presentation of the results is also different. In the revised version of the manuscript we present the discussion from cleaner sites to polluted sites. In the new figures 2, 4, and 5, the panes report different site types (Arctic, mountain, coastal, regional/rural and urban/sub-urban) and the boxes are colored in order to distinguish among the different geographic regions. All figures related to the two non-European stations of TRL and ZEP were moved in the supporting material.

This new way to present the results made the manuscript more readable. Moreover, new paragraphs in Section 3 were added.

Section 3 was reorganized and the following sub-sections were introduced:

3.1.1 $\sigma_{sp}$ at Arctic/Antarctic observatories;
3.1.2 $\sigma_{sp}$ at mountain observatories;
3.1.3 $\sigma_{sp}$ at coastal observatories;
3.1.4 $\sigma_{sp}$ at regional/rural observatories;
3.1.4 $\sigma_{sp}$ at urban/suburban observatories

3.2.1 Variability of SAE by geographical sector;
3.2.1 Variability of SAE by station type.

3.3.1 Variability of *g* by geographical sector
3.3.2 Variability of *g* by station type

Moreover, in the revised version of the manuscript the Section 3.4 presents the seasonal cycles, whereas the Section 3.5 presents the SAE and *g* vs. $\sigma_{sp}$ relationships.
Section 3.4 was divided in different subsection in order to present the results from cleaner environments to polluted environments:

3.4.1 Seasonal variability at Arctic observatories;
3.4.2 Seasonal variability at mountain observatories;
3.4.3 Seasonal variability at coastal observatories;
3.4.4 Seasonal variability at regional/rural observatories;
3.4.5 Seasonal variability at urban/sub-urban observatories.

**3)** ***The conclusions section is not well structured and a lot of information may get lost. Sharpening and shortening of this section is recommended. Finally, what are the key***

*points of the presented work? This should be clearly expressed.*

Following the Reviewer comment, the conclusions were modified in order to highlight the most important results from this study. However, given the large amount of information provided by this work, it was difficult to further shorten the Conclusion section. The modified Conclusion section is reported below:

[revised manuscript text omitted]

4) **Consideration of published results on the long-term evolution of aerosol optical properties: Although previous studies on aerosol scattering properties (see, e.g., Andrews et al. (2011), Collaud Coen et al., (2013) and Zanatta et al., (2016)) are**

*mentioned in the text, they have been included into the discussion only briefly. A discussion is missing (or got lost due to the current organization of the manuscript) whether the results presented here are in agreement with the published results, or whether they provide new findings, and then the question would be: where do differences come from? Finally, the discussion of the evolution of aerosol light scattering properties together with light absorption properties published in Zanatta et al. (2016) is completely missing.*

The comparison with Zanatta et al. (2016) is presented in the revised version of the manuscript. To compare with absorption measurements from Zanatta et al. (2016) the following sentence was added to the paragraph 3.1:

"The observed variation is consistent with the differences in particulate matter (PM) mass concentrations, PM chemical composition, particle number concentration and absorption coefficients observed across Europe, as described for example by Putaud et al. (2010), Asmi et al. (2011) and Zanatta et al. (2016).
Figures 3a and 3b show the relationship between the mean particle number concentration measured at different stations during 2008 to 2009 (and reported in Asmi et al. (2011)) and the mean σsp measured over the same period (where available). As reported in Figure 3, good correlations are observed between N50 (Figure 3a: mean/median particle number between 50 nm and 500 nm) and N100 (Figure 3b: mean/median particle number between 100 nm and 500 nm) and mean σsp. Figure 3c shows the relationship (for some stations) between absorption coefficients reported in Zanatta et al. (2016) and the total scattering. The good correlations reported in Figure 3c (especially high for the winter and autumn periods) suggest an increase of both scattering and absorption coefficients with increasing aerosol loading."

[Figure]

**Figure 3:** Relationship between: (a) N50 (mean particle number concentration between 50 nm and 500 nm), (b) N100 (mean particle number concentration between 100 nm and 500 nm), (c) absorption coefficient and mean aerosol particle total scattering coefficient. (a) and (b): data averaged over the period 2008 to 2009. For ZEP, BIR, KOS and PLA aerosol particle scattering measurements were not available during 2008 to 2009 and different periods were used. R2 values, highlighted in red, were obtained using the median values. (c) Data averaged as in Zanatta et al. (2016).

Comparison with the trend analysis performed by Collaud Coen et al. is reported in Section 3.6.3. In this Section we also provide possible reasons for the small discrepancies observed compared to the work from Collaud Coen et al. The sentence is reported below:

"These differences are thus likely due to the relative short period used in these trend analyses and the different sensitivity of the methods used to the presence of missing values or outliers especially at PAL where σsp is very low (cf. Fig. 2). For example, in this work the SAE calculated for PAL during the year 2007 was removed from the trend analysis due to the presence of too many extreme high SAE values, thus also likely explaining the difference observed for SAE with the work from Collaud Coen et al. (2013). Moreover, here we use de-seasonalized monthly means for trend analyses whereas Collaud-Coen et al. (2013) used de-seasonalized medians with different time granularity (3 days) thus likely affecting the comparison, especially over relatively short periods."

Moreover, the results of the comparison of trend analyses with the previous work from COllaud Coen et al. is also highlighted in the Conclusion section in the revised version of the manuscript.

Comparisons with the work from Andrews et al. (2011) are reported throughout the text. Below the sentences:

Variability of SAE:
"This high variability of SAE at mountain sites was also reported by Andrews et al. (2011). Andrews et al. (2011) reported SAE values from 11 mountaintop stations worldwide ranging from less than one to more than two."

Seasonal cycles at mountain sites:
"Similar findings were for example already reported by Nyeki et al. (1998) for JFJ and summarized by Andrews et al. (2011) for many mountain top stations worldwide and by Pandolfi et al. (2014) for MSA station."

g-$\sigma$sp relationship:

"The asymmetry parameter g shows the lowest values under very low $\sigma$sp suggesting the predominance of small fine mode particles. Andrews et al. (2011) reported similar g-$\sigma$sp relationships at different mountain sites and suggested that the removal of large particles by cloud scavenging or by deposition during transport could explain the observed low g under a clean atmosphere. They also suggested that the formation of new particles followed by condensation/coagulation could generate small but optically active particles. Here, we show that this behavior of BF or g as a function of $\sigma$sp was observed at all sites, not only at mountain sites. "

"Andrews et al. (2011), Pandolfi et al. (2014) and Sherman et al. (2015) showed that BF tends to decrease with increasing aerosol loading, consistent with the observed increase of g."

SAE-$\sigma$sp relationship
"Andrews et al. (2011), Pandolfi et al. (2014) and Sherman et al. (2015) showed that BF tends to decrease with increasing aerosol loading, consistent with the observed increase of g."

Variability of g by station type:
"On average, g values range between 0.49 to 0.64 at mountain sites with a mean value of 0.58±0.05. This value is consistent with the mean value of 0.61±0.05 reported by Andrews et al. (2011) at the mountain sites included in their work."

**MINOR COMMENTS**
5) *In section 2.2.2.1, the applied truncation correction is described. For light absorbing aerosol (single scattering albedo < 0.8) a method proposed by Müller et al. (2011) is applied. However, Massoli et al. (2009) presented a correction scheme particularly for light-absorbing aerosols measured with the TSI Model 3563 Integrating nephelometer. It should be briefly discussed why this approach has not been applied at the stations running TSI Model 3563 instruments.*

It is important to clarify that at all sites the correction for non ideal illumination and truncation are performed using the Angstrom based approach provided by Muller et al. (2011) for the Ecotech AURORA3000 and by Anderson and Ogren (1998) for the TSI nephelometer. Thus, no other methods for

correction are used by data providers. This is mostly due to the simplicity of the Angstrom based methods provided by the aforementioned papers. We think that the analysis of the different sensitivity of different correction schemes to different concentration of light absorbing species is far behind the objective of this work.

To clarify this point, the beginning of the section 2.2.2.1 has been modified as follow:

"2.2.2.1 Truncation correction
Data from the integrating nephelometers used here are corrected for non-ideal illumination of the light source (deviation from a Lambertian distribution of light) and for truncation of the sensing volumes in the near-forward (around 0-10°) and near-backward (around 170-180°) directions (Müller et al., 2009 and Anderson and Ogren, 1998). Correction schemes have been provided by Müller et al. (2009; 2011) for the RR M903 and Ecotech models M9003 and AURORA3000, and by Anderson and Ogren (1998) for the TSI3563. These schemes consist of a simple linear correction based on the scattering Ångström exponent (SAE) determined from the raw nephelometer data to take account of the size-distribution-dependent truncation error. It has been demonstrated that these simple correction schemes are accurate for a wide range of atmospheric aerosols and that the uncertainties in the corrections are not expected to be larger than 2% for an aerosol particle population with a single scattering albedos (SSA) greater than 0.8 (Bond et al., 2009)."

*6) In section 2.2.2.3, wavelength adjustments were conducted for sites where multiplewavelengths data do not exist. For that purpose SAE values were prescribed (1.5 for FKL and SIR and 2.0 for CMN). The choice of these SAE values should be justified.*

Given that the SAE is not available at SIR and FKL, in the revised version of the manuscript we calculated the differences after λ-adjustment using a reasonable range of SAE values (1.0 to 1.5). For CMN the SAE = 2 was calculated using data from the 3-λ nephelometer available from 2014 at CMN (before 2014 the 1-λ nephelometer was used)."

The section 2.2.2.3 was modified as follow:

"2.2.2.3 Available wavelengths
In this work we present and discuss the σsp, backscatter fraction (BF) and asymmetry parameter (g) measurements obtained using the green wavelength of the integrating nephelometers. The available wavelengths ranged from 520 nm (2 stations; CMN and VHL) to 550 nm (18 stations). Other wavelengths used are 525 nm (6 stations) and 532 nm (used at FKL until 2010; cf. Table 2). An exception is SIR, where only σsp values at 450 nm are available. The measurements of σsp reported here are not adjusted to 550 nm, which is generally the most common wavelength (e.g. Andrews et al., 2011) because of the different data availability of σsp and SAE at the measuring stations. As discussed in the following sections, the SAE is calculated for σsp data higher than 0.8 Mm-1, thus leading to different data coverage for σsp and SAE and preventing the adjustment of all measured σsp to 550 nm. Moreover, the SAE is not available at FKL and SIR (or at CMN until 2014) thus preventing any wavelength adjustment at these stations. Using the mean SAE calculated at those stations, where σsp is measured at wavelengths in addition to 550 nm (cf. Tables S4 and S5 in Supporting material), we estimate differences in the σsp values of less than 6% after adjusting to 550 nm. At FKL and SIR, where the SAE is not available, and assuming a reasonable SAE range between 1.5 and 1.0, the difference due to the adjustment to 550 nm is 4.9-3.0% at FKL and 26-18% at SIR. The higher difference at SIR is due to the fact that measurements at this station are performed at 450 nm. Finally, at CMN, the effect of the adjustment of σsp to 550 nm (from 520 nm) using a mean SAE of 2 (calculated using the 3-λ nephelometer data from 2014; cf. Table S5) is below 10%."

*7) Before resubmission, checking of the language by a native speaker is highly recommended.*

Following the Reviewer suggestion, the manuscript has been revised and edited by a native English speak.

---

## Author Response (AR2)

We thank the Reviewer#1 for his/her comments.

We added the following sentence to the Acknowledgments section to acknowledge the detailed and helpful comments to the manuscript from the two anonymous Referees and the co-editor Andreas Petzold.:

"We also thank the co-editor Andreas Petzold and the two anonymous reviewers for their constructive comments."

Below the answers to the Referee's comments. The modified text is highlighted with the yellow color in the 2d revision of this manuscript.

**The authors have done a great job of responding to almost all of my comments. The paper flows much better! I just have a few minor things that I think should still be addressed or at least considered.**

**1) Response #1 about comparing with Zanatta**
**That's a great addition. Lines of constant SSA could be added to the plot. I know you mentioned (response#2) that someone else is writing an SSA paper, but adding those lines shouldn't interfere with that SSA project.**

The sentence describing Figure 3 was modified as it follows:

"Figures 3a and 3b show the relationship between the mean particle number concentration measured at different stations during 2008 to 2009 (and reported in Asmi et al. (2011)) and the mean σsp measured over the same period (where available). As reported in Figure 3, good correlations are observed between N50 (Figure 3a: mean/median particle number between 50 nm and 500 nm) and N100 (Figure 3b: mean/median particle number between 100 nm and 500 nm) and mean σsp. Figure 3c shows the relationship (for some stations) between absorption coefficients reported in Zanatta et al. (2016) and the total scattering. The good correlations reported in Figure 3c (especially high for the winter and autumn periods) suggest an increase of both scattering and absorption coefficients with increasing aerosol loading. Figure 3c also reports the mean single scattering albedo (SSA). Figure 3c also reports the mean single scattering albedo (SSA). On average lower SSA is observed at IPR, whereas higher SSA is observed at the Nordic and Baltic VHL and BIR observatories."

Figure 3 (reported below) was modified accordingly to the Reviewer comment.

[Figure]

Figure 3: Relationship between: (a) N50 (mean particle number concentration between 50 nm and 500 nm), (b) N100 (mean particle number concentration between 100 nm and 500 nm), (c) absorption coefficient and mean aerosol particle total scattering coefficient. (a) and (b): data averaged over the period 2008 to 2009. For ZEP, BIR, KOS and PLA aerosol particle scattering measurements were not available during 2008 to 2009 and different periods were used. R2 values, highlighted in red, were obtained using the median values. (c) Data averaged as in Zanatta et al. (2016). Figure 3c also reports the geometric mean of SSA.

**2) Response #13 about skewness**
**The authors added some more text about skewness but it doesn't actually get at what I was asking about. ? I was hoping to see something more in the context of aerosols and less**

**mathematical. With the new text I still don't think it's really clear why a reader should care about skewness or what the implications are for aerosols. For example, what does skewness say about extreme events (if anything)? Or aerosol variability? Etc...**

To properly answer the Reviewer comment the following sentence:

"The skewness is defined as the third standardized moment of a probability distribution and it is a measure of the asymmetry of the probability distribution. Its value can be positive or negative. Positive skewness is usually observed for parameters which are defined to be positive and it indicates that the tail on the right side of the distribution is longer or fatter than that on the left side. Thus, for a right-skewed distribution, the mass of the distribution is concentrated on the left, and there is a higher probability of measuring a high value compared to a left-skewed distribution. Figure S2 in the Supporting Material shows the frequency and cumulative frequency distributions for ssp for each station, evidencing the presence of these right-skewed tails."

Was replaced with the following sentence:

"The skewness can be used to evaluate the asymmetry of a distribution. Positive skewness is usually observed for parameters which are defined to be positive and it indicates that the tail on the right side of the distribution is longer or fatter than that on the left side. Thus, for a right-skewed distribution, the mass of the distribution is concentrated on the left, and there is a higher probability of measuring a high value compared to a left-skewed distribution. For example Querol et al. (2009) used the skewness to assess the importance of Saharan dust outbreaks on $PM_{10}$ levels measured at different sites across the Mediterranean basin. They found a positive correlation between the calculated skewness and the net dust contribution to the measured $PM_{10}$ concentration (i.e. the strength of dust pollution episodes; cf. Fig. 6 in Querol et al., 2009). Figure S2 in the Supporting Material shows the frequency and cumulative frequency distributions for σsp for each station, evidencing the presence of these right-skewed tails. "

Querol, X., Pey, J., Pandolfi, M., Alastuey, A., Cusack, M., Pérez, N., Moreno, T., Viana, M., Mihalopoulos, N., Kallos, G., Kleanthous, S.: African dust contributions to mean ambient $PM_{10}$ mass-levels across the Mediterranean Basin, Atm. Env., 43, 4266–4277, doi:10.1016/j.atmosenv.2009.06.013, 2009.

**3) Response #15 about cut sizes**
**The authors did a bunch of work to look at this question of whether inlet cut size changes affected the resulting aerosol climatology at the sites they looked at. I don't think the figures they generated to respond to this question need to be added to the manuscript or supplemental materials but I think it would be useful to add one or two sentences to the manuscript to reflect this additional work they did. For example, right now there's a sentence that says "Consequently, the inlet change from PM1 to PM10 at KPS had probably only a minor effect on SAE." The authors have now looked at the data and can say something a little more definitive than 'probably'.**
**Also, there should be some mention of size cut changes in the methods section. I think the way to do this is to make an additional methods section: 2.2.2.4 'Inlet size cut changes'. There the authors should note that inlet size cut change could potentially affect both scattering and SAE and trends. I would move the parts of the second paragraph of section 3.2 describing the inlet size cut changes and previous investigations on the effects of size cut changes to this new section. Then I would suggest adding a few sentences to section 2.2.2.4 saying that the authors looked at climatologies of SAE (and scattering?) for different inlet sizes and for different time periods for and didn't see any obvious changes in the climatology as a function of size cut due to interannual variability (although I'm not sure that's totally true for KPS!). Then in section 3.2 you can just refer to section 2.2.2.4. Also in section 3.6 where you mention size cut issues you can again refer back to section 2.2.2.4.**

Following the Reviewer suggestion a new section (2.2.2.4) was added to the manuscript. All parts of the manuscript referring to the inlet changes were moved to this new section. The new section is reported below:

"2.2.2.4 Inlet size cut changes
It should be noted that any comparison of the $\sigma$sp and SAE values among the different stations and the presented trend analyses could be slightly biased by the different particle size cuts upstream of the integrating nephelometers used in this work (cf. Table 1). Currently, all ACTRIS integrating nephelometers measure whole air or PM10, with the exception of SIR, where the PM1 inlet is used. Whole air is currently measured at mountain observatories (BEO, CMN, JFJ, PUY, CHC), one coastal observatory (MHD) and one urban observatory (UGR) (cf. Table 1).
At some stations, the inlet was changed from whole air to PM10 at some point, namely at OPE, FKL and TRL. Given the lower scattering efficiency of aerosol particles larger than 10 $\mu$m, no important differences in the aerosol particle optical parameters should be expected between aerosol particles sampled with a whole air and a PM10 cut-off. At the other stations the inlet was changed during the measurement period from a cut-off lower than 10 $\mu$m (1 $\mu$m at KPS; 2.5 $\mu$m or 5 $\mu$m at PAL, MSA and MAD) to PM10. For PAL (where a median SAE of around 1.8 was measured; cf. Paragraph 3.2 and Table S5), Lihavainen et al. (2015a) assumed that the inlet changes (from PM5 to PM2.5 in 2005 and from PM2.5 to PM10, cf. Table 1) had only minor effects on scattering because the number concentration of coarse particles is very low at PAL. Similarly, the KPS observatory registers among the highest SAE values observed in the network (median value of around 2; cf. Paragraph 3.2 and Table S5) suggesting an aerosol particle size distribution dominated by fine particles. Moreover, at KPS, the inlet was changed in April 2008, less than 1.5 years after the measurements commenced, and thus likely has also a minor effect in the trend analyses and climatology performed at this site over the period 2006 to 2014. Two stations (MSA and MAD) changed the inlet from a PM2.5 diameter cut-off to PM10. For these two Southern European stations the inlet change may have had an effect on the SAE, especially during Saharan dust outbreaks, which are however sporadic events. Finally, the FKL observatory was removed from the trend analysis because the inlet was changed from whole air to PM10 in 2009, from PM10 to PM1 in 2011 and again from PM1 to PM10 in 2013 (cf. Table 1). These events likely had a major effect on the measured particle optical properties.
A sensitivity study (not shown) was performed to assess the effect of the inlet changes on the SAE values measured at the aforementioned stations. We looked at the climatology of SAE for different inlet sizes and for different time periods (with and without inlet size changes) and we did not observe any obvious change in the climatology as a function of size cut due to interannual variability.  Thus, despite the differences in the particle diameter cut-off, the comparison between the different stations seems feasible."

**Note: the manuscript says that MSA and MAD changed from PM2.5 to PM10 but the plot made for MSA in the response to the reviewers for comment#15 only shows box whiskers for changes between whole air and PM10. Not sure if that's a typo.**

We are sorry for this. It was a typo. MSA and MAD changed from PM2.5 to PM10.

**4) Section 3.1.3**
**Last sentence refers to 'Paragraph 3.5' – do the authors mean section 3.5?**

Yes, we mean section. The text was accordingly modified.

**5) Figure 9**
**The discussion about Figure 9 in Section 3.5 would probably be easier to follow if the y-axes for g and SAE were the same for all plots.**

The Figure 9 was changed accordingly to the Reviewer comment. The new Figure 9 is reported below.

[Figure]

**Figure 9**: Scatterplots between $\sigma_{sp}$ (x-axes) and SAE (right y-axes; red lines) and $g$ (left y-axes; black lines). Dashed lines represent median $\sigma_{sp}$ values at each station. Different colours highlight different geographical locations as in Figures 2, 4 and 5.

**6) Figure S1**
**The dashed vertical lines are really hard to see. Could you make them solid like the other two lines?**

The vertical lines in Figure S1 were changed accordingly to the Reviewer comment. The new Figure S1 is reported below.

[Figure]

**Figure S1:** Frequency distributions of sampled RH at ACTRIS observatories. Vertical lines: 40% GAW recommendation (black); 50% RH threshold used in this work (blue); median RH in winter (yellow); median RH in spring (magenta); median RH in summer (green); median RH in autumn (red).

---

## Author Response (AR3)

Reply to the co-Editor comment

**Thank you for the careful revision of the manuscript. It is now accepted for publication in ACP. I only request one very minor technical correction to the new Figure S1. You should indicate at some point that the x-scale refers to relative humidity. This is mentioned in the figure caption, but it should appear also in the figure itself.**

Following the co-Editor suggestion the Figure S1 was changed. The new Figure S1 is reported below:

[Figure]

**Figure S1:** Frequency distributions of sampled RH at ACTRIS observatories. Vertical lines: 40% GAW recommendation (black); 50% RH threshold used in this work (blue); median RH in winter (yellow); median RH in spring (magenta); median RH in summer (green); median RH in autumn (red).